# Eurodelta multi-model simulated and observed PM trends in Europe in the period of 1990-2010

Svetlana Tsyro[1], Wenche Aas[2], Augustin Colette[3], Camilla Andersson[4], Bertrand Bessagnet[3,†], Giancarlo Ciarelli[5], Florian Couvidat[3], Kees Cuvelier[6], Astrid Manders[7], Kathleen Mar[8], Mihaela Mircea[9], Noelia Otero[8,‡], Maria-Teresa Pay[10], Valentin Raffort[11], Yelva Roustan[11], Mark R. Theobald[12], Marta G. Vivanco[12], Hilde Fagerli[1], Peter Wind[1,13], Gino Briganti[9], Andrea Cappelletti[9], Massimo D'Isidoro[9], and Mario Adani[9]

[1]Norwegian Meteorological Institute, NO-0313, Oslo, Norway
[2]Norwegian Institute for Air Research (NILU), Box 100, 2027 Kjeller, Norway
[3]INERIS, National Institute for Industrial Environment and Risks, Parc Technologique ALATA, 60550, Verneuil-en-Halatte, France
[4]Swedish Meteorological and Hydrological Institute, 60176 Norrköping, Sweden
[5]Institute for Atmospheric and Earth System Research/Physics, Faculty of Science, University of Helsinki, Finland
[6]ex European Commission, Joint Research Centre (JRC), Ispra, Italy
[7]TNO, Dept. Climate, Air and Sustainability, P.O. Box 80015, 3508 TA Utrecht, the Netherlands
[8]Institute for Advanced Sustainability Studies, Postdam, Germany
[9]ENEA, Italian National Agency for New Technologies, Energy and Sustainable Economic Development Via Martiri di Monte Sole 4, 40129 Bologna, Italy
[10]BSC, Barcelona Supercomputing Center, Centro Nacional de Supercomputaciòn, Nexus II Building, Jordi Girona, 29, 08034 Barcelona, Spain
[11]CEREA, École des Ponts, EDF R&D, Île-de-France, France
[12]CIEMAT, Atmospheric Pollution Unit, Avda. Complutense, 22, 28040 Madrid, Spain
[13]Faculty of Science and Technology, University of Tromsø, Tromsø, Norway
[†]Now at European Commission, Joint Research Centre (JRC), Ispra, Italy
[‡]Now at the University of Bern, Switzerland

**Correspondence:** Svetlana Tsyro (svetlana.tsyro@met.no)

**Abstract.**

The Eurodelta-Trends multi-model experiment, aimed to assess the efficiency of emission mitigation measures in improving air quality in Europe during 1990-2010, was designed to answer a series of questions regarding European pollution trends. i.e. were there significant trends detected by observations? Do the models manage to reproduce observed trends? How close is the agreement between the models and how large are the deviations from observations? In this paper, we address these issues with respect to PM pollution. An in-depth trend analysis has been performed for $PM_{10}$ and $PM_{2.5}$ for the period of 2000-2010, based on results from six chemical transport models and observational data from the EMEP (Cooperative Programme for Monitoring and Evaluation of the Long-range Transmission of Air Pollutants in Europe) monitoring network. Given harmonization of set up and main input data, the differences in model results should mainly result from differences in the process formulations

within the models themselves, and the spread in the models simulated trends could be regarded as an indicator for modelling uncertainty.

The model ensemble simulations indicate overall decreasing trends in $PM_{10}$ and $PM_{2.5}$ from 2000 to 2010, with the total
reductions of annual mean concentrations by between 2 and 5 (7 for $PM_{10}$) $\mu$g m$^{-3}$ (or between 10 and 30%) across most of Europe (by 0.5-2 $\mu$g m$^{-3}$ in Fennoscandia, north-west of Russia and Eastern Europe) during the studied period. Compared to $PM_{2.5}$, relative $PM_{10}$ trends are weaker due to large inter-annual variability of natural coarse PM within the former. The changes in the concentrations of PM individual components are in general consistent with emission reductions. There is a reasonable agreement in PM trends estimated by the individual models, with the inter-model variability below 30-40% over
most of Europe, increasing to 50-60% in northern and eastern parts of EDT domain.

Averaged over measurement sites (26 for $PM_{10}$ and 13 for $PM_{2.5}$), the mean ensemble simulated trends are -0.24 and -0.22 $\mu$g m$^{-3}$ yr$^{-1}$ for $PM_{10}$ and $PM_{2.5}$, which are somewhat weaker than the observed trends of -0.35 and -0.40 $\mu$g m$^{-3}$ yr$^{-1}$, respectively, partly due to models underestimation of PM concentrations. The correspondence is better in relative $PM_{10}$ and $PM_{2.5}$ trends, which are -1.7 and -2.0 % yr$^{-1}$ from the model ensemble and -2.1 and -2.9 % yr$^{-1}$ from the observations,
respectively. The observations identify significant trends (at 95% confidence level) for $PM_{10}$ at 56 % of the sites and for $PM_{2.5}$ at 36% of the sites, which is somewhat less that the fractions of significant modelled trends. Further, we find somewhat smaller spatial variability of modelled PM trends with respect to the observed ones across Europe and also within individual countries.

The strongest decreasing PM trends and the largest number of sites with significant trends is found for the summer season,
according to both the model ensemble and observations. The winter PM trends are very weak and mostly insignificant. One important reason for that is the very modest reductions and even increases in the emissions of primary PM from residential heating in winter. It should be kept in mind that all findings regarding modeled versus observed PM trends are limited the regions where the sites are located.

The analysis reveals a considerable variability of the role of the individual aerosols in $PM_{10}$ trends across European coun-
tries. The multi-model simulations, supported by available observations, point to decreases in $SO_4^{-2}$ concentrations playing an overall dominant role. Also, we see relatively large contributions of the trends of $NH_4^+$ and $NO_3^-$ to $PM_{10}$ decreasing trends in Germany, Denmark, Poland and the Po Valley, while the reductions of primary PM emissions appears to be a dominant factor in bringing down $PM_{10}$ in France, Norway, Portugal, Greece and parts of the UK and Russia. Further discussions are given with respect to emission uncertainties (including the implications of not accounting for forest fires and natural mineral dust by
some of the models) and the effect of inter-annual meteorological variability on the trend analysis.

## 1   Introduction

The Convention on Long-range Transboundary Air Pollution (LRTAP), signed in 1979, addresses some of the major environmental problems of the UNECE region through scientific collaboration and policy negotiation (UNECE, 2004). Parties develop policies and strategies to combat the release of pollutants in the atmosphere through exchanges of information, consultation,

research and monitoring. During the 1980s, 1990s and 2000s, the concentrations of particulate matter (PM) were decreasing due to the decrease of secondary inorganic aerosols (SIA) as a result of the reductions of the emissions of their gaseous precursors in order to address the acidification and eutrophication problems (Fagerli and Aas, 2008; Aas et al., 2019), mainly of SOx due to the 1st and 2nd Sulphur Protocols, and also NOx and NH3 in line with the 1999 Gothenburg Protocol to Abate Acidification, Eutrophication and Ground-level Ozone (UNECE, 2004). The emissions of primary PM were not then regulated, but still were decreasing as a side-effect of the reductions of gaseous pollutants. In the end of 1990s, the issue of adverse effects of particulate pollution on human health came into focus, and in 2012, emissions of primary $PM_{2.5}$ were included in the revised Gothenburg Protocol, stating that fine particulate matter is "the pollutant whose ambient air concentrations notoriously exceed air quality standards throughout Europe".

The Eurodelta-Trends (EDT) multi-model experiment, involving eight chemical transport models (CTMs), has been designed in order to better understand the evolution of air pollution and its drivers since the early 1990s. The main objective of the experiment is to assess the efficiency of air pollutant emissions mitigation measures in improving regional scale air quality in Europe. The multi-model trend analysis is a contribution to the assessment of the evolution of air pollution in the EMEP region over the 1990-2012 period coordinated by the Task Force on Monitoring and Modelling (TFMM) of EMEP (Cooperative Programme for Monitoring and Evaluation of the Long-range Transmission of Air Pollutants in Europe). The synthesis of the observational and modelling evidences of atmospheric composition and deposition change in response to actions taken by to control emissions were given in Colette et al. (2016).

A number of studies of European (and global) PM trends for the 1990s and 2000s have been performed and published recently. Some studies analysed observed PM trends (e.g. Guerreiro et al., 2014; Barmpadimos et al., 2012; Cusack et al., 2012; EEA, 2009; Crippa et al., 2016), including those derived from remote sensing observations (Van Donkelaar et al., 2015), whereas a limited number of analyses also included model simulations (e.g. Colette et al., 2011; Mortier et al., 2020; Colette et al., 2021; Myhre et al., 2017). Rather a large spread of observed and modelled PM trends, both decreasing and increasing, has been reported for the period between 1998-2002 and 2008-2014. In those studies, the setup of model runs was only partly harmonised, i.e. the models in the same study used the same emissions, but otherwise different meteorology, grid resolution etc. Analysis of EMEP observed 2002-2012 trends, also performed under TFMM coordination by Colette et al. (2016), reported the median trends of -0.35 $\mu$g m$^{-3}$ yr$^{-1}$ $PM_{10}$ and -0.29 for $PM_{2.5}$, resulting in the reduction over the period by -29 and -31%, respectively, with 95% probability. As we discuss in this paper, being overall consistent with the earlier trend assessments, the results presented here are believed to be more robust as they rely on a multi-modelling approach.

The main science and policy questions addressed by the EDT modelling experiment are formulated in Colette et al. (2017a), in which also the design and technical specifics of the modelling exercise are described in detail. The studied period covered a 21-year time-span, from 1990 through 2010, and in total eight regional CTMs have participated. In this paper, we present the results of trend study with respect to Particulate Matter (PM) pollution in Europe. An in-depth trend analysis for $PM_{10}$ and $PM_{2.5}$ has been performed for a period of 2000-2010, based on multi-model simulations and EMEP monitoring data. The shorter period for PM trend study than the 1990-2010 EDT period was chosen due to the lack of appropriate $PM_{10}$ and $PM_{2.5}$ observations prior to 2000. Not all of the eight EDT models had resources to perform all simulations (Sec. 2.1, therefore trend

analysis presented in this work are based on the results from six of the models. Also, multi-model simulated PM trends during the whole 1990-2010 period are briefly discussed here. The strength of the presented assessment is that the model ensemble simulated PM trends represent more a robust estimate as compared to either of the individual models, while the multi-model simulations allowed us investigating into the variability of modelled results, obtained under this controlled setup. Finally, the model simulations allow interpreting PM trends in term of the trends in the individual aerosols. This is a valuable contribution to better understanding the correspondence between emission changes and PM concentration levels across Europe, given the lack of observational data on PM chemical composition.

The paper is structured as follows: Section 2 describes the methods used, including brief information on the models, runs' setup, observations and trend calculations; Section 3 summarizes model evaluation with respect to PM; Section 4 presents emission trends; Section 5 is dedicated to PM 2000-2010 trend analysis for the whole Europe and for set of measurement sites, discusses PM seasonal trends and the relative contribution of PM components; in Section 6 we show modelled PM trends for the 1990-2010 period. Further discussion of the result is given in Section (including emission uncertainties and effect of meteorological variability); and finally the main outcomes and findings can be found in Section 8.

## 2  Methods

### 2.1  Models, runs setup

The trend analysis is based on the results from six of the EDT models, namely the ones which provided a complete series of 2000-2010 simulations. Those models are CHIMERE (CHIM), EMEP MSC-W (EMEP), LOTOS-EUROS (LOTO), MATCH, MINNI and Polair3D (POLR). These models, with the exception of POLR, also performed simulations for the 1990-1999 period. A comprehensive description of the models that participated in the Eurodelta-Trends experiment, the simulations setup, input data and the overview of the computations performed is given in Colette et al. (2017a).

Briefly, the setup and input data for the EDT simulations were harmonized as far as possible. The models performed the simulations on the same grid with a resolution of $0.25° \times 0.4°$ in latitude-longitude coordinates. The simulations were driven by the same meteorological input from hindcast simulations of the CORDEX project (Jacob et al. (2014) and Stegehuis et al. (2015)) using the WRF (Weather Research and Forecast) model (Skamarock et al., 2005) at $0.44° \times 0.44°$ resolution and using boundary conditions from ERA-Interim reanalysis (Dee et al., 2011). The exceptions were LOTO and MATCH, which used ERA-Interim reanalysis donwscaled respectively by RACMO2 (Van Meijgaard et al., 2012) and HIRLAM (Dahlgren et al., 2016).

Furthermore, the models used the same gridded anthropogenic emissions of $SO_2$, $NO_x$, $NH_3$, NMVOC, CO, $PM_{10}$ and $PM_{2.5}$ (Terrenoire et al. (2015) and Bessagnet et al. (2016)). The national emissions were based on the ECLIPSE_V5 dataset, constructed by the Greenhouse Gasses and Air pollution INteraction and Synergies (GAINS) model (Amann et al. (2011), Amann (2012), Klimont et al. (2016), Klimont et al. (2017)) and provided in SNAP (Selected Nomenclature for reporting of Air Pollutants) sectors. Spatial distribution of the national sectoral emissions was performed by INERIS applying auxiliary information which included road maps (for SNAP sector 7), shipping routes (for SNAP 8) and population density (for SNAP

2), the European Pollutant Release and Transfer Register (for SNAP 1, 3, 4), TNO-MACC inventory for $NH_3$ emissions, as well as bottom-up emission inventories for the UK and France (see details in Colette et al. (2017a) and references therein).

Time changes in the spatial distribution was accounted for only for industrial emissions. Vertical distribution and temporal profiles for the emissions used in the model simulations were those used in the EMEP model standard setup (Simpson et al., 2012). The ECLIPSE_V5 emissions were available for the years 1990, 1995, 2000, 2005 and 2010, while for the intermediate years the emissions were derived through linear interpolations (Colette et al., 2017a). For temporal distribution of ECLIPSE annual emissions, the models applied the same monthly and hourly profiles based on Denier van der Gon et al. (2011); they

also used the same static vertical profiles for the emissions, based Bieser et al. (2011), applied per SNAP activity sector (none of the models included explicit plume rise simulations). Regarding, chemical speciation of $PM_{10}$ and $PM_{2.5}$, the models were allowed to use their own preferred factors to split PM emission to elemental and primary organic carbon (e.g. based on Kuenen et al. (2014), or as in Simpson et al. (2012), see Summary Table in Fig. A16).

At a rather late stage of the experiment, an error was detected in the emissions of primary particulate matter from international

shipping and also from Russia and North Africa for the period 1991-1999. Since this error was identified late in the analysis process it was not possible to re-run the simulations with corrected emissions. The additional analysis of the impact of this error carried out with the CHIMERE model showed that these errors are relatively small compared with the overall uncertainty of the model estimates and the uncertainty of the observations (see more details in Theobald et al. (2019)). Nevertheless, the main focus of this paper is on the analysis of PM trends in the course of the 2000s, i.e. the period for which model results were

not affected by the emission error.

Natural emissions of biogenic VOCs, soil NOx, sea salt and mineral dust were calculated or prescribed within the models individually. Online computations of windblown dust from erodible soils were performed by EMEP, LOTO and MINNI, whereas the other models included solely mineral dust from boundary conditions. Emissions from forest fires and volcanoes were not included in the EDT simulations as the main research focus was to investigate whether the models could reproduce the trends

caused by anthropogenic emission changes and changes in meteorology (see discussions on possible implications of . Finally, the common boundary conditions, provided by the EMEP group were based mainly on a climatology of observational data (Simpson et al., 2012). Given harmonization of set up and main input data (with a few exceptions), the differences in model results should mainly result from differences in the process formulations within the models themselves.

### 2.2    Observations

The observations collected at the EMEP monitoring network are annually reported to the Chemical Coordinating Centre of EMEP (Tørseth et al., 2012). All submitted observational data, after routine quality and consistency control, are available in EBAS (http://ebas.nilu.no). At most of the sites, 24-hourly samples were taken on a daily basis (See Table A1). Most of the sites used a gravimetric method for both size fractions, though some used monitors. The same methods are used during the whole period. Details about site locations and applied methods are found in Table A1.

As documented in Colette et al. (2016), the selection criteria for sites included in the trend analysis were: i) the data capture should be at least 75 % for a specific year to be counted and ii) the number of these counted years should be at least 75 % of

the total number of years in the period, and had undergone visual screening tests. The datasets used in this work include yearly measurements of observed trends from respectively 26 and 13 sites of $PM_{10}$ and $PM_{2.5}$ for the period 2000-2010 (Table A1 and Fig. 5, upper panel).

Among those 'trend-sites', $PM_{10}$ observations are available for all eleven years of the 2000-2010 period at 16 sites, and at 4 sites for $PM_{2.5}$ (Table A1). The reason for gap years is either PM was not measured in that year, or the criterion of 75% for data coverage was not satisfied. For most of the sites with incomplete data series, 2000 is a gap-year, as PM monitoring was not started before in 2001 at those sites. The other gap years are: 2009 at the Czech CZ0003R site, 2003 and 2004 at the British GB0043R, and 2009 for PM10 and 2010 for PM2.5 at the Swedish SE0002R (for detailed info see Table A1).

## 2.3  Trend calculation

The Mann Kendall (MK) method (Mann (1945) and Kendall (1975)) has been applied to both modelling results and observed data for identification of significant trends. The linear trends have been calculated using the Theil-Sen slope method (known to be robust to outliers), applying the probability level of 95% as a threshold for trend significance. The trend calculation method used here is consistent with that in trend assessment reported in Colette et al. (2016). In addition to absolute concentration

trends, relative trends have been calculated using an estimated concentration at the start of the period (i.e. the year of 2000) as reference (see Appendix A3 in Colette et al., 2016). This concentration value corresponds to PM concentration in 2000 according to the trend line, and is considered to be less sensitive to inter-annual variability than the actual observed or modelled ones.

    A synthetic testing of the efficiency of MK methodology to identify significant trends and estimate Sen's slopes has been per-

formed (S. Solberg, *personal comm.*, https://https://wiki.met.no/_media/emep/emep-experts/mannkendall_note.pdf). It showed that the chance that the MK method detects the long-term trend decreased for shorter data-series, large natural variability and relatively weak trends. The extent to which these factors could have affected the results of our trend analysis is discussed in Sec. 7.3. Furthermore, the aforementioned document also demonstrates that averaging significant trends only would overestimate mean absolute trends, therefore both significant and insignificant trends have been included when calculating site-average

PM trends.

## 3  Models evaluation

Model simulated $PM_{10}$ and $PM_{2.5}$ have been evaluated against observations at the trend-sites (26 and 13 respectively) for the years from 2000 through 2010, and averaged over the measurement sites performance statistics in terms of annual mean bias and spatial correlations are summarized in Fig. 1 and Tables A2 and A3 (Appendix A).

Figure 1 shows the relative biases (in %) for the individual model and the ensemble mean. The modelled $PM_{10}$ and $PM_{2.5}$ tend to be biased low compared to the observations (marked by blue colours of different intensity). On average, the model ensemble underestimates annual mean $PM_{10}$ by 12% and $PM_{2.5}$ by 14 % over the period 2000-2010 (rather different biases for 2000 are due to fewer sites with data). $PM_{10}$ mean relative biases for the individual models are in a range of 5-11 %, that

is somewhat smaller than their biases of 5-20 % for $PM_{2.5}$ (with POLR standing out with $PM_{10}$ bias of -31% as erroneously simulated coarse sea salt had to be excluded).

Furthermore, we find a quite moderate year-to-year variability of the model ensemble bias, namely between -7 and -18 % for $PM_{10}$ and between -2 and -20 % for $PM_{2.5}$. This robustness in PM simulation also applies to the individual models, i.e. the inter-annual bias variations are mostly within 5 % (up to 10 %). The consistency in terms of bias can be noticed between the models (e.g. smaller underestimation of $PM_{10}$ for 2000, 2001, 2007, 2008 and 2009, whilst slightly larger underestimation for the years 2003, 2006 and 2010, characterised by elevated PM levels).

The average annual coefficients of spatial correlation (R) are 0.54 (0.41 - 0.58) for $PM_{10}$ and 0.65 (0.58-0.72) for $PM_{2.5}$. Similar to model biases, the correlation varies only moderately between the years and the models (Tables A2 and A3). Models' evaluation for the individual aerosol components and their gaseous precursors can be found in the other EDT publications (e.g. Ciarelli et al., 2019; Theobald et al., 2019).

## 4 Emission trends

The graphs in Fig. A1 present the changes in European annual emissions, used in this work. The total emissions of aerosol gaseous precursors $SO_2$, $NO_x$ and $NH_3$ and primary fine and coarse PM ($PM_{2.5}$ and $PM_{10-2.5}$) are shown for the whole period of EDT study, i.e. 1990-2010. The total emissions of all pollutants decrease during this period, although at different rates. From 1990 to 2010, the greatest decrease by 69 % is in $SO_2$ emissions, following by $NO_x$ emissions which are decreased by 39 %. The reduction in $NH_3$ emissions is rather moderate 15 %. Quite considerable decrease is seen in primary PM emissions, which go down by 67 and 47 % for coarse PM and $PM_{2.5}$ respectively.

During the period of 2000-2010, which is in a focus of this publication, the total emission decreases are: 37 % for $SO_2$, 17 % for $NO_x$, 6 % for $NH_3$, 27 % for $PM_{2.5}$, 36 % for coarse PM, 33 % for NMVOC. For EU area, where the measurement sites with PM observations available for the trend analysis are located, $SO_2$ is reduced by 24 %, $NO_x$ by 22 %, $NH_3$, $PM_{2.5}$ and coarse PM by 10 % during the same period.

Further details on emission changes across the EDT domain are provided in Fig. A2, which shows the maps with annual mean trends in the emissions of primary PM and their gaseous precursors during 2000-2010 and 1990-2010. During the period of our attention 2000-2010, the emissions of $SO_2$ and $NO_x$ go down in all countries, but there are many hot-spots with upward trends (also in some Eastern and south-Eastern countries for $NO_x$). The negative trends of $SO_2$ emissions are 3-7 % $yr^{-1}$ in most countries, exceeding 7 % $yr^{-1}$ in Italy, Hungary, Portugal, Ireland and parts of Sweden and Finland (below 3% $yr^{-1}$ in Western Balkan, Norway and Russia). $NO_x$ emissions show a reduction of 3-5% $yr^{-1}$ in Central Europe and Italy, going up 5-7 % $yr^{-1}$ and above in Sweden, some spots in Finland, Denmark, the UK and Portugal. $NO_x$ decreases less (by 1-3 % $yr^{-1}$) in Norway, parts of Spain and Eastern Europe, and increases by 1-3 % $yr^{-1}$ in Russia, Belarus, parts of Poland. $SO_2$ and $NO_x$ emissions from international shipping decrease in the North Atlantic and the Baltic Sea, but increase in the Mediterranean Sea. Also $NH_3$ emissions show negative trends in most of the domain, with decrease by 0.5-3 % $yr^{-1}$ in most of Europe (by 3-5 %

yr$^{-1}$ in Denmark), but they remain nearly unchanged in Scandinavia and even increase by 1-3 % yr$^{-1}$ in Belarus, Lithuania, Estonia and Bosnia and Herzegovina and by (0.5-1.5 % yr$^{-1}$) in Poland.

During 2000-2010, PM$_{2.5}$ emissions show downward trends in Central Europe and Norway (-(3-5) % yr$^{-1}$) and in the rest of Eastern Europe, Spain and Scandinavia (-(1-3)%/yr), while they go up (by 1-4 % yr$^{-1}$) in Italy, Poland, Denmark, Bosnia and Herzegovina, Serbia, Moldova and Turkey. Finally, the largest decrease in coarse PM emissions is in Portugal (by (3-5) % yr$^{-1}$), and in the UK, Belgium and parts of Central and South-Eastern Europe (by (1-5) % yr$^{-1}$), but there are hot-spots with 1-4 % yr$^{-1}$ emission increase in the latter areas. PM coarse emissions also increase in parts of Scandinavia and Finland, in the Baltic countries and in Russia (by 1-4 % yr$^{-1}$), whereas they change little elsewhere.

## 5    PM trends for the period 2000-2010

### 5.1    Modelled and observed European trends

Figure 2 shows the maps of mean annual trends (Sen's slopes) of PM$_{10}$ and PM$_{2.5}$ over Europe for the period of 2000-2010, calculated by the ensemble of six models (mean of EMEP, CHIM, LOTO, MINNI, MATCH and POLR) and observed at EMEP sites. The trends are presented in terms of absolute (in $\mu$g m$^{-3}$ yr$^{-1}$) and relative to the starting year of 2000 (% yr$^{-1}$) annual changes. Significant trends are represented by coloured contour maps (modelled) and triangles (observed), whereas the insignificant trends are shown as grey areas and circles respectively.

The model results over the simulation domain and the observations at the trend-sites show overall decreasing trends of PM$_{10}$ and PM$_{2.5}$ levels between 2000 and 2010. The modelled mean decreasing trends vary over the studied domain from below 0.1 $\mu$g m$^{-3}$ yr$^{-1}$ in northern Europe to 0.1-0.3 $\mu$g m$^{-3}$ yr$^{-1}$ in the eastern parts, and to 0.3-0.5 $\mu$g m$^{-3}$ yr$^{-1}$ in central Europe and most of the UK, with PM$_{2.5}$ downward trends being just slightly smaller than those for PM$_{10}$. Starting from the concentration levels in 2000, the mean relative decreasing trends range mostly from 0.1 to 0.3 % yr$^{-1}$ for PM$_{10}$ and PM$_{2.5}$. Compared to the distribution of absolute trends, steeper slopes of relative decreasing trends are also seen in the southern parts of Fennoscandia in addition to Central Europe and the UK.

The 6-model simulated mean trends are in general comparable to the observed ones, still some discrepancies are seen in their geographical distribution. For instance, quite strong decreasing trends for PM$_{10}$ and for PM$_{2.5}$ are observed at three of the Spanish sites, while the model ensemble hardly indicates any significant trends over Spain. It should be noted that the models do calculate negative PM trends for the Spanish sites (as seen in A7), but due to considerable inter-annual variability most of them are not identified as significant. Furthermore, the models calculated strongest decreasing trends of 0.5-0.7 $\mu$g m$^{-3}$ yr$^{-1}$ for PM$_{10}$ and PM$_{2.5}$ in Portugal and Benelux, but no measurements were available to validate the modelled results. For Germany, the slopes of observed trends are similar or somewhat lower that the modelled, but unlike the model results, none of the observed trends was identified as significant. In the next sections, the trends at the individual monitoring sites will be considered more closely.

Figure 3 illustrates the inter-model variability in PM trend slopes, showing the Coefficient of Variability (COV) of the trends simulated by the individual models relative to the ensemble mean (STD/ensemble mean) for PM$_{10}$ and PM$_{2.5}$. The COV is

somewhat larger for the modelled $PM_{10}$ trends compared to those for $PM_{2.5}$. This reflects larger uncertainties in modelling of the coarse fraction of PM, which is mostly due to natural origin, i.e. sea salt and windblown dust. As shown in Table A16, the models used different parameterisations for the source functions of natural aerosols (also some of them did not include online simulations of windblown dust, but only mineral dust from boundary conditions).

The lowest spread in the modelled trends (below 20 %) appears in Central Europe (Germany, Czech Republic), and also parts of Spain, northern regions of Italy and in the very south of Scandinavia for $PM_{2.5}$. Those regions correspond with the strongest simulated PM trends. Otherwise, the COV is 20-40 % over most of Europe, increasing to 40-60 % in Poland, western and northern Fenno-Skandia, the Baltic countries and parts of Russia, where the modelled trends are relatively low or insignificant.

The maps with annual mean $PM_{10}$ and $PM_{2.5}$ trend slopes calculated by the individual models are provided in the Appendix. Figures A3 and A4 show the Sen's slopes of $PM_{10}$ and $PM_{2.5}$ simulated by the six models and the observed trends for the period of 2000-2010. The significant modeled slopes are in general quite close to each other, indicating decreasing from 2000 to 2010 trends. Also the spatial variability of the Sen's slopes in the individual models' results shows much similarity, with the strongest decreasing trends identified in Central Europe (in particular in the Benelux countries and Germany). The EMEP and LOTO calculated respectively the largest and the weakest negative mean trend slopes, as well as the largest and the smallest fraction of the modelling domain with significant PM trends, namely 45 % and 57 % grid-cells according to EMEP and 17 and 38 % according to LOTO for respectively $PM_{10}$ and $PM_{2.5}$, with the results from the other for models lie between those values. As most of the input and setup for the model runs were harmonized (Sec. 2.1), the differences we see here are due to differences in model configurations and process descriptions (see Table A16), leading to different responses of the models to the changes in emissions and inter-annual meteorological variability. Differences in the formulations of secondary aerosol formations (inorganic and organic) can be pointed at as a very important reason for discrepancies in PM modelled trends. Differences in aerosol removal, in particular wet scavenging efficiency, also play a certain role (besides LOTO and MATCH were driven with different meteorology). Further note that the models have a different thickness of the lowest layer which affects the concentrations, removal and transport distances of primary PM and their gaseous precursors.

Relative to the year 2000, all the models simulate stronger trends for $PM_{2.5}$ compared to $PM_{10}$, as seen in Fig. A5 and A6. This is to be expected as the natural contribution, which is strongly meteorology dependent, is greater in $PM_{10}$. The distribution patterns of relative trends from the models are in general similar to those for corresponding absolute trends. However, there is a difference between the models in the locations of their strongest simulated relative trends, namely in Central Europe (e.g EMEP, MINNI, POLR) or in Northern Europe (e.g. CHIM, LOTO, MATCH). The fraction of the EDT domain with significant PM trends simulated with the individual models ranges from 17 (LOTO) to 45 (EMEP) % for $PM_{10}$ and from 38 (LOTO and MINNI) to 57 (EMEP) % for $PM_{2.5}$.

Figure 4 presents observed and modelled annual mean series of $PM_{10}$ and $PM_{2.5}$ at the trend-sites for the period 2000-2010. Shown are the mean values from the 6-model ensemble (dotted curves in Fig. 4a) and from the individual models' results (Fig. 4b and c). Note that in the year of 2000 is a gap-year at for 7 out of 26 sites for $PM_{10}$ and at 8 out of 13 sites for $PM_{2.5}$, as described in Section 2.2. In particular, none of Spanish sites are included for 2000, bringing in some inconsistency in site averaged $PM_{10}$ and $PM_{2.5}$ annual mean series.

Although they are underestimated with respect to the observations, the annual mean concentrations of $PM_{10}$ and $PM_{2.5}$ from the 6-model ensemble follow the observed year-to-year PM variations well, with a peak in 2003 and a smaller one in 2006 (the years with heatwave occurrences, which facilitated enhanced photo-chemical formation of sulphate and secondary organic aerosols and inhibited aerosol wet removal). Furthermore, the observations show a trend stagnation for $PM_{10}$ and increase of $PM_{2.5}$ towards the end of the period at the sites considered. This is not reproduced accurately by the models. A look at the individual sites reveals that the observed increase is the result of $PM_{2.5}$ going up from 2008/2009 to 2010 at 7 out of 13 sites. According to assessments of PM pollution in 2009 and 2010, presented in EMEP Status Reports 4/2011 and 4/2012 (www.emep.int), about half of the sites with PM measurements reported an increase in annual mean $PM_{10}$ and $PM_{2.5}$ with respect to the year before. As documented in those reports, a 3-4 % decrease per year of $PM_{10}$ was registered between 2008 and 2010, whereas average $PM_{2.5}$ levels were similar in 2008 and 2009 and increased by 4 % in 2010, averaged over all sites with PM data. However, large variations between monitoring sites were observed. For instance, enhanced annual mean $PM_{10}$, and particularly $PM_{2.5}$ levels, were reported for 2010 at Austrian, German, Swiss, and Finnish sites, which are among the trend-sites included in the present trend analysis. The major reason for elevated annual PM levels is often the occurrence of winter pollution episodes (caused by stagnant conditions within a very low boundary layer and exacerbated by enhanced emissions from domestic heating), which are not always accurately modelled due to either an overestimation of mixing layer height by relatively coarse vertical resolution or/and underestimation in the emission input data.

In general, the EDT model ensemble reproduces the observed annual 2000-2010 series of PM at the trend sites quite well, showing a high correlation of 0.95 for both $PM_{10}$ and $PM_{2.5}$. Overall, the ensemble simulated $PM_{10}$ and $PM_{2.5}$ concentrations are lower than observed values by 31 and 19 % respectively (a greater bias for $PM_{10}$ is partly caused by the POLR model - see below). A fairly good correspondence with respect to PM year-to-year changes is seen in Fig. 4 (b, c) for the individual models compared with observations (with the exception of $PM_{10}$ concentrations from POLR having a low bias because the contribution from coarse sea salt was not accounted for). Some deviations of LOTO's results for 2003 and 2006 are probably due to a different meteorological driver used in the model runs (see Sec. 2.1). The correlation between the modelled and measured series of annual mean $PM_{10}$ and $PM_{2.5}$ is high, with the following correlation coefficients: 0.96 and 0.93 for CHIM, 0.93 and 0.93 for EMEP, 0.77 and 0.85 for LOTO, 0.93 and 0.90 for MATCH, 0.93 and 0.88 for MINNI, and 0.70 and 0.87 for POLR, for $PM_{10}$ and $PM_{2.5}$ respectively. These results give credibility to the results of the models and their ability to accurately simulate the changes in the PM levels due to emission changes, as well as represent the inter-annual variability due to meteorological conditions. These results also show that the model ensemble correlates better with the observations than the individual models when both $PM_{10}$ and $PM_{2.5}$ annual series are considered.

Averaged over all sites (see Table 1), the mean ensemble simulated trends (the standard deviations STD are in parentheses) are -0.24 (STD=0.09) $\mu g\,m^{-3}\,yr^{-1}$ for $PM_{10}$ and -0.21 (0.10) $\mu g\,m^{-3}\,yr^{-1}$ for $PM_{2.5}$. These are smaller compared with the observed -0.35 (STD=0.35) and -0.40 (0.38) $\mu g\,m^{-3}\,yr^{-1}$, respectively, but can be anticipated given models' underestimation of PM concentrations. The correspondence between model results and observations is better in terms of relative 2000-2010 trends (the STD are in parentheses), which are -1.7 (0.40) and -2.0 (0.33) % $yr^{-1}$ from the model ensemble and -2.1 (1.19) and -2.9 (1.48) % $yr^{-1}$ from the observations, for $PM_{10}$ and $PM_{2.5}$ respectively.

## 5.2 PM trends at the individual sites

Figure 5 presents observed and simulated (by the 6-model ensemble) $PM_{10}$ and $PM_{2.5}$ trend slopes for each site for the period 2000-2010. The sites at which significant trends were observed are marked with a star. The modelled significant and insignificant trends are represented respectively by dark and light blue bars.

The observed and ensemble-modelled $PM_{10}$ and $PM_{2.5}$ trends at all sites are decreasing. Figure 5 shows quite a large variability in the trends observed at different sites, ranging between -0.08 and -0.88 $\mu g\,m^{-3}\,yr^{-1}$ for $PM_{10}$ and between -0.05

and -1.5 $\mu g\,m^{-3}\,yr^{-1}$ for $PM_{2.5}$. Compared with the observations, ensemble-modelled trend slopes show less variability across the sites, with the Standard Deviations of 0.09 and 0.10 $\mu g\,m^{-3}\,yr^{-1}$ versus 0.23 and 0.38 $\mu g\,m^{-3}\,yr^{-1}$ in the observations for $PM_{10}$ and $PM_{2.5}$ respectively (Table 1). The modelled trends are mostly within -0.5 $\mu g\,m^{-3}\,yr^{-1}$, and rather poorly correlated with the observations between the trend sites. The strongest negative $PM_{10}$ trends were observed at three of the Spanish sites and one Austrian site (with decreases greater than 0.7 $\mu g\,m^{-3}\,yr^{-1}$), while the weakest (and mostly non-significant)

trends were registered at British, Norwegian and some German sites (below -0.15 $\mu g\,m^{-3}\,yr^{-1}$). The strongest significant $PM_{10}$ decreasing trend slopes were modelled for German and some other sites in Central Europe. For most of the Spanish sites, the model ensemble simulated PM decrease by 0.2-0.3 $\mu g\,m^{-3}\,yr^{-1}$, but the trends were classified as insignificant. In general, we see a similar pattern in the results for $PM_{2.5}$, with the exception that the strongest trend was both observed (-1.5 $\mu g\,m^{-3}\,yr^{-1}$) and modelled (-0.4 $\mu g\,m^{-3}\,yr^{-1}$) for Ispra (IT0004) in the Po Valley. Uncertainties in the emission trends and

spatial distribution could be one of the main reasons for the discrepancies between the model ensemble and observations (see Sec. 7 for more discussion).

The observed relative trends range from -0.5 to -4.5 % $yr^{-1}$ for $PM_{10}$ and from -0.5 to -5.2 % $yr^{-1}$ for $PM_{2.5}$ (Fig. 6). Also in this case, ensemble-simulated relative trends show less variability, with values between -1.0 and -2.5 % $yr^{-1}$. The strongest negative $PM_{10}$ trends (with rates of decrease greater than -3.5 % $yr^{-1}$) were observed at three of the Spanish sites and the

Swedish one, whereas the weakest and mostly non-significant trends (under -1 % $yr^{-1}$) were registered at the British and some German sites. The indicated in the previous paragraph reason for model vs. observation differences applies also for relative trends, but for the latter the estimated PM at the start of the period (see 2.3) affects the results as well.

All in all, the observations show significant $PM_{10}$ trends at 11 out of 26 sites and significant $PM_{2.5}$ trends at only 5 out of 13 sites. A closer look at $PM_{10}$ and $PM_{2.5}$ annual series at the individual sites (not shown) reveals that the sites where

no significant trend was identified in the observations have a particularly large inter-annual variability of PM concentrations. Model ensemble results identify significant trends at more sites compared with the observations, namely at 18 sites for $PM_{10}$ and at 8 for $PM_{2.5}$. As can also be seen on the trend maps (Fig. 2), the model ensemble and the observations do not always agree regarding the significance of trends at specific locations, even within the same country. For example in Spain, strong decreasing significant trends were observed at 4 out of 6 sites for $PM_{10}$ and at 3 out of 4 sites for $PM_{2.5}$, whereas the model ensemble

mostly estimates non-significant trends. This is in contrast to the German sites, for which the models simulate significant and quite appreciable $PM_{10}$ and $PM_{2.5}$ trends for all sites (as a result of emission reductions in the whole country), but significant observed trends are found for only 1 out of 7 sites for $PM_{10}$ and for neither of 2 sites for $PM_{2.5}$. The reason for this seems

to be that the trends were distorted by particular high annual mean PM concentrations in 2003, 2006 and 2010 at most of the German sites (not shown here).

Similar to Fig. 5 for the model ensemble, Fig. A7 presents $PM_{10}$ and $PM_{2.5}$ mean trends calculated by the individual models, with only significant modelled trends shown. For any specific site, the trend slope values from the models are in general agreement (Fig. A7), while there are discrepancies between the models with regards to the significance levels of simulated trends. The largest number of significant $PM_{10}$ and $PM_{2.5}$ trends were simulated by EMEP (23 and 14, respectively) and the smallest number by MINNI (10 and 7) (see also Table 1).

The relative trends from the individual models are compared with each other and with observed relative trends in Fig. A8 for the set of trend sites.

Averaged over all sites (see Table 1), the trends simulated with the individual models range from -0.16 to -0.33 $\mu$g m$^{-3}$ yr$^{-1}$ for $PM_{10}$ and from -0.19 to -0.26 $\mu$g m$^{-3}$ yr$^{-1}$ for $PM_{2.5}$ and are weaker than observed trends (-0.35 and -0.40 $\mu$g m$^{-3}$ yr$^{-1}$ respectively). The agreement among the models appears to be better in terms of relative trends that range from -1.4 % yr$^{-1}$ to

-2.2 % yr$^{-1}$ for $PM_{10}$ and from -1.8 % yr$^{-1}$ to -2.4 % yr$^{-1}$ for $PM_{2.5}$ (site averages). Compared to absolute trends, those correspond better with observed trends (-2.1 and 2.9 % yr$^{-1}$ respectively).

## 5.3   PM seasonal trends

Figure 7 presents the maps of 2000-2010 seasonal mean trends of $PM_{10}$ and $PM_{2.5}$ from the 6-model ensemble and the observations. For the winter season, the model ensemble estimates significant $PM_{10}$ and $PM_{2.5}$ trends only in small areas,

mostly in southern parts of Europe. The observational data do not show any significant trends for $PM_{10}$. For $PM_{2.5}$, the observations indicate quite strong significant trends at only three sites, i.e. in the north-east of Spain (also identified by the model ensemble), north Italy and south of Sweden. One probable reason for the limited number of sites with significant observed trends is negligible reductions and even increases in the emissions of primary PM from residential heating, most important in the winter period, which were not efficiently regulated.

For the summer period, both the model ensemble and observations estimate the strongest negative trends out of all seasons. Significant trends are simulated for most of the domain (except Northern Europe, south of Spain and most eastern parts of the domain). The number of sites with observed significant trends is also the strongest for summer, namely 12 out of 26 for $PM_{10}$ and 10 out of 13 for $PM_{2.5}$. In the spring and autumn periods, both modelled and observed trend slope values and the fraction of sites with significant trends are between those of winter and summer.

It can be noted that Ispra in northern Italy (IT0004) is the only site where significant $PM_{2.5}$ trends were observed and modelled for all seasons, with the exception of the modelled winter trend. For $PM_{10}$, the quite strong significant mean trends at four Spanish sites (ES0007, ES0008, ES0013 and ES0014) appear to be due to strong summer trends, whereas the trends are insignificant in the other seasons. Among the German sites, significant observed $PM_{10}$ trends are only identified at DE0001 and DE0007 and only for the spring period. The models agree with that, but also calculate significant trends for summer and

autumn.

Figures 8 (a, b) present the annual series of the 6-model ensemble and observed seasonal mean trends of $PM_{10}$ and $PM_{2.5}$ for the period 2000-2010, averaged over all trend sites. The values of absolute and relative trend slopes are summarized in Table 2.

Averaged over the trend-sites, the largest decrease in PM during the 2000-2010 period took place in the summer months for both $PM_{10}$, with the mean seasonal trend of -0.32 $\mu$g m$^{-3}$ yr$^{-1}$ from the model ensemble and -0.56 $\mu$g m$^{-3}$ yr$^{-1}$ from the observations, and for $PM_{2.5}$ (-0.26 and -0.51 $\mu$g m$^{-3}$ yr$^{-1}$, respectively). The weakest trends were found for the winter season from the models and observations for $PM_{10}$(-0.13 and -0.19 $\mu$g m$^{-3}$ yr$^{-1}$, respectively) and also for modelled $PM_{2.5}$ (-0.10 $\mu$g m$^{-3}$ yr$^{-1}$), whereas the observed $PM_{2.5}$ trend has a minimum of -0.27 $\mu$g m$^{-3}$ yr$^{-1}$ in the autumn season. The weakest winter trends are partly due to the larger amplitudes of the inter-annual changes in mean PM levels. In particular, the elevated winter levels of $PM_{10}$ and $PM_{2.5}$ in 2006, and especially in 2010, contribute to reduce the mean seasonal trend.

Figure 9 presents the seasonal mean trends simulated by the individual models and the model ensemble, along with the observed trends. The graphs nicely visualize the seasonal variations of PM trend slopes discussed above. They also show quite a good correspondence between the trend seasonality from the individual models. Relative trends of PM show quite similar seasonal patterns, with the strongest trends in the summer and weaker ones in the cold seasons of 2000-2010 (Fig. 9). For $PM_{10}$, observed relative trends are -2.9 % yr$^{-1}$ in the winter period and -3.7 % yr$^{-1}$ in the summer period; the respective numbers from the model ensemble are -2.3 and -2.5 % yr$^{-1}$. For $PM_{2.5}$, the observed and modelled summer trends are -3.7 and -2.9 % yr$^{-1}$, whereas the weakest observed mean trend of -2.0 % yr$^{-1}$ was in the autumn and the weakest modelled trend of -1.4 % yr$^{-1}$ was estimated for the winter period. The individual models largely agree on the seasonal profiles of the relative trends, although some variability exists between the simulated trend slopes (similar to those for seasonal absolute trends).

## 5.4 Contribution of individual components to PM trends

$PM_{10}$ and $PM_{2.5}$ is a complex mixture of different aerosol components originating from a variety of anthropogenic and natural emission sources and so PM trends are basically the sum of individual trends of its constituents. Thus, for a better understanding of the effects of emission reductions of different pollutants, it is imperative to look at the role of the individual aerosol components in the changes of PM concentrations.

A comprehensive study of the trends for individual aerosols is beyond the scope of this paper. Besides, there is practically no available observational data for individual PM components collocated with PM measurements during the period 2000-2010. In fact, Birkenes in the south of Norway is the only site for which observational data for both $PM_{10}$ and $PM_{2.5}$, and for secondary inorganic aerosols (SIA) meet the required criteria for the trend study. Still, we think that for a better interpretation of PM trends discussed in this paper, it is relevant to have a brief insight into the trends of PM components. Here, we summarize the main results of modelled and observed trends of some PM components for 2000-2010. For more a detailed analysis of inorganic gases and aerosols, the reader is referred to Ciarelli et al. (2019).

Figure A9 shows the maps of model ensemble simulated and observed annual mean 2000-2010 trends for $SO_4^{-2}$, $NO_3^-$ and $NH_4^+$ aerosols. Note that due to the lack of consistent observational data sets (as pointed out above), the set of sites for SIA are

not the same between the species and also different from those used in PM trend analysis. The number of sites used here is 39, 14 and 13 for $SO_4^{-2}$, $NO_3^-$ and $NH_4^+$.

The absolute trends are all decreasing, though the rates are not directly comparable (since they are expressed as $\mu g\ m^{-3}$ (S) $yr^{-1}$ and $\mu g\ m^{-3}$ (N) $yr^{-1}$). The maps of relative trend slopes show the strongest trends all over Europe for $SO_4^{-2}$ (between -2 and -4 % $yr^{-1}$ over most of the domain, exceeding -5 % $yr^{-1}$ in Spain), closely followed by $NH_4^+$. For $NO_3^-$, the models only estimated significant downward trends in Central European countries and Italy. The modelled trends for SIA are decreasing over the entire domain, whereas the observations indicate significant increasing trends of $SO_4^{-2}$ and $NO_3^-$ at the Polish site Sniezka (close to the Czech border). In addition, rather strong, though non-significant, positive trends of $NO_3^-$ and $NH_4^+$ were observed at two Dutch sites, and somewhat weaker positive trends at a few other sites. No observational datasets long enough (or obtained with consistent analytical methods) for trend studies of carbonaceous aerosols were available at EMEP sites. Shorter series for total carbon, available for three-four sites, show a 4-5 % decreasing trend between 2003/2004 and 2010.

In summary, the results presented here, and the analysis by Ciarelli et al. (2019), indicate that the models estimate a somewhat larger than observed decrease of $SO_4^{-2}$ in Central (also missing some positive trends) and Northern Europe and a smaller decrease in Spain. The models appear to overestimate the observed negative trends for $NO_3^-$ and also for $NH_4^+$, though to a smaller degree (one should keep in mind that for $NO_3^-$ and $NH_4^+$ there is limited number of measurement sites covering a limited geographic area). It should be noted that none of the models accounts base cations (i.e. $Na^+$, $K^+$, $Ca^{2+}$ and $Mg^{2+}$) in gas-aerosol partitioning of $HNO_3$ (see Table in A16). Those base cations are significant components of sea salt and mineral dust. They participate in aerosol chemistry and facilitate the formation of coarse $NO_3^-$, consuming $HNO_3$ and thus making less of it available for $NH_4NO_3$ formation. As the emissions of sea salt and mineral dust strongly depend on meteorology (especially on surface wind speed), $NO_3^-$ formed on the base cations (and consequently total $NO_3^-$) is subject to inter-annual variability, which could weaken $NO_3^-$ trends and lead to a larger fraction of insignificant trends. Thus, not including base cations in aerosol chemistry could be one reason for models' overestimating of the observed $NO_3^-$ trends (see also discussion in 7. Among the EDT models, MINNI and POLR did not included coarse $NO_3^-$, CHIM and LOTO included $NO_3^-$ formation on sea salt $Na^+$, while EMEP and MATCH used constant reaction rates for coarse $NO_3^-$ formation from $HNO_3$, irrespective of base cation availability (Table A16). However, we could not see any consistent differences in the relative trends of $NO_3^-$ and $NH_4^+$ between the models with and without coarse $NO_3^-$ (not shown here), neither the comparison of $NO_3^-$ trends from the individual models with observations at the rather limited number of sites gave conclusive results.

The relative contributions of $SO_4^{-2}$, $NH_4^+$, $NO_3^-$, total primary particulate matter (TPPM$_{10}$) and anthropogenic SOA (ASOA) to $PM_{10}$ trends in the period 2000-2010 estimated by the model ensemble are presented in Fig. 10. The maps reveal considerable variability of the role of the individual aerosol species $PM_{10}$ trends across European countries. The decrease in $SO_4^{-2}$ concentrations (Fig. 10a) played the dominating role over most of the EDT domain, except from parts of Central Europe and Northern Italy. Namely, relatively large contributions of $NO_3^-$ to $PM_{10}$ trends are seen in Germany (and neighbouring parts of France, Czechia and Poland), Denmark, the Netherlands, and in the Po Valley (Fig. 10c). The reduction of $NH_4^+$ levels, which includes both ammonium sulphate and ammonium nitrate, appears to be quite an important contributor to the $PM_{10}$ decreasing trends, with the largest effects estimated for Poland, Denmark, and the Po Valley (Fig. 10b). The reduction of pri-

mary PM emissions was according to the model ensemble simulations the dominating factor for $PM_{10}$ trends in Portugal and southern parts of Balkan; as well as in many European cities (due to emission reductions from traffic and residential heating) (Fig. 10d). Finally, ASOA is also estimated to have quite a notable contribution of 3-7 % to $PM_{10}$ downward trends (though ASOA modelling is still associated with rather large uncertainties). The model results imply that the chemical composition of European $PM_{10}$ has changed somewhat during the 2000-2010 period, with $NO_3^-$ (and probably ASOA) becoming an increasingly important constituent compared with the other anthropogenic aerosols, i.e. $SO_4^{-2}$, $NH_4^+$ and primary emitted PM (elemental and primary organic carbon, dust and metals).

The relative contributions of $SO_4^{-2}$, $NH_4^+$, $NO_3^-$, and ASOA to $PM_{10}$ trends in the period 2000-2010, as calculated by the individual models can be found in Appendix (Fig. A11). Most of the models (but for POLR), agree that over most of the EDT domain, except from some central European countries, decreases in $SO_4^{-2}$ concentrations were the main cause of $PM_{10}$ downward trends, with somewhat smaller contribution from decreasing $NO_3^-$ levels. This is consistent with the emission trends shown in Fig. A1. The largest emission reductions were achieved for $SO_x$, which explains the relatively strong trends in $SO_4^{-2}$ (and also appreciable trends in $NH_4^+$ in the form of ammonium sulphate) concentrations. The reductions of $NO_x$ and $NH_3$ emissions from 2000 to 2010 were smaller compared with $SO_4^{-2}$. Thus, as the formation of ammonium sulphate was decreasing in the 2000s, more and more $NH_3$ was becoming available for the formation of ammonium nitrate $NH_4NO_3$. Notably in Germany, as well as in the Benelux countries and the Po Valley, $NO_3^-$ is estimated by the models to have the largest contribution to the $PM_{10}$ trends. However it should be kept in mind that in the regions influenced by mineral dust and/or sea salt, some of nitric acid would be consumed in the formation of $NO_3^-$ associated with base cations (as discussed above, this is not fully accounted for in the EDT models), so that less $NH_4NO_3$ would be formed compared to what the EDT models simulate.

Furthermore, the estimates by LOTO point to primary anthropogenic $PM_{10}$ as the main component driving $PM_{10}$ levels down in a large part of the simulation domain. CHIM, MINNI and to some extent EMEP agree with the LOTO estimates for Northern Europe and the area covering Benelux, northern parts of Germany and France, and the south of the UK. In contrast to the other models, POLR estimated that $NO_3^-$ contributed the most to the $PM_{10}$ trends, whereas the contribution of $SO_4^{-2}$ and $NH_4^+$ were rather moderate in Central Europe, the UK, the Baltic countries. The modelled contributions of ASOA to $PM_{10}$ trends is below 5 % according to CHIM, MATCH and MINNI, whereas EMEP simulates contributions of 5-10 % and POLR 5-30 % . This variability can be explained by the different ways of handling SOA chemistry in the models. Furthermore, somewhat weaker PM trends from LOTO could probably be explained by not including SOA chemistry in these simulations. Similar results are seen with respect to the relative contributions of the individual aerosols to modelled $PM_{2.5}$ trends between 2000 and 2010 (Fig. A12).

As far as natural aerosols are concerned, emissions are largely driven by meteorological conditions (e.g. by the surface wind in the case of sea salt and windblown dust, while the air temperature controls emissions of biogenic VOCs - precursors of biogenic secondary organic aerosol, BSOA). In addition, the generation of mineral dust is dependent on the availability of erodible (snow and vegetation free) soil and its moisture (which in turn depends on precipitation frequency and amount), whereas the temperature and salinity of sea water affect sea spray formation, though those conditions are less variable. Of

course, similar to anthropogenic aerosol, the transport and removal of the natural particles are determined by atmospheric dynamics and precipitation. In short, year-to-year changes in the concentrations of natural aerosols are driven primarily by inter-annual meteorological variability. Among natural aerosols, only formation of BSOA has some dependency on anthropogenic emissions, as BSOA can be formed from biogenic VOCs condensing on primary organic aerosols from anthropogenic sources. Thus, BSOA production is somewhat affected by the trend in PM emissions. In addition, as discussed above, the changes in $NO_3^-$ formed from anthropogenic $NO_x$ emissions are in fact dependent on the variability of natural aerosols of sea salt and mineral dust.

Not all natural particles were calculated in a consistent way by all of the models. The missing components are: BVOC from LOTO and MATCH, sea salt from POLR; and only EMEP and LOTO simulated trends of windblown dust in the modelling domain, whereas the other models only included mineral dust from boundary conditions. Figures 10 (f, g, h) present the computed contributions of natural aerosols estimated by the models, i.e. biogenic SOA, sea salt and mineral dust, to $PM_{10}$ trends, where the negative contributions (blue colours) mean increasing trends in the natural aerosols.

The model ensemble simulated decreasing BSOA trends that contribute 1-3 % of $PM_{10}$ decreasing trends over almost all land area (Fig. 10f), with the largest contribution (5-10%) in Fennoscandia and north-western Russia. The contribution of sea salt trends (derived as 3.26*sea salt Na, assuming 30.7 % sodium content in sea salt aerosols, the same as in sea water) to $PM_{10}$ trends is, on average, 2-5 % over land and exceeds 10 % in areas influenced more by the sea and less polluted regions (Fig. 10g). Comparison of the modelled sea salt trend with rather sparse observations can be found in Fig. A10 (a).

Furthermore, from the EMEP and LOTO results, we see contributions of 1-3 % from mineral dust to decreasing $PM_{10}$ trends over most of Europe (in excess of 10% in Spain, Italy), but also some negative contributions due to increasing dust trends in Greece, Portugal and south-eastern Europe and Russia (Fig. 10h). All in all, the inter-annual variability and increasing modelled trends for natural aerosols for some regions do not appear to have reversed the decreasing $PM_{10}$ trends in the 2000-2010 period (with some exceptions for windblown dust).

Model analysis of the seasonal trend of the individual $PM_{10}$ and $PM_{2.5}$ components shows the strongest trends of SIA ($SO_4^{-2}$, $NH_4^+$ and $NO_3^-$) in summer and also in spring for $NO_3^-$, while the weakest trends of all SIA are calculated for winter. On the contrary, the strongest trends for primary PM are simulated for winter and the weakest for summer.

## 6   PM trends in the period 1990 – 2010

As no regular measurements of PM were conducted prior to 2000, this paper mainly focus on the period 2000-2010. As far as the years prior to 2000 are concerned, we have to rely solely on model simulations to assess the effect of emission reductions on European levels of particulate pollution in the 1990s. Given that, any deep analysis of that decade is beyond the scope of the paper, but still we think it is relevant to present a multi-model assessment of PM trends during the whole 1990-2010 period, studied within the EDT framework. It should be kept in mind while looking at those results, that the emission data, in particular for PM, are much less reliable before 2000.

Figure A13 shows annual mean trends for the period 1990-2010 for $PM_{10}$ and $PM_{2.5}$, absolute and relative to 1990, produced by the ensemble of five models (all the above except POLR). Over the whole European domain, the models simulate significant decreasing PM trends. The strongest trends (0.75-1.0 $\mu g\,m^{-3}\,yr^{-1}$, or 2.5-3 % $yr^{-1}$) were simulated for Central Europe (extending eastward over Ukraine and European Russia for $PM_{2.5}$). The weakest trends of less than 0.3 $\mu g\,m^{-3}\,yr^{-1}$

(1.5-2 % $yr^{-1}$) are seen in Northern Europe and Russia and in Southern Europe. The rest of the domain experienced intermediate trends of 0.3-0.75 $\mu g\,m^{-3}\,yr^{-1}$ (1.5-2.5 % $yr^{-1}$ relative to the year 1990). Notably, the weakest decreasing trends (below 1.5 % $yr^{-1}$) are modelled for $PM_{10}$ in the southernmost parts of Mediterranean countries, which are heavily influenced by Saharan dust and so PM trends due to the reductions of anthopogenic emissions are distorted. The mean annual trends during the period of 1990-2010 are stronger compared with those for the 2000-2010 period (Fig. 2). This is a consequence of larger

emission reductions in the 1990s compared with the 2000s. Thus, the EDT model ensemble simulated that annual mean $PM_{10}$ and $PM_{2.5}$ concentrations decreased by between 5 and 15 $\mu g\,m^{-3}$ across most of Europe (by 2-5 $\mu g\,m^{-3}$ in the Northern Europe) from 1990 to 2010.

## 6.1  PM trends in European countries in 1990-2000-2010 periods

The graphs in Fig. A14 provide more details regarding $PM_{10}$ trends in individual European countries and compare the trends

in the 1990s and 2000s.

Figure A14a shows the trends of $PM_{10}$ between 1990 and 2010 simulated by the five models for the individual countries and sea areas. The strongest annual mean trends, with decreases greater than -0.6 $\mu g\,m^{-3}\,yr^{-1}$, (leftmost countries in the graph) were simulated for Central European (Germany, Hungary, Czech Republic) and the Benelux countries, which were the regions with among the highest PM levels. The weakest downward trends are modelled for relatively cleaner North European (Iceland,

Norway, Finland, Sweden) and Baltic countries, but also in Mediterranean countries influenced by shipping emissions and African dust intrusions (rightmost countries in the graph). The models are in general agreement regarding the ranking of $PM_{10}$ national trends, and the spread between PM national trends calculated with the individual models is rather moderate (the mean STD between the models is 0.054 $\mu g\,m^{-3}\,yr^{-1}$, varying between 0.005 and 0.104 $\mu g\,m^{-3}\,yr^{-1}$ for different countries). The variation of $PM_{2.5}$ trends across Europe is quite similar (therefore not shown here), with the only difference that the trends in

the Benelux countries were the strongest.

Figure A14b shows for the individual countries and regions, the $PM_{10}$ annual trends calculated by the model ensemble for the 1900-2000 and the 2000-2010 periods separately. For most of the countries, the largest reductions of $PM_{10}$ levels took place in the 1990s compared with the 2000s, which is consistent with considerably larger emission reductions of PM emissions and their gaseous precursors (except from ammonia) during the first of those decades. This is especially pronounced

in Central Europe, where the 1990-2000 trends were around 1 $\mu g\,m^{-3}\,yr^{-1}$ compared with around 0.3 $\mu g\,m^{-3}\,yr^{-1}$ in the 2000-2010 period. The exceptions are North-European countries, and also relatively small emitters of pollution, such as Malta, Liechtenstein, Cyprus, where $PM_{10}$ trends were similar during both decades.

The $PM_{10}$ relative trends (i.e. with respect to the starting years of 1990 and 2000) in the 1990-2000 period are also considerably stronger than those in the 2000-2010 period (not shown, or in Supplement). The model results indicate a large variability

in 1990-2000 trends between the countries (from -1.1 % $yr^{-1}$ in Central Europe to -(0.0-0.2) % $yr^{-1}$ in Northern Europe, Cyprus, Malta), whereas the 2000-2010 trends are more homogeneous across the countries, ranging between 0 and -3 % $yr^{-1}$.

## 7  Discussion

### 7.1  Discussion of main results

The ensemble of six EDT models simulated that, from 2000 to 2010, the annual mean $PM_{10}$ and $PM_{2.5}$ concentrations de-
creased by between 10 and 20 % over most of Europe, and respectively by up to 25 % and 30 % in Germany, the Netherlands, Belgium, parts of the UK, Portugal, north/centre of Italy and large parts of Scandinavia. Notably, despite lower $PM_{2.5}$ concentrations, the $PM_{2.5}$ absolute downward trends appear only slightly smaller than those for $PM_{10}$, indicating a trend-masking role of coarse PM of natural origin. On average, we found a fair agreement between modelled and observed concentration reductions at 26 (for $PM_{10}$) and 13 (for $PM_{2.5}$) measurement sites. In the course of those 11 years, $PM_{10}$ and $PM_{2.5}$ con-
centrations at the studied sites decreased respectively by 17 and 20 % according to the model ensemble and by 21 and 29 % as derived from observational data. Moreover, we found a larger spatial variability of PM trends registered by observations compared with those estimated by the model, with observed decreasing trends ranging between approximately 5 % (at British site GB0036) and 50 % (at Swedish site SE0012). We also see some discrepancies in the geography of trends from the observations and EDT model, with the largest observed decreases (above 30 %) at the sites in Sweden, Finland and Spain (also
the Po Valley for $PM_{2.5}$), whereas the models simulate the strongest trends for German sites (mostly above 20 %) and do not identify significant trends for Spanish sites (though 10–20 % decreases in $PM_{10}$ and $PM_{2.5}$ is simulated).

Modelled PM concentrations are to a large degree determined by the emission data used and modelled PM trends reflect the trends in national emissions. For instance, relatively strong simulated PM trends in Germany, the Benelux, the UK and Portugal are due to considerable reductions of all gaseous precursors and primary PM in those countries (Fig. A2). Poland is
among the countries with the greatest reduction of SOx and considerable reductions in NOx emissions from 2000 to 2010, but the increase in NH3 emissions contributed to additional SIA formation during those years. Besides, the emissions of primary $PM_{2.5}$ in Poland increased during the same period. Thus, the resulting modelled downward trends are relatively weaker (and insignificant in parts of the country). In Northern Europe, the appreciable decrease of PM concentrations is not only due to reductions in NOx and primary $PM_{2.5}$ emissions in those countries, but is also due to decreased long-range transport from
Central Europe and the UK (somewhat lessened by the increased NOx emissions from international shipping in the North and Baltic seas). For Spain, the model ensemble simulated a substantial decrease in PM concentrations (though the PM trends were characterised as insignificant), mostly resulting from emission reductions of gaseous precursors, while the reductions in emissions of primary PM (especially coarse PM) were relatively smaller. Only the EMEP model (and MATCH for $PM_{2.5}$) simulated significant PM trends for most of Spain, whereas PM trends from the other models were found to be insignificant due
to smaller PM decreases from 2000 to 2010 or/and larger inter-annual variability (as in the results from LOTO and MATCH, using different meteorology).

Furthermore, the analysis showed a considerable variability in the observed trends within the same country, which the models could not fully reproduce. This can be due to local emissions, unaccounted for, or misrepresented spatially and temporally in the model input. In some countries, the differences in trends could also be related to a complex topography leading to localised pollution transport dynamics (e.g. Switzerland and Austria), unresolved by meteorological drivers.

As PM is a complex pollutant, consisting of different aerosol species, the concentrations and trends of PM are the result of an intricate interplay of the effects of their direct emissions and gaseous precursors from a variety of anthropogenic and natural sources. As discussed in Sec. 5.4, the emissions of $SO_x$ went down by 37 % from 2000 to 2010, resulting in the decrease of ammonium sulphate concentrations and thus more ammonia available for reactions with nitric acid. The reduction of $NO_x$ emissions in the same period (17 %) was smaller than that of $SO_2$. Given rather moderate reductions of $NH_3$ emissions (only 6% on average), the concentrations of ammonium nitrate decreased less compared with ammonium sulphate. The model ensemble calculated the decrease for $SO_4^{-2}$ to be in a range of 25-45 % (45-55 % in Spain and Portugal) and for $NH_4^+$ in a range of 15-40 % over Europe from 2000 to 2010 (Fig. A9, a-f). The modelled decrease of $NO_3^-$ concentrations is mostly under 30 % and the trends are insignificant in most countries. For more detailed discussion on SIA trends, we refer the reader to the analysis published in (Ciarelli et al., 2019). In that publication, relatively moderate trends in $SO_4^{-2}$ compared with the emission reductions of $SO_2$ was explained by an increase in the availability of oxidant species and more efficient pH-dependent cloud chemistry resulting from those emission reductions. (Ciarelli et al., 2019) also discusses a shift in the thermodynamic equilibrium between $HNO_3+NH_3$ vs. $NH_4NO_3$, favouring aerosol formation. Furthermore, the reduction of anthropogenic VOC emissions, including aromatic hydrocarbons - precursors of SOA, by 33 %, on average, led to a decrease in ASOA concentrations by 15-30 % from 2000-2010 (Fig. A9, g, h). Finally, the emissions of both $PM_{2.5}$ and coarse PM reduced, on average, over the modelled domain by 10 %, thus making primary PM an important driver of $PM_{10}$ and $PM_{2.5}$ decreases in some European regions (not shown here).

Due to the lack of long-term observational data of $PM_{10}$ and $PM_{2.5}$ supplemented with chemical analyses, the model results regarding the role of the individual components in $PM_{10}$ and $PM_{2.5}$ trends during 2000-2010 cannot be thoroughly validated. We can only make a crude estimate, using observations of SIA and OC, which are not necessarily collocated, available at a limited number of sites. The observed average trends were the strongest for organic aerosols (-3.8 % $yr^{-1}$ at 4 sites), followed by $NH_4^+$ (-2.9 % $yr^{-1}$ at 13 sites), $SO_4^{-2}$ (-2.6 % $yr^{-1}$ at 39 sites), and finally the weakest trends were for $NO_3^-$ (-0.5 % $yr^{-1}$ at 14 sites).

## 7.2  Uncertainties in emissions

As shown in the previous section, the modelled trends in PM and its components quite closely reflect emission reductions, though inter-annual variability of meteorological conditions also plays an important role in PM pollution levels (see 7.3). This means that good quality emission data is essential for accurate model simulations of the trends.

Emission estimates are associated with uncertainties due to missing or incomplete information, or limited understanding with respect to activity data, emission factors, source locations etc. (Klimont et al., 2017).

No publication with a detailed and quantitative uncertainty estimate of the GAINS dataset used here (ECLIPSE_V5) is available, but (Amann et al., 2011) and (Schöpp et al., 2005) described the treatment of uncertainties in the context of the GAINS model. For example, for 1990, (Schöpp et al., 2005) estimated that the national total emissions used in the RAINS integrated assessment model had an uncertainty of $\pm(6$–$23)$ % for $SO_2$, $\pm(8$–$26)$ % for $NO_x$ and $\pm(9$–$23)$ % for $NH_3$ (95% confidence interval). However since that assessment, steps have been taken to reduce the uncertainty in the emission data sets (Klimont et al., 2017). The European Environment Agency indicated somewhat larger uncertainties in typically top-down emission estimates in the EU LRTAP inventory, namely around $\pm10$ % for $SO_2$, $\pm20$ % for $NO_x$ and $\pm30$ % for $NH_3$ and NMVOCs (EEA, 2008). Primary PM2.5 and PM10 emission data is said to be of relatively higher uncertainty compared to emission estimates for the secondary PM precursors. Clearly, uncertainties in emissions will inevitably be reflected in the uncertainties in absolute trends of PM.

Furthermore, EEA (2008) suggested that the emission trends are likely to be more accurate than the individual absolute annual values, although the use of gap-filling when countries have not reported emissions for one of more years can potentially lead to artificial trends. Regarding primary PM emissions, ECLIPSE_V5 was the first assessment of $PM_{10}$ and $PM_{2.5}$ emissions, performed using a consistent bottom-up approach across all sources and regions and, therefore, only limited comparison to other works was possible (Klimont et al., 2017).

One of the biggest sources of emissions-related uncertainty is likely to be residential wood-burning emissions of PM and VOCs (forming ASOA) (Simpson et al., 2020). Emissions of primary organic matter (POM) from residential wood burning have been known to be problematic for many years (Simpson et al., 2020; Denier van der Gon et al., 2015; Simpson and Denier van der Gon, 2015), with different countries accounting for, or omitting, semi-volatile compounds in different and often unknown ways. Given that wood burning for heating houses accounts for a significant percentage of European PM emissions, the lack of consistent treatment between countries has obvious implications for the reliability of any trend estimates. There is an increasing recognition that emissions of some potentially important SOA precursors, namely semi-volatile and intermediate-volatility organic compounds (SVOC, IVOC) from traffic sources, are also missing from national inventories and these can have significant impacts on ambient organic matter (OM) (Ots et al., 2016). Emissions of SVOCs and IVOCs are very dependent on e.g. the fuel and type of catalyst used in cars (Jathar et al., 2014; Platt et al., 2017), with older vehicles likely emitting substantially more than new ones, again complicating any analysis of trends. Even for the same country, condensable organics might be included or excluded differently for different sectors. Inclusion or exclusion, or the extent of inclusion of condensables, has also changed over the years, which directly affects the accuracy of trend analyses (Aas et al., 2021). It is also worth noting that the models did not account for the dependence of residential heating emissions on the outdoor temperature, i.e. they increase as it gets colder. This may lead to model underestimation of winter pollution episodes, resulting in under-predictions of annual mean PM (as for 2010, see 5.1). Finally, with respect to anthropogenic sources, assumed invariant spatial distribution of emissions (except from industrial sectors) may cause inaccuracy in modelled trends in some areas.

As far as natural emissions are concerned, biogenic VOC (BVOC) emissions estimates have also many uncertainties, both for isoprene and monoterpenes (e.g. Simpson et al., 1999; Langner et al., 2012; Messina et al., 2016). The models in this study calculate BSOA formed from the oxidation of isoprene and terpenes (CHIMERE also includes sesquiterpenes), but additionally

BSOA can also be formed from the oxidation of stress-induced emissions of other VOCs that are not included in the emissions; this process is likely to be quite frequent, but can only be accounted for in speculative terms with current knowledge (Bergström et al., 2014). Beside uncertainties in emission estimates, the emission data used in the model runs omit some sources of PM. Among the omitted sources of OM are primary biological material, which can contribute e.g. 20-30% of $PM_{10}$ in Nordic areas in summer-early autumn (Yttri et al., 2011) (though it is likely to be much less as an annual average (Winiwarter et al., 2009)).

Marine sources of OM also contribute to observed ambient OM (e.g. Spracklen et al., 2008), but the models used here have not accounted for those (some models, such as EMEP, have assumed background levels of OM which account for such diverse sources, but only in a crude way and with the same levels assumed for all years).

As described in 2.1, pollution from forest fires were not accounted for in EDT simulations mainly because of considerable uncertainties in forest fire emissions and modelling of those, but also because we aimed to look at PM trends due to emission

regulation in Europe. An in-depth analysis of the effect of forest fires on PM trends is beyond the scope of the paper, but we have tested whether the discrepancies between the modelled and observed trends, in particular in terms of a relatively larger fraction of significant trends from the model results, could be due to not including forest fire emissions in the EDT simulations. Additional simulations suggest that the effects from even large fires during the studied period (like 2010 Russian forest fires) were mostly negligible outside the regions where wild fires occurred. In fact, the pollution from major forest fires

did not seem to have any large impact on simulated annual mean PM at the EDT sites in the 2000-2010 period. Therefore we are certain that not accounting for forest fires in EDT analysis did not have any significant consequences for models vs observations comparison. The same applies to not including volcano emissions in the trend simulations. For example, EMEP source-receptor calculations indicate a rather limited contribution to PM2.5 in European countries from volcano emissions (see for example the contributions from Italian Etna, Stromboli and Vulkano and also Eyjafjallajökull eruption in 2010 in EMEP

(2012)).

## 7.3    Effect of inter-annual variability

As pointed out in Sec. 2.3, the probability of trend detection using the Mann-Kendall method decreases for shorter data-series, large natural variability and relatively weak trends. The bottom-line is that the weaker the trend is relative to the inter-annual meteorological variability, the longer the time series that is needed in order to identify a significant trend. The estimates in

675    /https://wiki.met.no/_media/emep/emep-experts/mannkendall_note.pdf indicate that for an 11-year series, the chances for MK methodology to detect significant trends are very small for trends of -1 % $yr^{-1}$, with only 36 % of significant trends identified for inter-annual variability of just 5 % (going down to 9% for inter-annual variability of 15 %). The probability for stronger trends to be identified as significant increases, but still will be between 37 and 71 % for a 10 % variability and down to between 19 and 39 % for a 15 % variability, for -2 to -3 % $yr^{-1}$ respectively.

Most of aerosol processes (some emissions, gaseous and especially heterogeneous chemistry, transport and removal) depend on the meteorological conditions. The model simulations performed in this work indicate that during 2000-2010, the inter-annual variability of PM concentrations due to meteorological variability is mostly between 5 and 10% over most of Europe, 10-12 % in parts of Scandinavia and the UK, and goes up to 15-17 % in the Iberian Peninsula (not shown here). That means

that in the part of Europe, where the modelled trends are relatively strong ($-(1.5\text{-}2.5)$ % $\mathrm{yr}^{-1}$), the MK analyses has identified more significant trends (e.g. in Central and South/South-Eastern Europe). In the Iberian Peninsula, significant modelled trends are only seen in Portugal, where the PM trends are quite strong $-(2\text{-}3)$ % $\mathrm{yr}^{-1}$, but not in Spain with $-(1\text{-}2)$ % $\mathrm{yr}^{-1}$ trends. Also in southern parts of Scandinavia with PM inter-annual variability of 10-12 %, PM modelled trends of $-(2\text{-}2.5)$ % $\mathrm{yr}^{-1}$ are found significant in most of modelling grid-cells. As already mentioned, compared to ensemble modelling, MK analysis could not see significant trends in PM observations at a larger number of the trend sites. This is due to a relatively large inter-annual variability with respect to trend magnitudes in PM observed concentrations (e.g. at German, Austria and Swiss sites, as discussed 5.2).

In addition, we have looked at the relative effects of emission changes and inter-annual meteorological variability on PM trends by calculating the so-called normalised relative trends (NRT) introduced in Solberg et al. (2009) and also applied in Colette et al. (2011). For this purpose, we used additional model results obtained from model runs with fixed 2010 emissions for the meteorological conditions 1990 to 2010 (i.e. Tier 3B as described in Colette et al. (2017a)). The effect of the emissions on PM trends was assumed to be represented by the difference in PM concentrations obtained for corresponding years in the trend runs (Tier3A) and the runs with constant emissions (Tier3B); and the inter-annual variability due to meteorological conditions was quantified by standard deviation of annual PM concentrations in the runs with constant emissions. That is to say, we calculated the ratio of the difference of Sen's slopes ($\mathrm{PM_{Tier3A}}$ - $\mathrm{PM_{Tier3B}}$) to $\mathrm{STD(PM_{Tier3B})}$. The model ensemble NRT for $\mathrm{PM_{10}}$ and $\mathrm{PM_{2.5}}$ are presented in Fig. 11, where absolute NRT values greater than one indicate a larger importance of emission changes with respect to the inter-annual meteorological variability.

Figure 11 shows that the apparent significance of emission reduction on decreasing PM trends appears to be partially masked by inter-annual meteorological variability in large part of Europe in the 2000-2010 period. It should be noted that the individual EDT models have different sensitivity to meteorological variability (besides MATCH and LOTO used different meteorological drivers), which may mask the effects of emission changes. The emission reductions play a larger role in $\mathrm{PM_{2.5}}$ trends, as $\mathrm{PM_{10}}$ concentrations (particularly the coarse fraction of natural origin) are more affected by variability in meteorological conditions. Evidently, the most pronounced effects of emission reductions are associated with the regions with greater emission reductions, e.g. Portugal, Benelux, some parts of South-Eastern European and the Balkan countries. These results are consistent with the main conclusions from the study of PM trends in the period 1998-2007 by Colette et al. (2011). Colette et al. (2017b) arrived to somewhat different conclusions based on a different approach, namely the decomposition of the differences in EDT modelled PM concentrations in 2000 and 2010 to discriminate the role of emissions, meteorology and boundary conditions. Their analysis suggested a relatively larger on average role of emissions compared with the meteorology, though the estimated uncertainties were non-negligible. Due to different premises used by Colette et al. (2017b) and this paper, discrepancies in the outcomes are to be anticipated. That is, here we compared 11-year PM trends with year-to-year PM variability due to meteorological conditions, whereas Colette et al. (2017b) looked at the difference between 2010 and 2000.

To summarise, given rather moderate reductions (and even some increases) in the emissions of some PM precursors and primary PM between 2000 and 2010, we estimate that the effect of emission decreases on 2000-2010 PM trends is roughly of the same order of magnitude as the effect of inter-annual meteorological variability. Separating the effects of emission changes

and meteorological variability on PM trends, we get additional insights regarding their relative roles. PM trend slopes due to emission trends (Fig. 11) appear to be quite similar to the total trends wherever the latter are more significant (Fig. 2). The remarkable difference between them is that the trends due to emissions are significant for nearly the entire domain. Model simulated PM trends due to solely inter-annual meteorological variability (not shown) are by far and large very small ($\pm$ 0.05 $\mu$g m$^{-3}$ yr$^{-1}$) and non-significant everywhere. Thus, our results suggest that the main impact of variable meteorological conditions is to reduce the significance level of PM trends due to emission reductions, while the effects on PM trend slopes are much smaller. For comparison, since the emission reduction during the 1990s were overall larger than in the 2000s, the effect of emission reductions on the decreasing PM trends is estimated to dominate meteorological variability in most of Central, Eastern and South-Eastern Europe (Fig. 11).

## 8 Summary

The Eurodelta-Trends multi-model experiment, aimed to assess the efficiency of emission mitigation measures in improving air quality in Europe, was designed to answer a series of questions regarding European pollution trends in the period of 1990-2010. Among these questions are: Were there significant trends detected by observations? Do the models manage to reproduce observed trends? How close is the agreement between the models and how large are the deviations from observations? In this paper, we address these issues with respect to PM pollution.

An in-depth trend analysis has been performed for $PM_{10}$ and $PM_{2.5}$ for the period of 2000-2010 (limited by the availability of observations), based on results from six CTMs and observational data from the EMEP monitoring network. Given harmonization of set up and main input data (with a few exceptions), the differences in model results should mainly result from differences in the process formulations within the models themselves, and the spread in the models simulated trends could be regarded as an indicator for modelling uncertainty.

The results of the analysis strongly indicate overall decreasing trends of annual mean $PM_{10}$ and $PM_{2.5}$ concentrations between 2000 and 2010, although the trends are not characterized as significant everywhere. The model ensemble simulated mean negative trends that vary from below 0.1 $\mu$g m$^{-3}$ yr$^{-1}$ in northern Europe to 0.1-0.4 $\mu$g m$^{-3}$ yr$^{-1}$ in the eastern parts, and to 0.4-0.7 $\mu$g m$^{-3}$ yr$^{-1}$ in central Europe and most of the UK, with $PM_{2.5}$ negative trends being slightly weaker than those for $PM_{10}$, with the total reductions of annual mean concentrations by between 2 and 5 (7 for $PM_{10}$) $\mu$g m$^{-3}$ (or between 10 and 30 %) across most of Europe (by 0.5-2 $\mu$g m$^{-3}$ in Fennoscandia, north-west of Russia and Eastern Europe) during the studied period.

That would mean that the annual mean PM concentrations decreased by between 2 and 5 (7 for $PM_{10}$) $\mu$g m$^{-3}$ across most of Europe (by 0.5-2 $\mu$g m$^{-3}$ in Fennoscandia, north-west of Russia and Eastern Europe) during the 2000-2010 period. In relative terms, the decrease of annual mean $PM_{10}$ and $PM_{2.5}$ was between 10 and 20 % over most of Europe (up to 25-30 % in Germany, the Netherlands, Belgium, parts of the UK, Portugal, north/center of Italy and large parts of Scandinavia) from 2000 to 2010. We find that the modelled PM trends are fairly consistent with emission reductions in the ECLIPSE_V5 data set used here. Among possible reasons for deviations between the modelled and observed PM trends are emission uncertain-

ties, impacts of inter-annual variability in meteorological conditions (on pollutant transport and removal, secondary aerosol formation, natural PM emissions etc.), model uncertainties associated with aerosol formation and removal processes, i.e. SOA formation, cloud pH dependency of SO4 formation, heterogeneous chemistry (including gas/aerosol partitioning of anthro-

pogenic precursors and aerosol formation on base cations of natural origin), SO2 and NH3 co-deposition etc.. Not accounting for forest fires in EDT simulations should also affect the accuracy of simulated PM trends, at least in the regions of large fires, whilst this does not appear to have a major impact on the modelled trends at the EDT sites. Furthermore, we find a fairly good general agreement in PM trends estimated by the individual models, with the inter-model variability below 30-40 % over much of Europe (up to 50-60 % in northern and eastern parts of EDT domain). Somewhat greater variability in the modelled $PM_{10}$

trends reflects larger uncertainties in modelling of the coarse fraction of PM, which is mostly due to natural origin.

Averaged over measurement sites (26 for $PM_{10}$ and 13 for $PM_{2.5}$), the mean ensemble simulated trends are -0.24 $\mu$g m$^{-3}$ yr$^{-1}$ for $PM_{10}$ and -0.21 $\mu$g m$^{-3}$ yr$^{-1}$ for $PM_{2.5}$, which are somewhat weaker than the observed trends of -0.35 and -0.40 $\mu$g m$^{-3}$ yr$^{-1}$, respectively. This is partly related to models' underestimation of PM concentrations. The correspondence be-tween model results and observations appears better in terms of relative trends for the same period, which are -1.7 and -2.0

% yr$^{-1}$ from the model ensemble and -2.1 and -2.9 % yr$^{-1}$ from the observations for $PM_{10}$ and $PM_{2.5}$ respectively. We see somewhat larger spatial variability of observed PM trends with respect to the modelled trends across Europe and within individual countries, which could partly be explained by the uncertainties associated with national sectoral emissions and their spatial distribution. In addition, the regional models have difficulties to accurately resolve pollution at some of the sites located in the regions with complex topography. The observations identify significant trends for $PM_{10}$ at 56 % of the sites and for

$PM_{2.5}$ at 36 % of the sites, which is somewhat less than those identified by the models.

The strongest decreasing trends and the largest number of sites (and larger areas) with significant trends were observed and modelled for summer concentrations of $PM_{10}$ and $PM_{2.5}$. On the other hand for the winter season, the model ensemble identifies significant PM trends for very limited areas, mostly in southern parts of Europe, whilst the observed trends are not significant at any of the sites for $PM_{10}$ and only at 3 out of 14 sites for $PM_{2.5}$. One important reason for that is the very modest

reductions and even increases in the emissions of primary PM from residential heating in winter.

The analysis reveals a considerable variability of the role of the individual aerosols in $PM_{10}$ trends across European coun-tries. The multi-model simulations, supported by available observations, point to decreases in $SO_4^{-2}$ concentrations playing an overall dominant role, although with some exceptions. Namely, we see relatively large contributions of the trends of $NH_4^+$ and $NO_3^-$ to $PM_{10}$ decreasing trends in Germany, Denmark, Poland and the Po Valley, while the reductions of primary PM

emissions appears to be a dominant factor in bringing down $PM_{10}$ in France, Norway, Portugal, Greece and parts of the UK and Russia.

The analysis also suggests that year-to-year variability in meteorological conditions masks decreasing PM trends due to emission reductions, leading to non-significant trends in many areas and at many monitoring sites between 2000 and 2010. Still, the role of emission reduction measures is pronounced in the regions with greater reductions, where significant trends

of $PM_{10}$ and $PM_{2.5}$ are both modelled and observed. The EDT model results show that the mean annual trends during the period of 1990-2010 were stronger compared with those in the 2000-2010 period, which is a consequence of larger emission

reductions in the 1990s compared with those in the 2000s. The EDT model ensemble estimates that annual mean $PM_{10}$ and $PM_{2.5}$ concentrations decreased by between 5 and 15 $\mu g\ m^{-3}$ across most of Europe (by 2-5 $\mu g\ m^{-3}$ in the Northern Europe) from 1990 to 2021.

*Data availability.* Technical details of the EURODELTA project simulations that permit the replication of the experiment are available on the wiki of the EMEP Task Force on Measurement and Modelling (https://wiki.met.no/emep/emep-experts/tfmmtrendeurodelta, last access: 22 November 2021), which also includes ESGF links to corresponding input forcing data. The EURODELTATrends model results are made available for public use on the AeroCom server (information to gain access to the AeroCom server are available at https://wiki.met.no/aerocom/user-server, last access: 22 November 2021). Model input and output data are permanently stored under the /metno/aerocom-users-database/EURODELTA

folder on the AeroCom Server. See Colette et al. (2017) for full terms and conditions for the use of these data.

The original data used for calculating aggregated concentrations are all available from the database infrastructure EBAS (https://ebas.nilu.no)

# 9 Tables and Figures

**Table 1.** Observed and modelled (ensemble mean and individual models), $PM_{10}$ and $PM_{2.5}$ annual mean trends for the period 2000-2010, averaged over all trend-sites. The standard deviation is included in parentheses. Units are $\mu g\ m^{-3}\ yr^{-1}$ and $\%\ yr^{-1}$ for absolute (Abs) and relative (Rel) trends, respectively. The number of sites with significant trends identified by observations and models (Nsign) is also provided.

| Parameter | Trends | Obs | ENSmean | CHIM | EMEP | LOTO | MATCH | MINNI | POLR |
|---|---|---|---|---|---|---|---|---|---|
| $PM_{10}$ | Abs | -0.35(0.23) | -0.24(0.09) | -0.22(0.09) | -0.33(0.11) | -0.23(0.11) | -0.27(0.08) | -0.24(0.10) | -0.16(0.16) |
| 26 sites | Rel | -2.1(1.19) | -1.7(0.4) | -1.6(0.36) | -2.2(0.36) | -1.6(0.60) | -2.1(0.43) | -1.6(0.41) | -1.4(1.27) |
| | Nsign | 14 | | 14 | 23 | 16 | 20 | 10 | 14 |
| $PM_{2.5}$ | Abs | -0.40(0.38) | -0.21(0.10) | -0.21(0.1) | -0.26(0.12) | -0.19(0.11) | -0.21(0.08) | -0.21(0.1) | -0.21(0.14) |
| 13 sites | Rel | -2.9(1.48) | -2.0(0.33) | -1.8(0.35) | -2.4(0.43) | -2.0(0.53) | -1.9(0.40) | -1.8(0.44) | -2.1(0.77) |
| | Nsign | 5 | | 9 | 12 | 8 | 11 | 7 | 8 |

**Table 2.** Observed (Obs) and modelled (6-model ensemble; ENS) mean seasonal trends and Standard deviations (in parentheses) for 2000-2010 at all trend-sites. Units are $\mu g\ m^{-3}\ yr^{-1}$ and $\%\ yr^{-1}$ for absolute (Abs) and relative (Rel) trends, respectively. The numbers of sites with significant trends are given in square brackets.

| Parameter | | winter | spring | summer | autumn |
|---|---|---|---|---|---|
| $PM_{10}$ | Obs ($\mu g\ m^{-3}\ yr^{-1}$) | -0.19(0.29) [0] | -0.33(0.27) [5] | -0.56(0.31) [12] | -0.26(0.25) [4] |
| | ENS ($\mu g\ m^{-3}\ yr^{-1}$) | -0.13(0.10) [3] | -0.28(0.13) [10] | -0.32(0.17) [17] | -0.26(0.13) [7] |
| $PM_{2.5}$ | Obs ($\mu g\ m^{-3}\ yr^{-1}$) | -0.38(0.51) [4] | -0.42(0.47) [4] | -0.51(0.34) [10] | -0.27(0.34) [2] |
| | ENS ($\mu g\ m^{-3}\ yr^{-1}$) | -0.10(0.10) [1] | -0.23(0.15) [3] | -0.26(0.13) [8] | -0.24(0.14) [7] |
| $PM_{10}$ | Obs ($\%\ yr^{-1}$) | -1.4(1.7) | -1.8(1.2) | -2.9(1.0) | -1.6(1.8) |
| | ENS ($\%\ yr^{-1}$) | -1.0(0.8) | -1.8(0.6) | -2.4(0.9) | -1.8(0.7) |
| $PM_{2.5}$ | Obs ($\%\ yr^{-1}$) | -2.8(2.2) | -2.7(1.7) | -3.8(1.5) | -2.0(1.9) |
| | ENS ($\%\ yr^{-1}$) | -0.9(1.1) | -1.8(0.9) | -2.5(0.9) | -2.1(0.6) |

|  | CHIM | EMEP | LOTO | MATCH | MINNI | POLR | mean |
|---|---|---|---|---|---|---|---|
| 2000 | -2 | 2 | -6 | -11 | -4 | -24 | -8 |
| 2001 | -4 | -8 | -4 | -9 | -8 | -30 | -11 |
| 2002 | -7 | -14 | -13 | -16 | -14 | -35 | -17 |
| 2003 | -8 | -13 | -16 | -12 | -12 | -36 | -16 |
| 2004 | -7 | -14 | -10 | -14 | -9 | -33 | -15 |
| 2005 | -7 | -11 | -5 | -12 | -7 | -31 | -12 |
| 2006 | -6 | -12 | 4 | -8 | -10 | -32 | -11 |
| 2007 | -3 | -10 | -11 | -10 | -7 | -28 | -12 |
| 2008 | -4 | -10 | -8 | -11 | -8 | -28 | -12 |
| 2009 | 0 | -8 | 2 | -5 | -3 | -27 | -7 |
| 2010 | -9 | -16 | -15 | -18 | -17 | -35 | -18 |
| mean | -5 | -10 | -7 | -11 | -9 | -31 | -12 |

|  | CHIM | EMEP | LOTO | MATCH | MINNI | POLR | mean |
|---|---|---|---|---|---|---|---|
| 2000 | -2 | -5 | -6 | -3 | 6 | -2 | -2 |
| 2001 | -18 | -21 | -18 | -18 | -10 | -22 | -18 |
| 2002 | -19 | -25 | -18 | -21 | -12 | -26 | -20 |
| 2003 | -18 | -20 | -25 | -16 | -6 | -25 | -18 |
| 2004 | -17 | -23 | -15 | -16 | -6 | -23 | -17 |
| 2005 | -21 | -24 | -15 | -17 | -9 | -22 | -18 |
| 2006 | -20 | -24 | -12 | -16 | -9 | -25 | -18 |
| 2007 | -18 | -25 | -18 | -16 | -9 | -22 | -18 |
| 2008 | -11 | -17 | -7 | -6 | 1 | -12 | -9 |
| 2009 | -7 | -14 | 1 | -4 | 6 | -12 | -5 |
| 2010 | -17 | -24 | -14 | -14 | -10 | -19 | -16 |
| mean | -15 | -20 | -13 | -13 | -5 | -19 | -14 |

**Figure 1.** Model biases (%) with respect to observations for $PM_{10}$ (left) and $PM_{2.5}$ (right) for the period 2000-2010. Note: coarse sea salt is excluded in $PM_{10}$ from POLR.

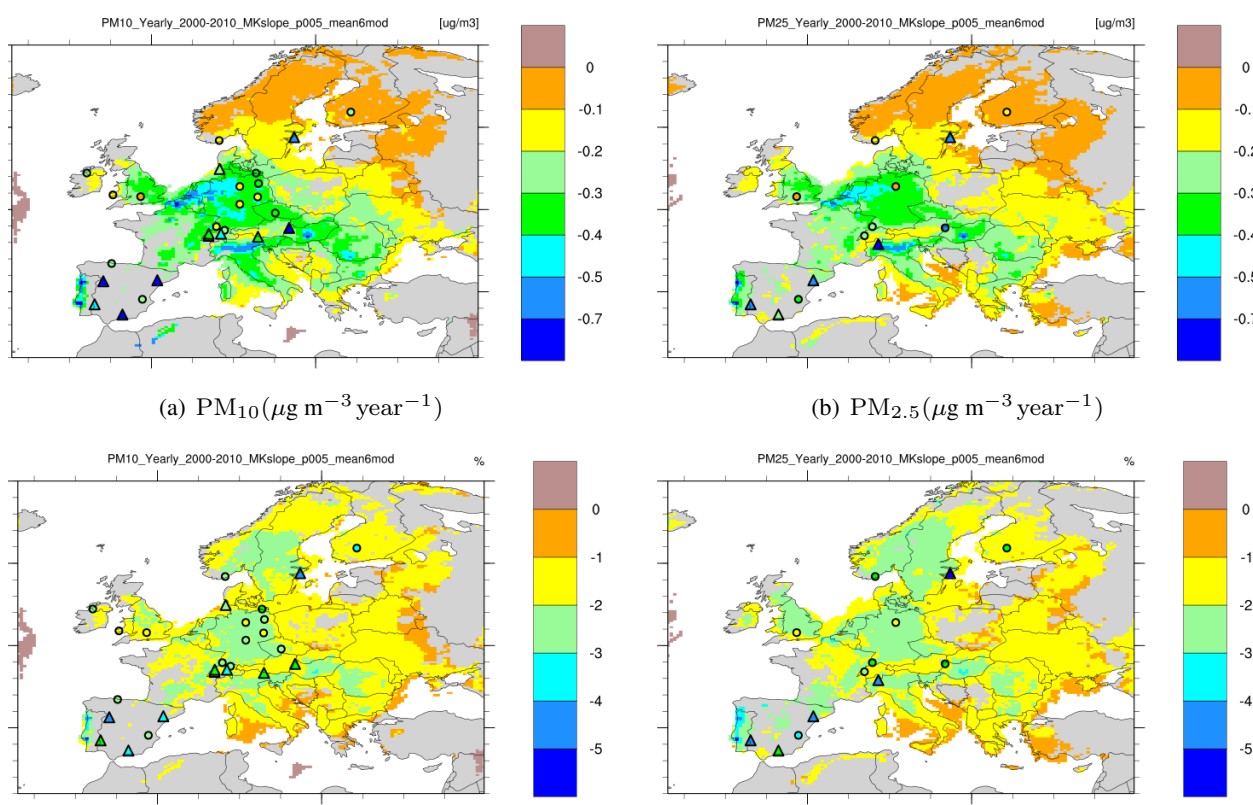

(a) $PM_{10}(\mu g\,m^{-3}\,year^{-1})$  (b) $PM_{2.5}(\mu g\,m^{-3}\,year^{-1})$

**Figure 2.** Mean Sen's slopes for $PM_{10}$ and $PM_{2.5}$ trends in 2000-2010: absolute (a, b) and relative (c, d) slopes calculated by the 6-model ensemble (described in Colette et al. (2017a)), Appendix A3. Modelled trends – coloured contour map (grey or white means non-significant trends) and observed trends - coloured triangles (significant) and circles (non-significant).

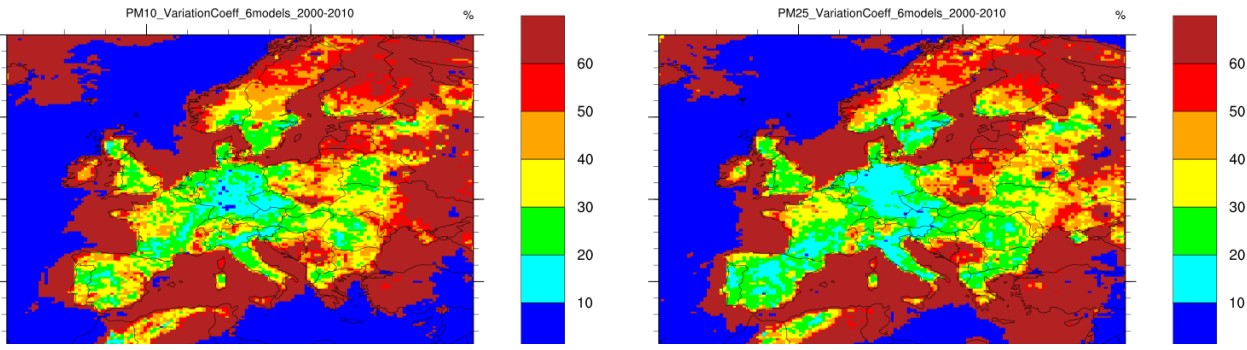

**Figure 3.** The Coefficient of Variation of $PM_{10}$ (left) and $PM_{2.5}$ (right) trends simulated with the individual models relative to the 6-model ensemble mean for the period 2000-2010.

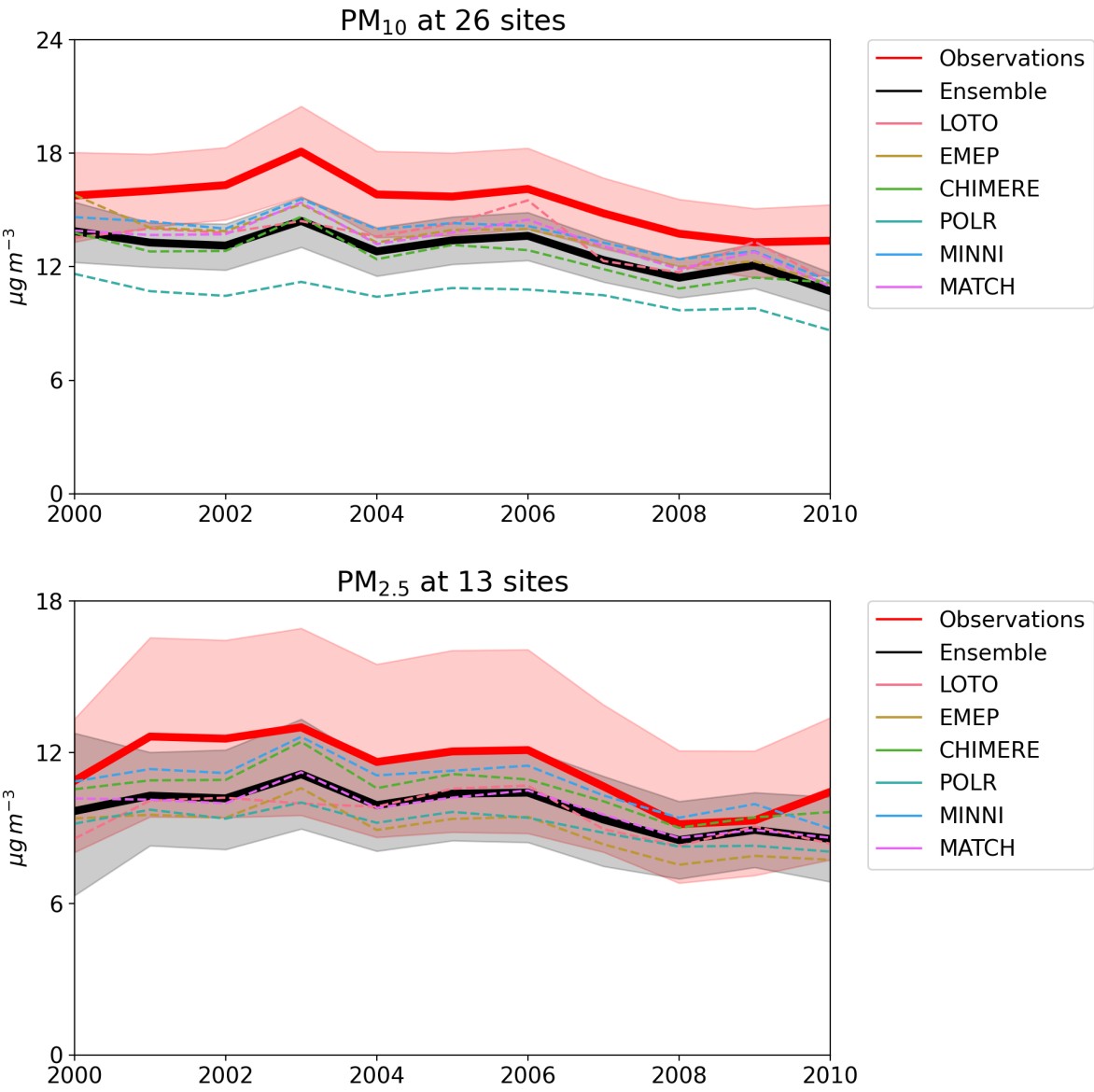

**Figure 4.** Observed and simulated with 6-model ensemble and the individual models annual mean concentrations of $PM_{10}$ (upper panel) and $PM_{2.5}$ (lower panel) for the period 2000-2010, averaged over the trend sites. The 95 % confidence intervals for observed and ensemble modelled PM concentrations are shown with shaded areas. The number of sites with available observations for the individual years can be found in Table 1. (Note: $PM_{10}$ from POLR does not include coarse sea salt, see the text for explanations).

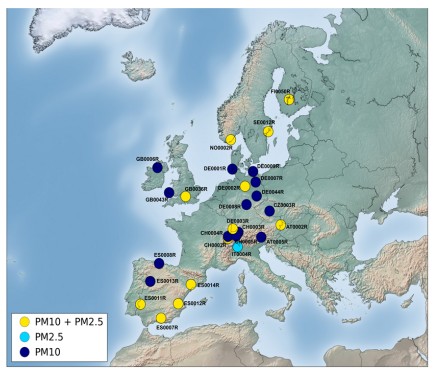

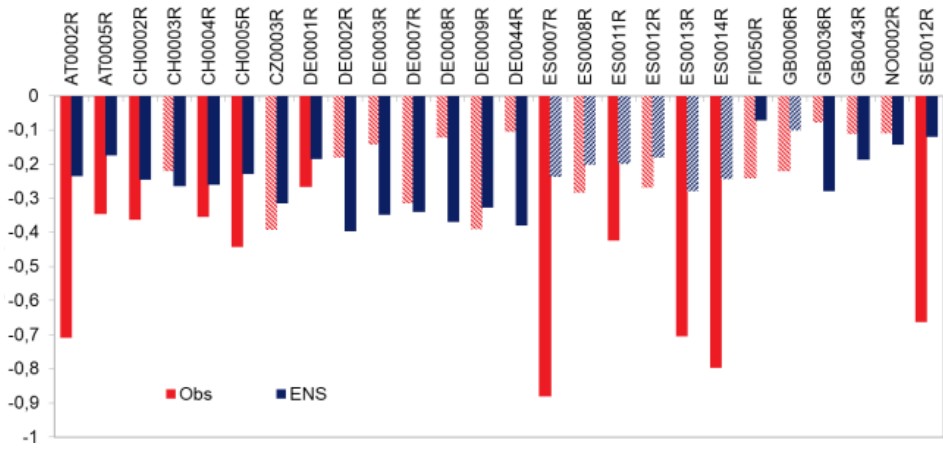

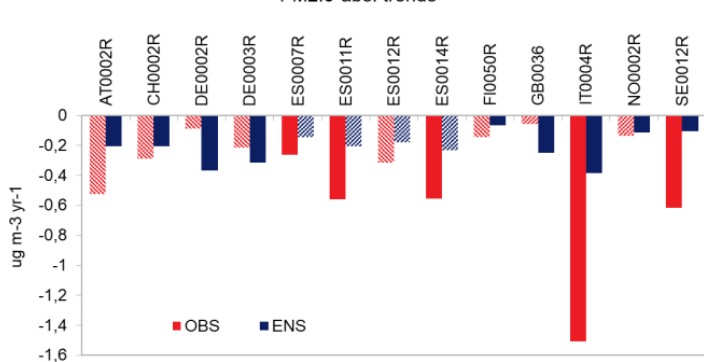

**Figure 5.** Observed and modelled (6-model ensemble) trend slopes ($\mu$g m$^{-3}$ yr$^{-1}$) for the period 2000-2010 at the trend sites for PM$_{10}$ (middle) and PM$_{2.5}$ (bottom). Significant modelled trends are shown in dark blue, not-significant in light blue. Sites with non-significant trends are represented by striped bars. The trend sites are shown on the map (upper panel).

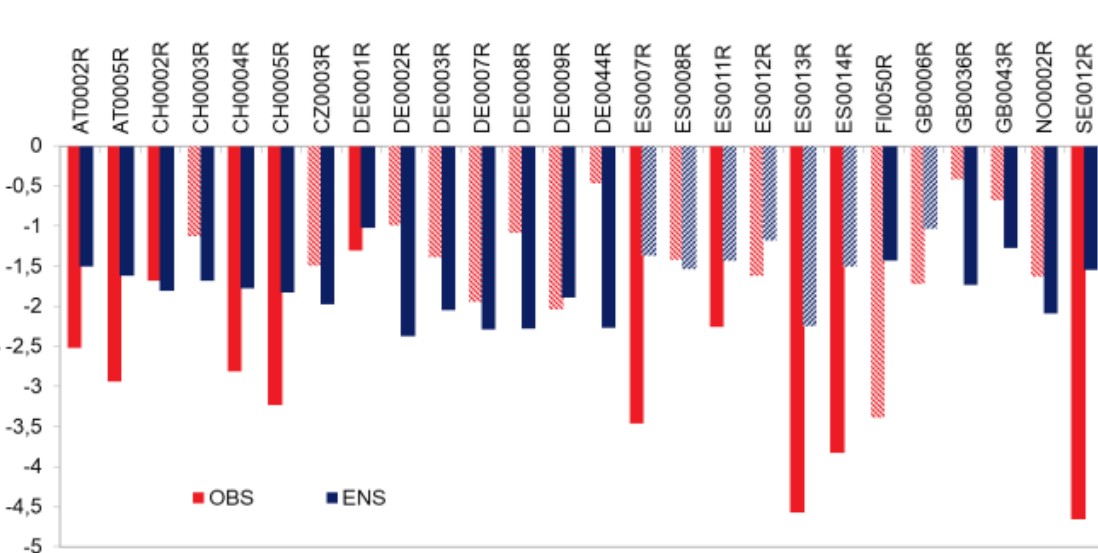

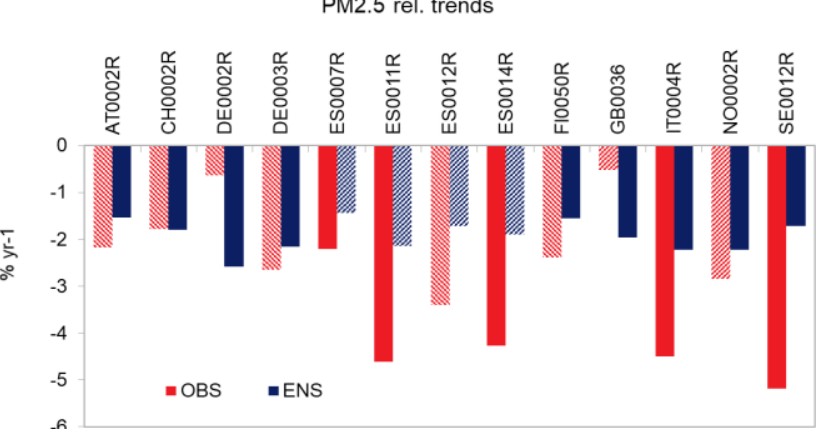

**Figure 6.** Same as Fig. 5, but for relative trends (% yr$^{-1}$). The trend sites are shown in Fig. 5, upper panel.

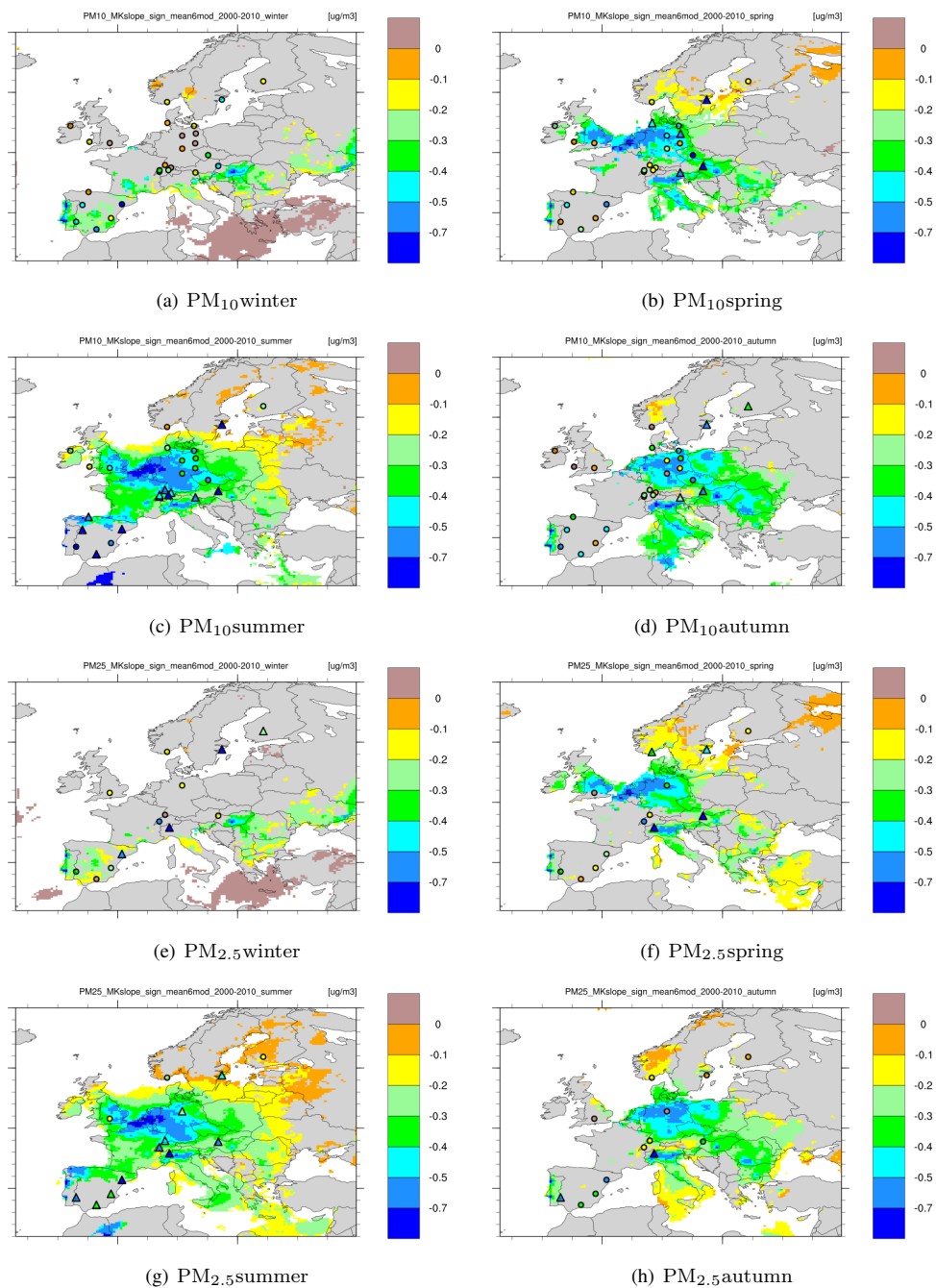

(a) PM$_{10}$winter

(b) PM$_{10}$spring

(c) PM$_{10}$summer

(d) PM$_{10}$autumn

(e) PM$_{2.5}$winter

(f) PM$_{2.5}$spring

(g) PM$_{2.5}$summer

(h) PM$_{2.5}$autumn

**Figure 7.** Mean Sen's slopes for PM$_{10}$ and PM$_{2.5}$ seasonal trends for 2000-2010, calculated by the 6-model ensemble (see Figure 2 for explanation).

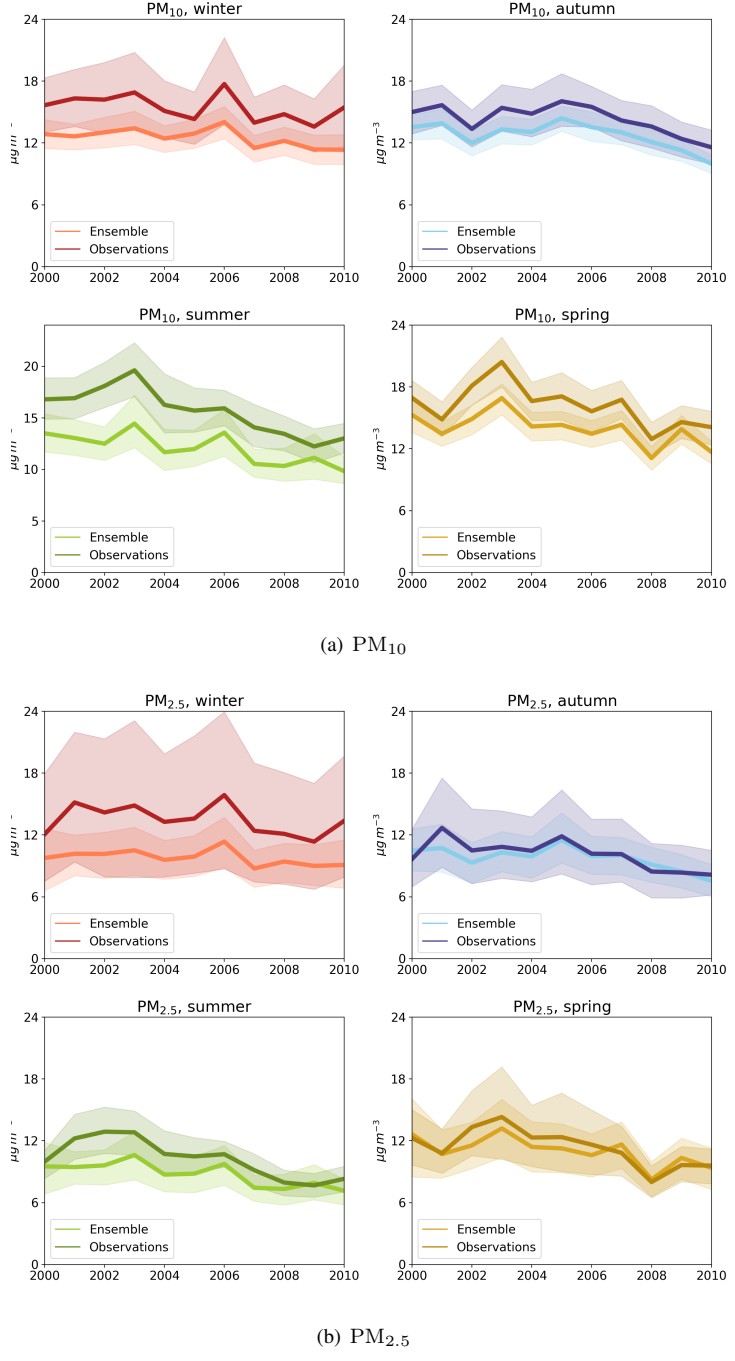

**Figure 8.** Observed and simulated with 6-model ensemble changes in seasonal mean $PM_{10}$ and $PM_{2.5}$ concentrations in the period 2000-2010, averaged over the trend-sites. The 95 % confidence intervals are shown with shaded areas. The number of sites with available observations for the individual years can be found in Table 2.

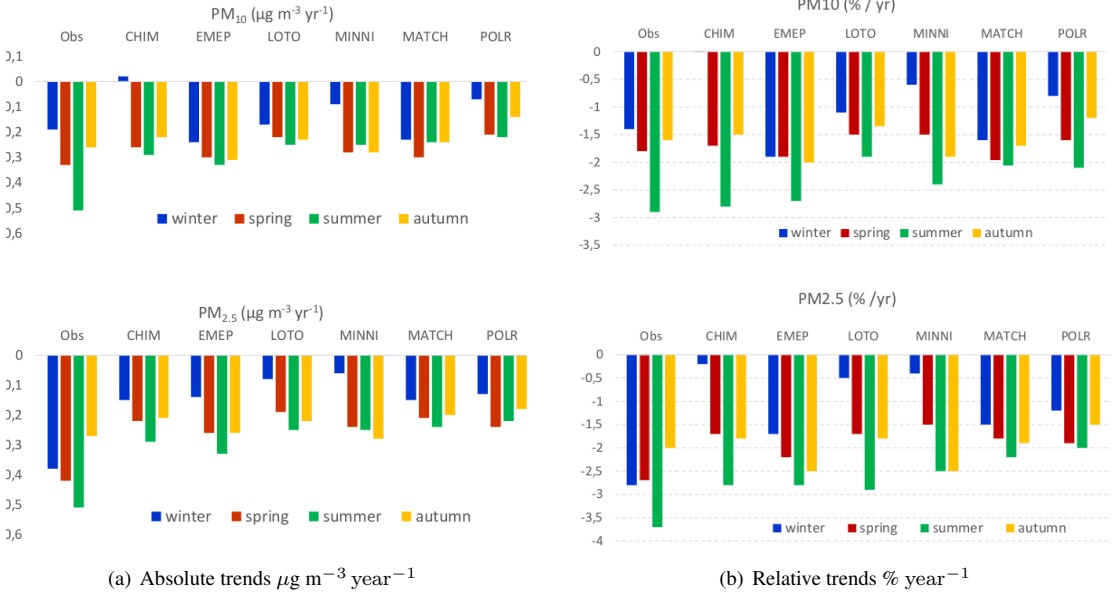

(a) Absolute trends $\mu g\,m^{-3}\,year^{-1}$

(b) Relative trends $\%\,year^{-1}$

**Figure 9.** Mean relative seasonal trends in the period 2000-2010 at the trend-sites for $PM_{10}$ and $PM_{2.5}$: The trends from the observations, the individual models and the 6-model ensemble are shown.

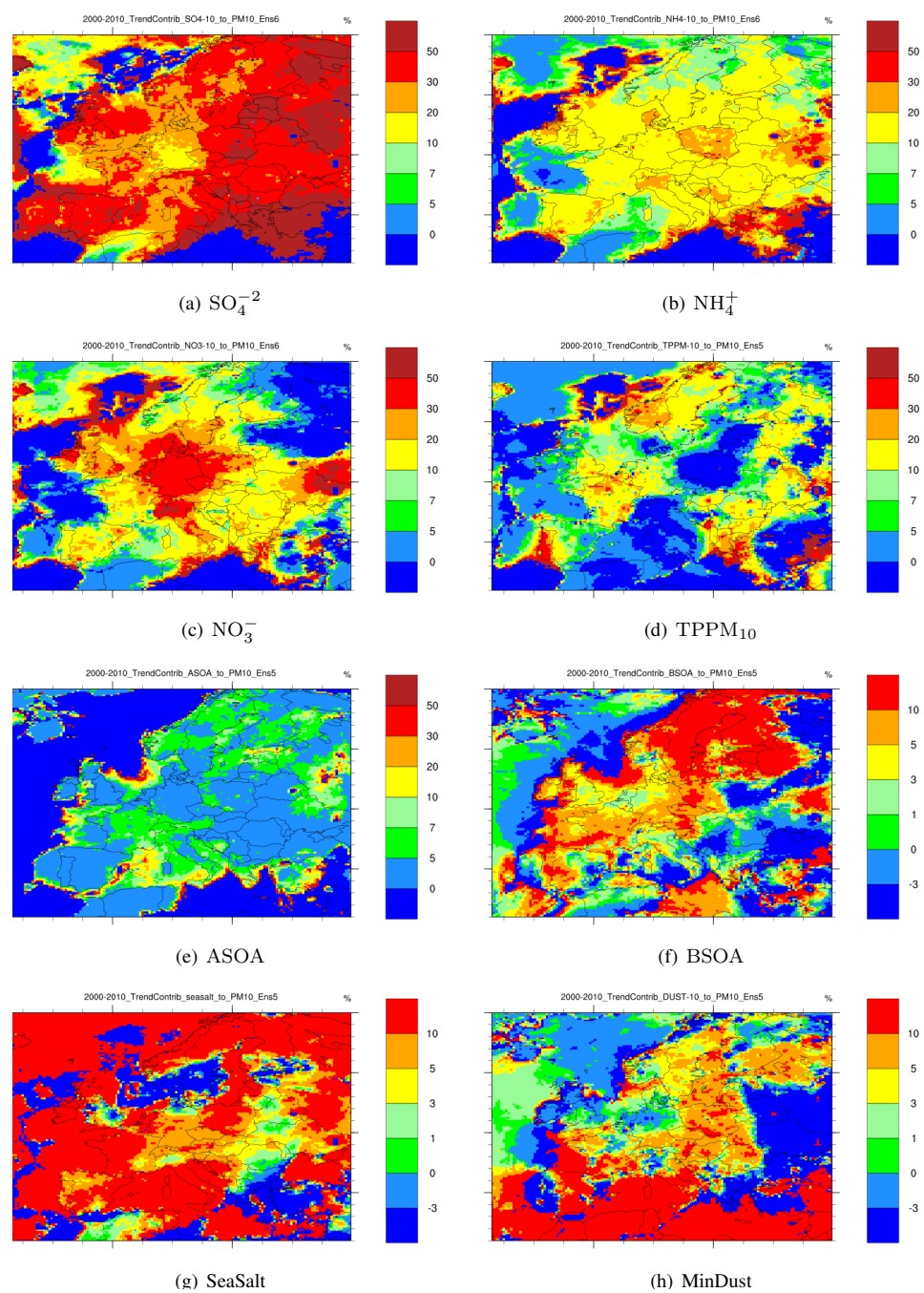

**Figure 10.** Model ensemble simulated relative contribution to $PM_{10}$ 2000-2010 trends from anthropogenic aerosols: $SO_4^{-2}$, $NH_4^+$, $NO_3^-$, total primary $TPPM_{10}$ (except POLR) and anthropogenic SOA (except LOTO), and from natural aerosols: biogenic SOA (except LOTO), sea salt (except POLR) and mineral dust particles (except MATCH). Note that a different colour scale is used for the natural aerosols.

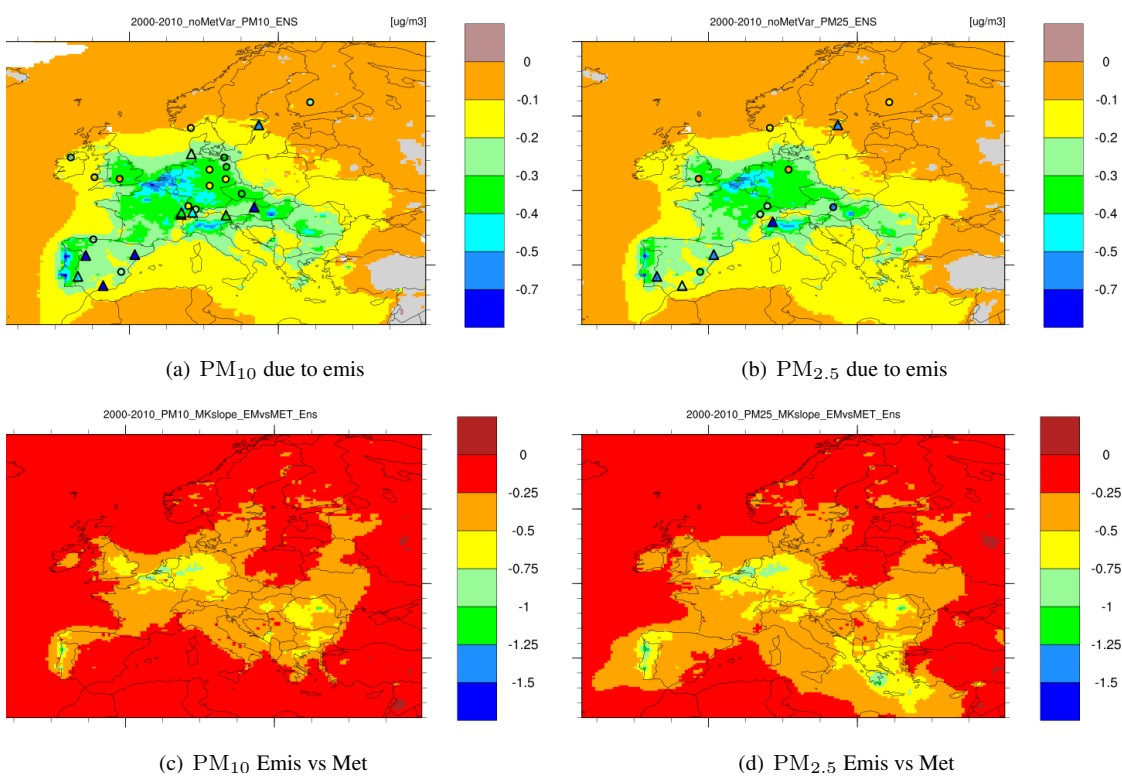

(a) PM$_{10}$ due to emis

(b) PM$_{2.5}$ due to emis

(c) PM$_{10}$ Emis vs Met

(d) PM$_{2.5}$ Emis vs Met

**Figure 11.** PM trends due to emission changes (upper panels) and the ratio of PM changes due to emission changes to those due to inter-annual meteorological variability (lower panels) for PM$_{10}$ and PM$_{2.5}$ in the 2000-2010 period. Observed trends are shown as coloured triangles (significant) and circles (non-significant).

**Appendix A:  Supplementary Figures and Tables**

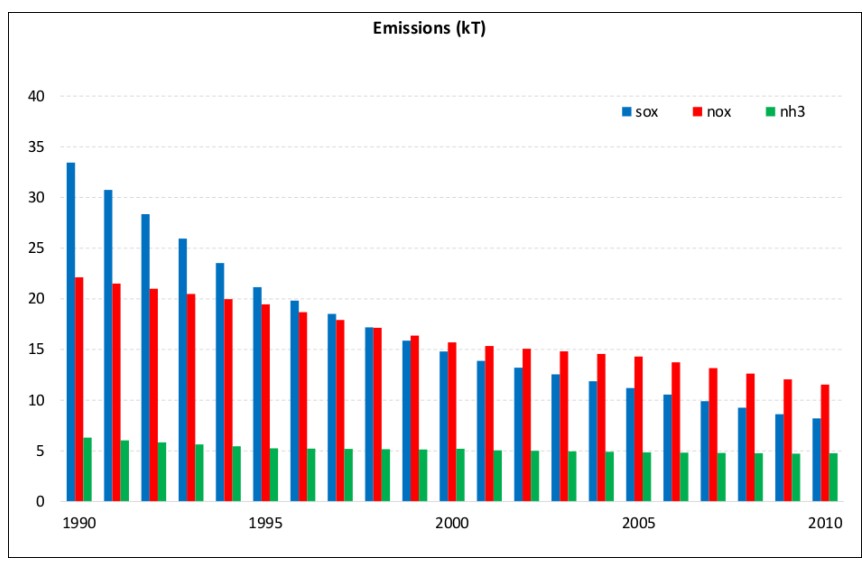

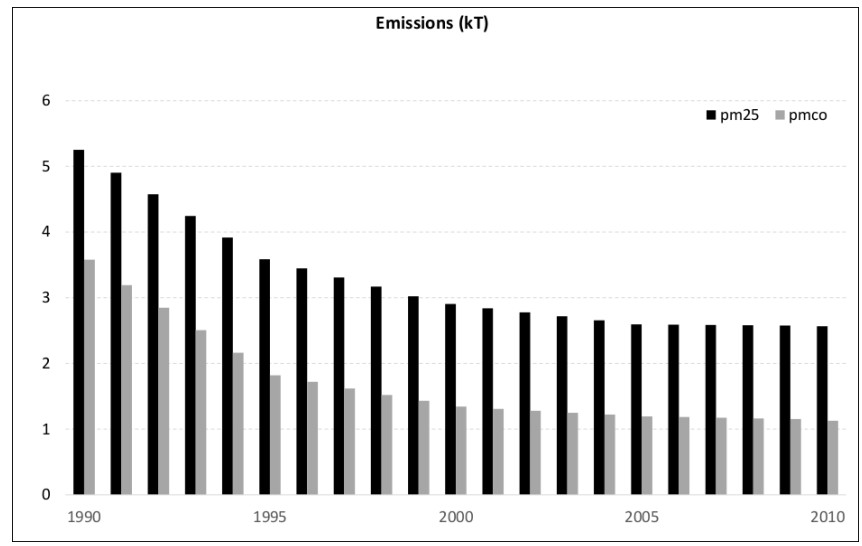

**Figure A1.** Annual emissions of $SO_x$, $NO_x$, $NH_3$, $PM_{2.5}$ and PM coarse (pmco) in the period 1990-2010 (all countries). Units: ktonnes.

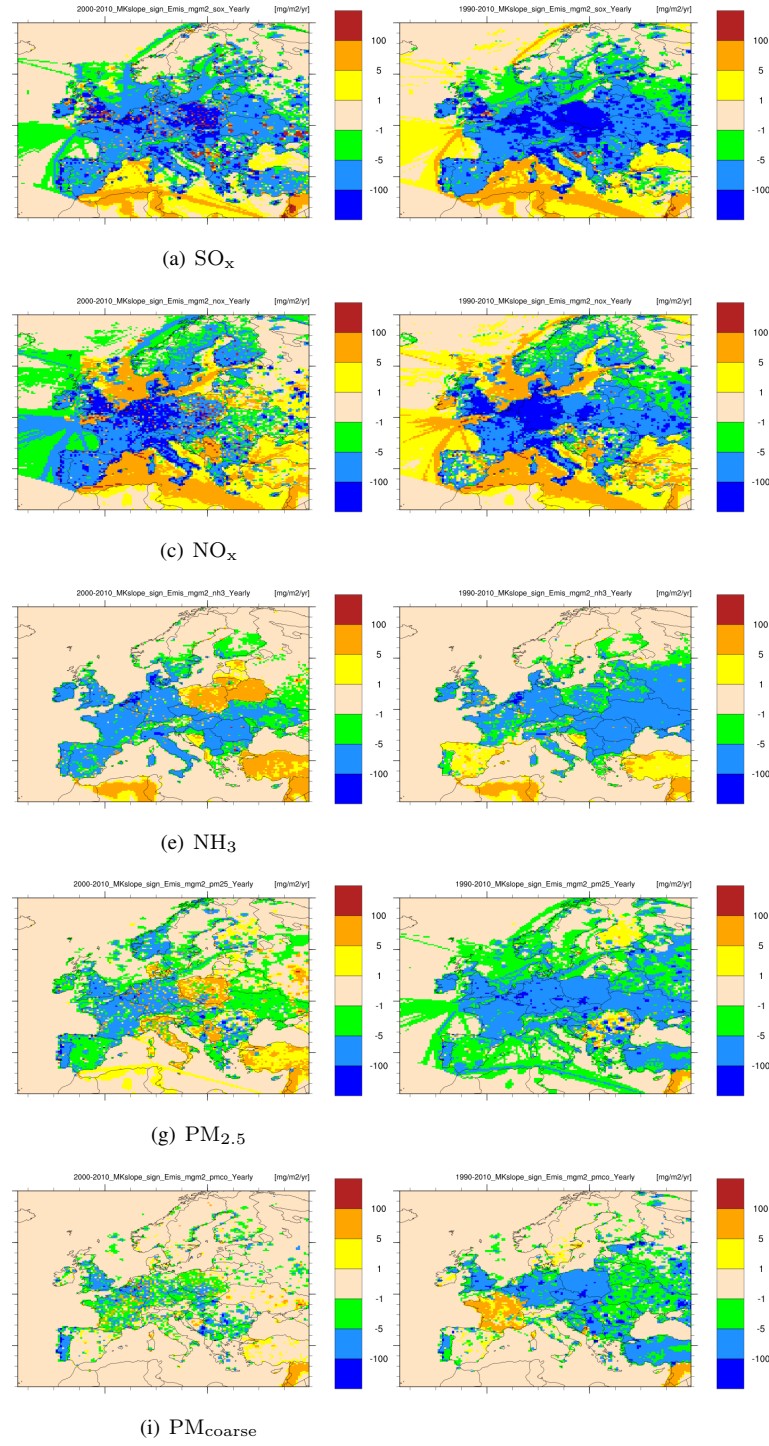

(a) $SO_x$

(c) $NO_x$

(e) $NH_3$

(g) $PM_{2.5}$

(i) $PM_{coarse}$

**Figure A2.** Emission trends for 2000-2010 (left) and 1990-2010 (right).

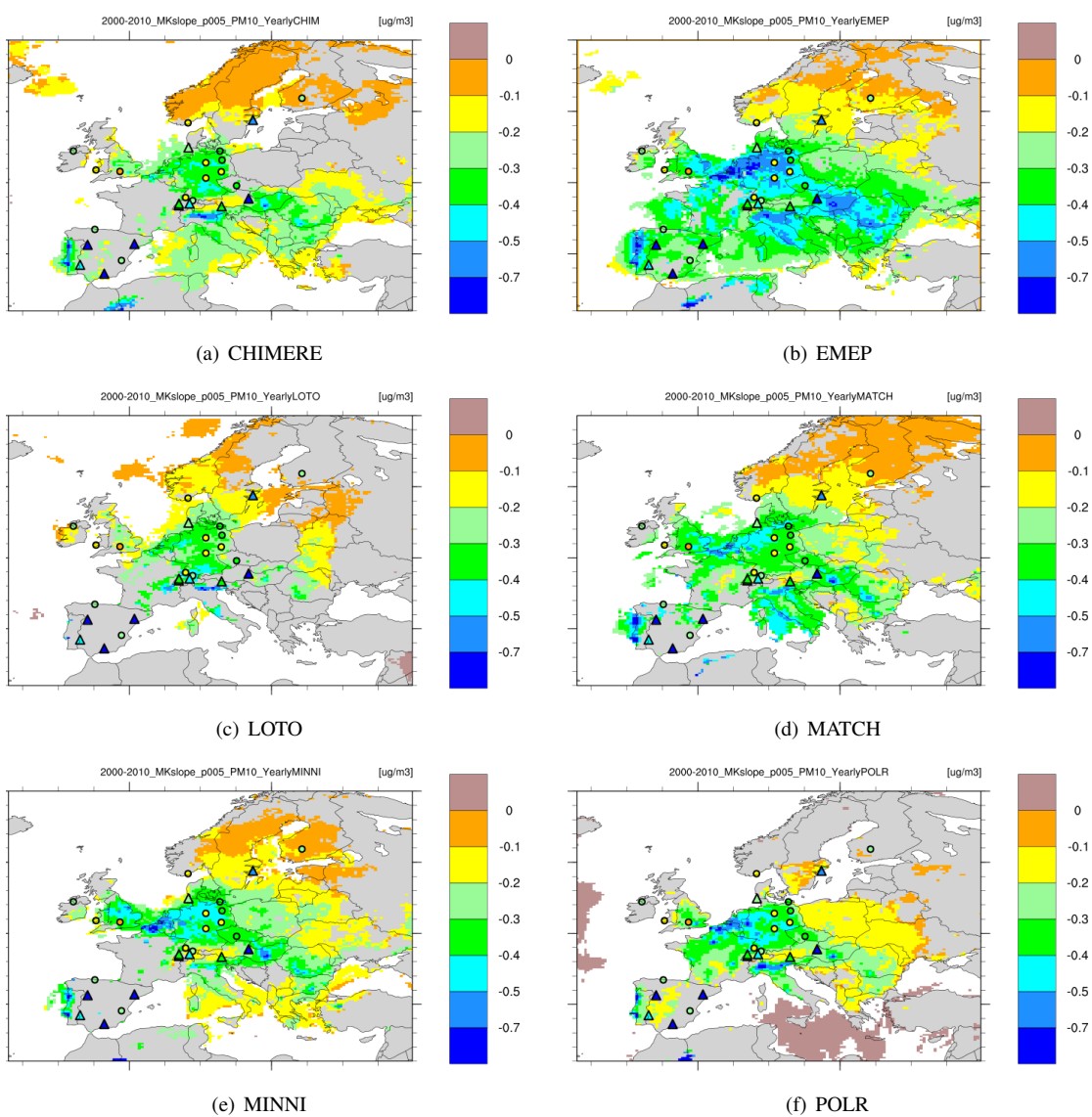

**Figure A3.** Annual mean trends (Sen's slopes) for $PM_{10}$ in the period 2000-2010 as calculated by the individual models. The modelled trends are shown as coloured contour map (grey or white means non-significant trends) and the observed trends as coloured triangles (significant) and circles (non-significant). Units: $\mu g\, m^{-3}\, yr^{-1}$).

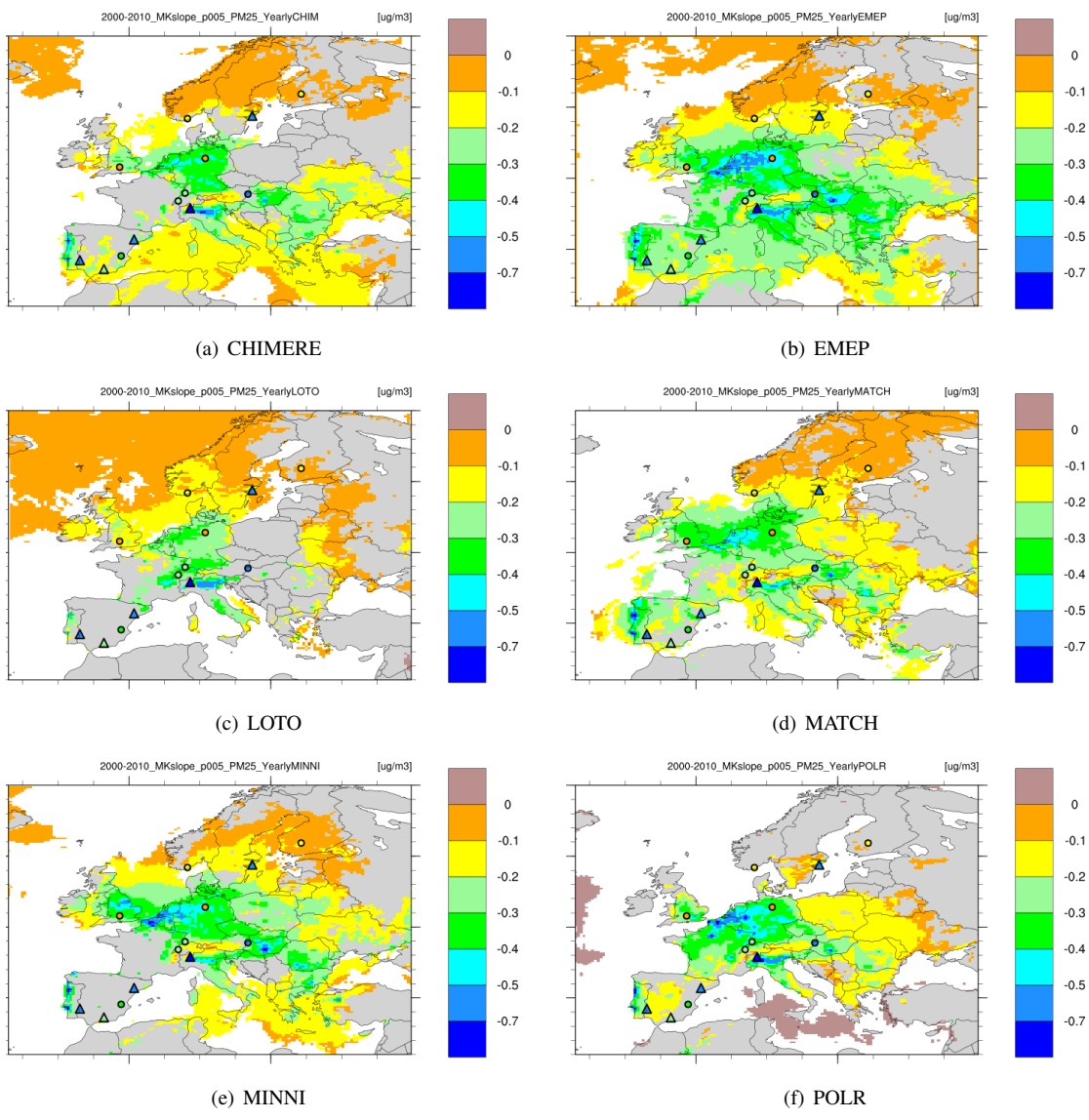

**Figure A4.** Same as Fig. A3, but for $PM_{2.5}$. Units: $\mu g\,m^{-3}\,yr^{-1}$).

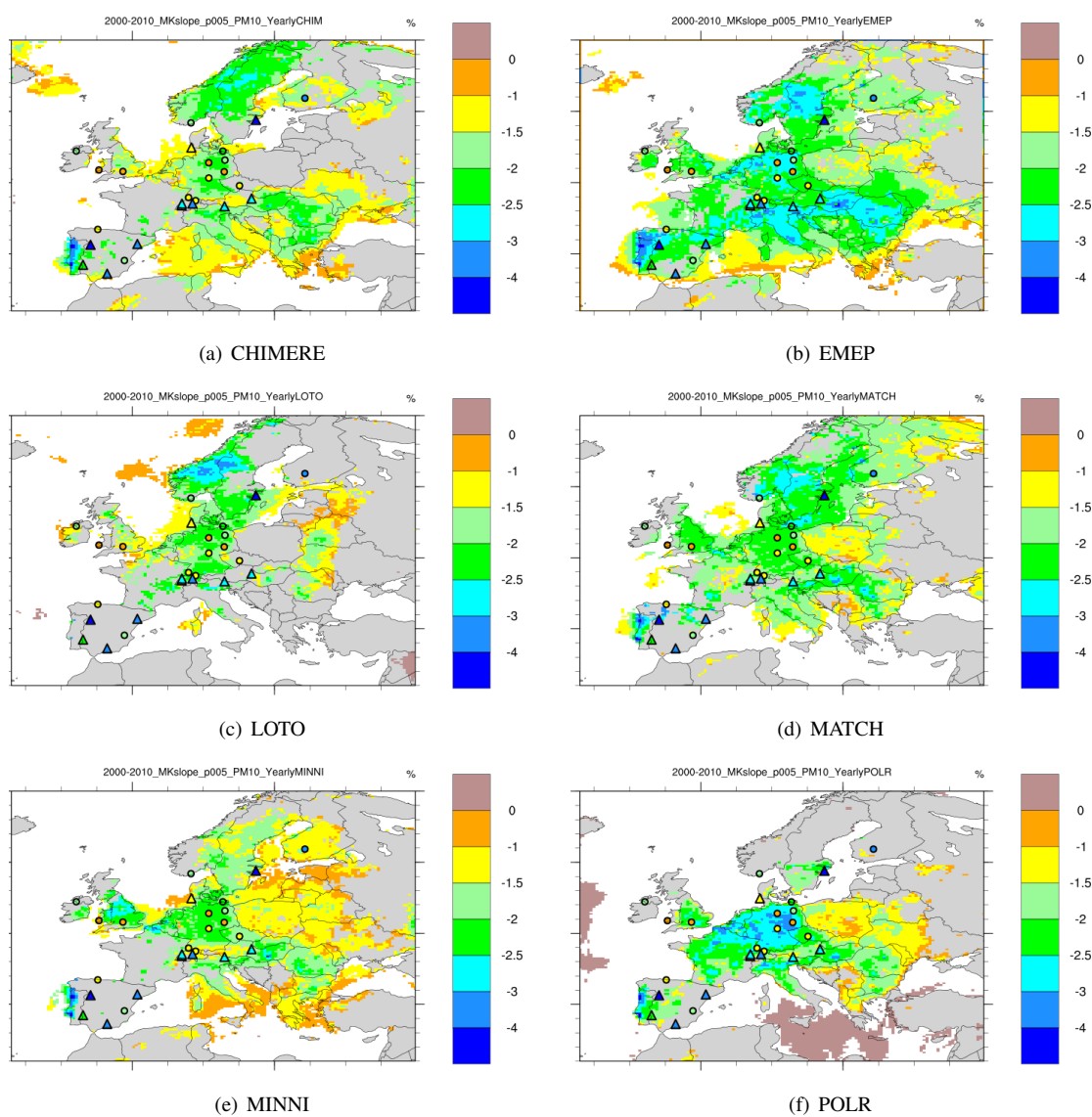

**Figure A5.** Mean Sen's slopes relative to the starting year of 2000 (% yr$^{-1}$) for $PM_{10}$ trends in the period 2000-2010 calculated by the individual models. The modelled trends are shown as coloured contour map (grey or white means non-significant trends) and the observed trends as coloured triangles (significant) and circles (non-significant).

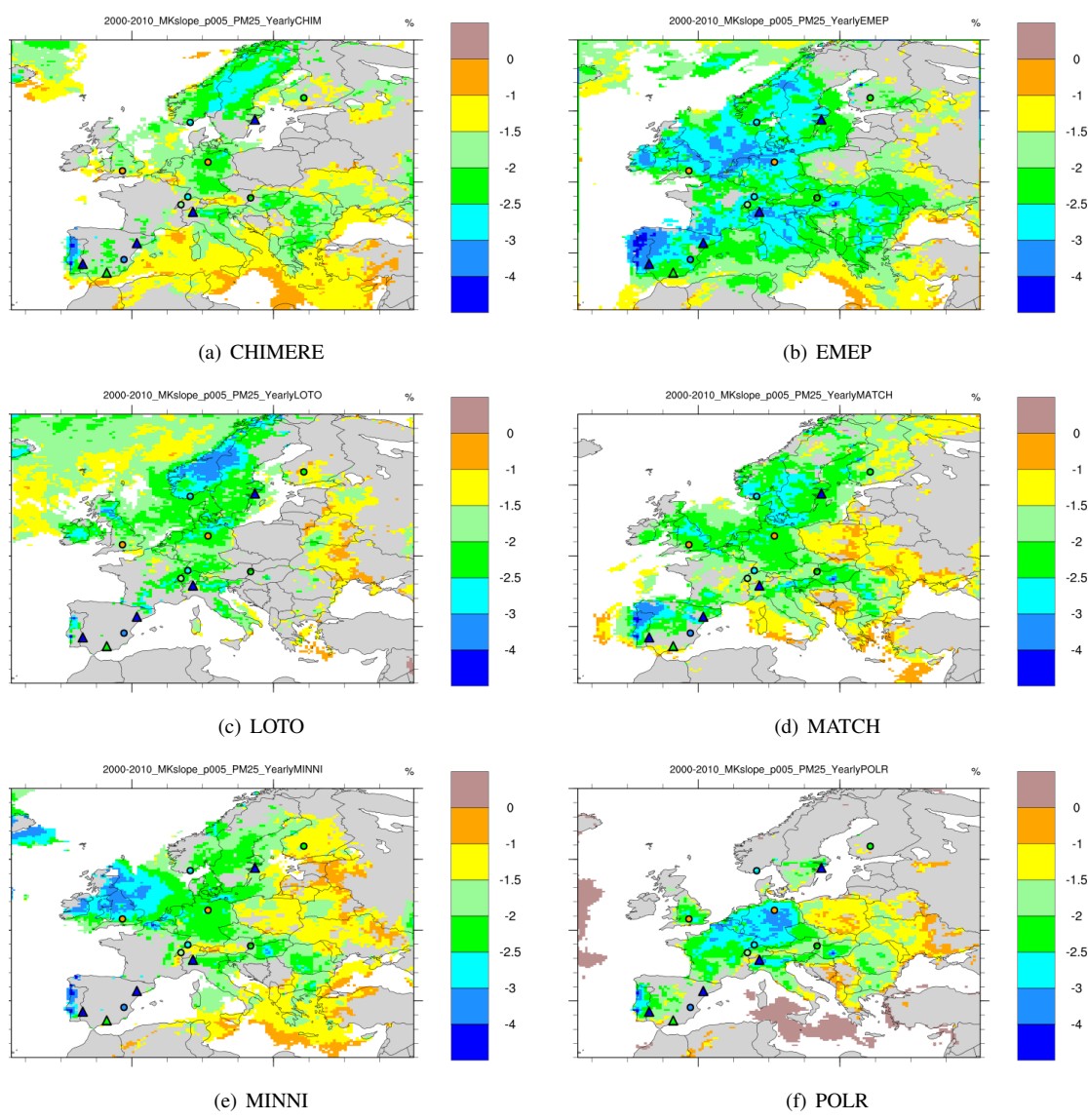

**Figure A6.** Same as Fig. A5, but for $PM_{2.5}$. Units: % $yr^{-1}$.

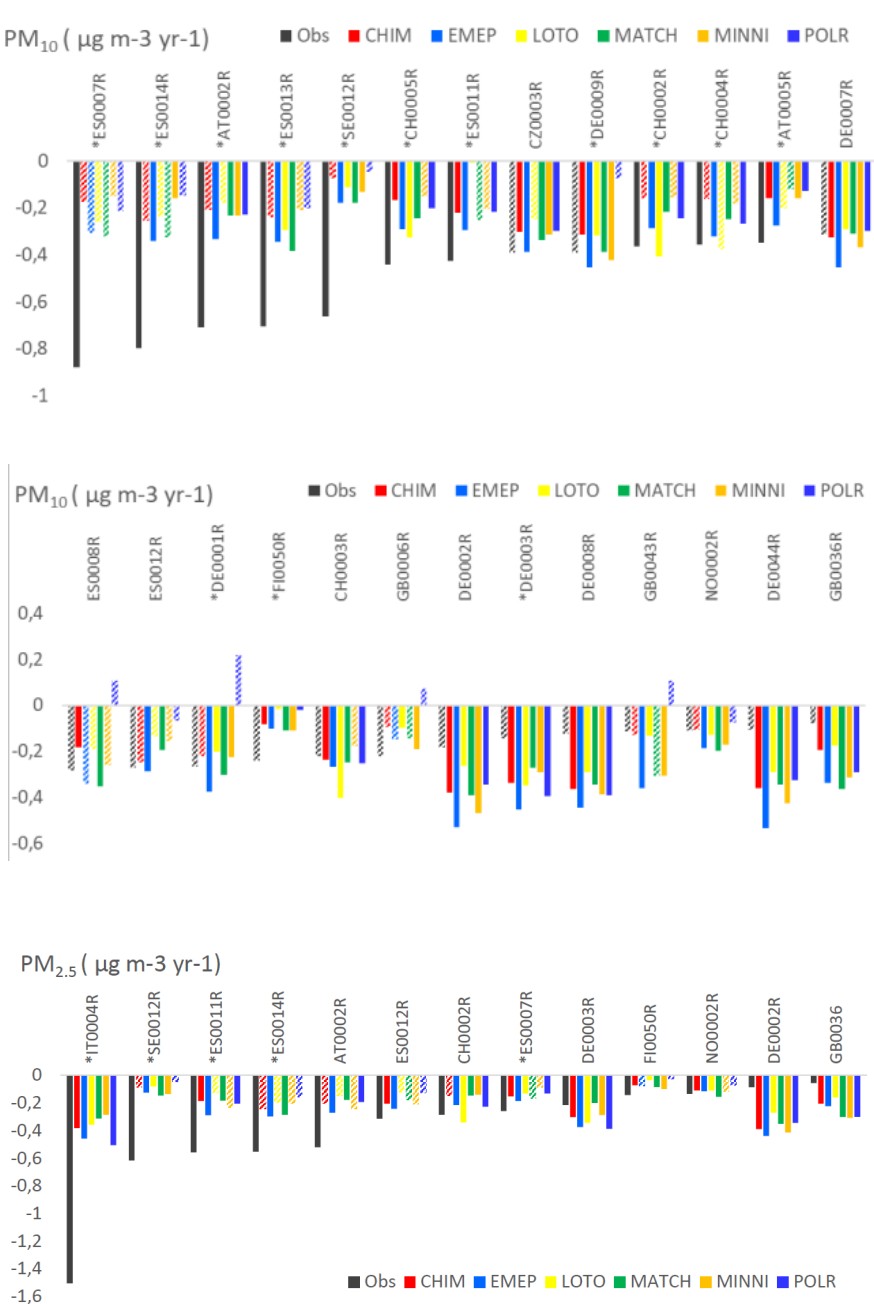

**Figure A7.** Observed and modelled trend slopes ($\mu$g m$^{-3}$ yr$^{-1}$) for the period 2000-2010 at the trend sites: upper two plots for PM$_{10}$, lower plot for PM$_{2.5}$. The sites are sorted by decreasing observed negative trends; insignificant trends are shown as striped bars. Units: $\mu$g m$^{-3}$ yr$^{-1}$.

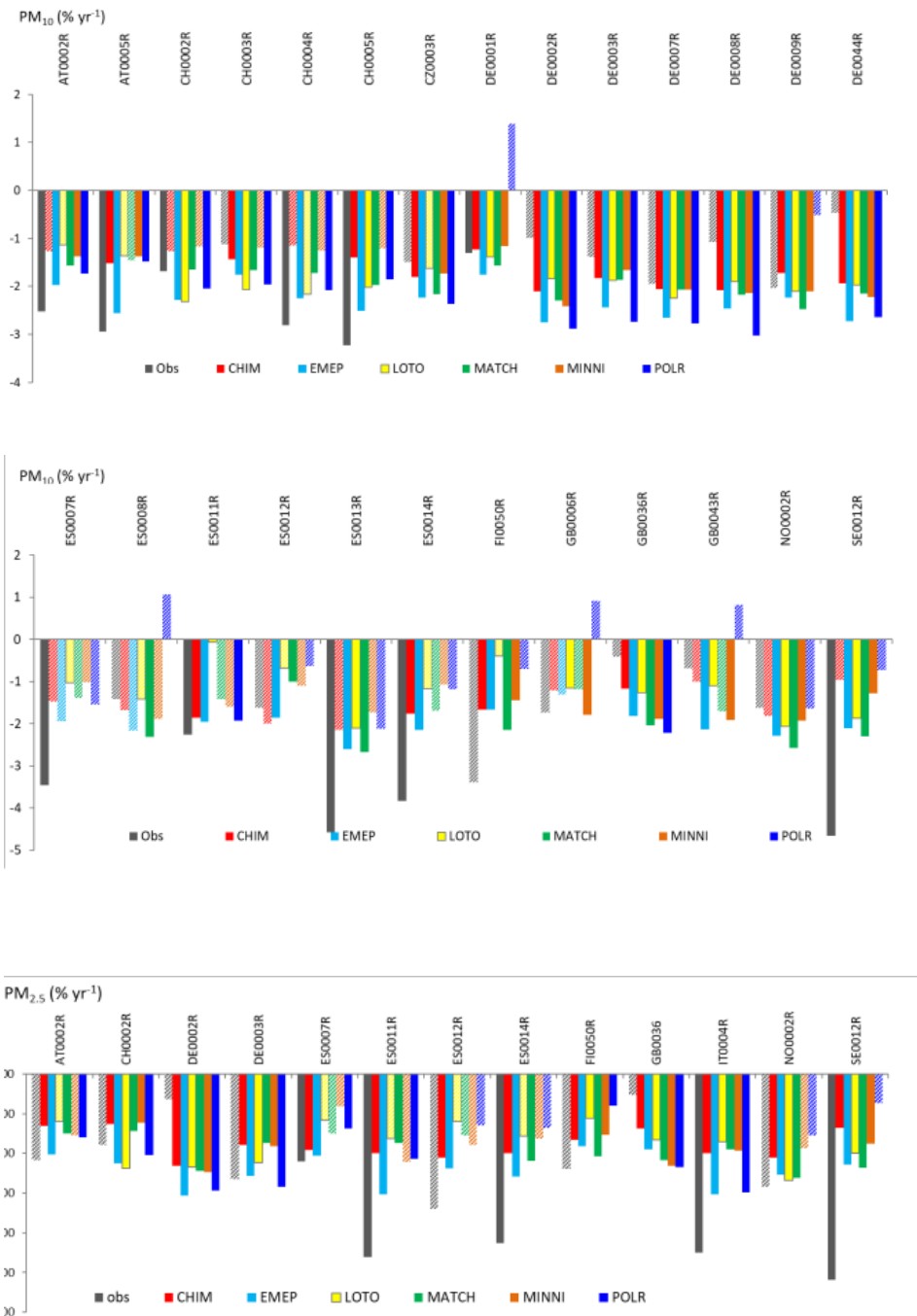

**Figure A8.** Mean observed (black) and modelled (coloured) relative trends for $PM_{10}$ (upper two graphs) and $PM_{2.5}$ (lower graph) in the period 2000-2010 at the individual trend-sites. Insignificant modelled trends are shown as striped bars. Units: $\% \ \mathrm{yr}^{-1}$.

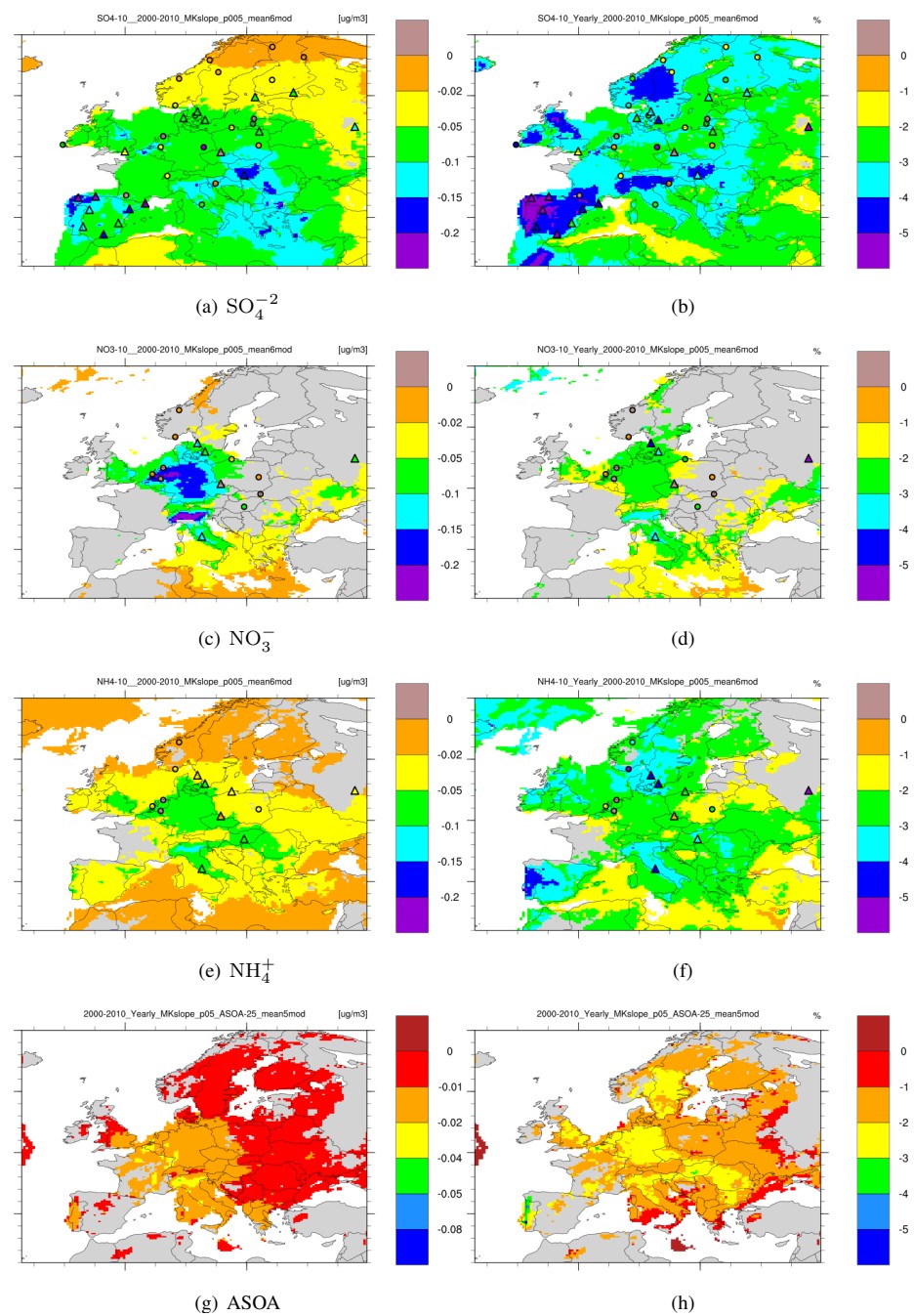

**Figure A9.** Mean observed and from 6-model ensemble Sen's trend slopes for 2000-2010 for anthropogenic aerosols $SO_4^{-2}$, $NO_3^-$ and $NH_4^+$ (a-f), and simulated with 5-model ensemble for ASOA (note different color scale). Left panels – absolute ($\mu$g m$^{-3}$ yr$^{-1}$) and right panels – relative (% yr$^{-1}$) trends. The modelled trends are shown as coloured contour map (grey or white means non-significant trends) and the observed trends as coloured triangles (significant) and circles (non-significant).

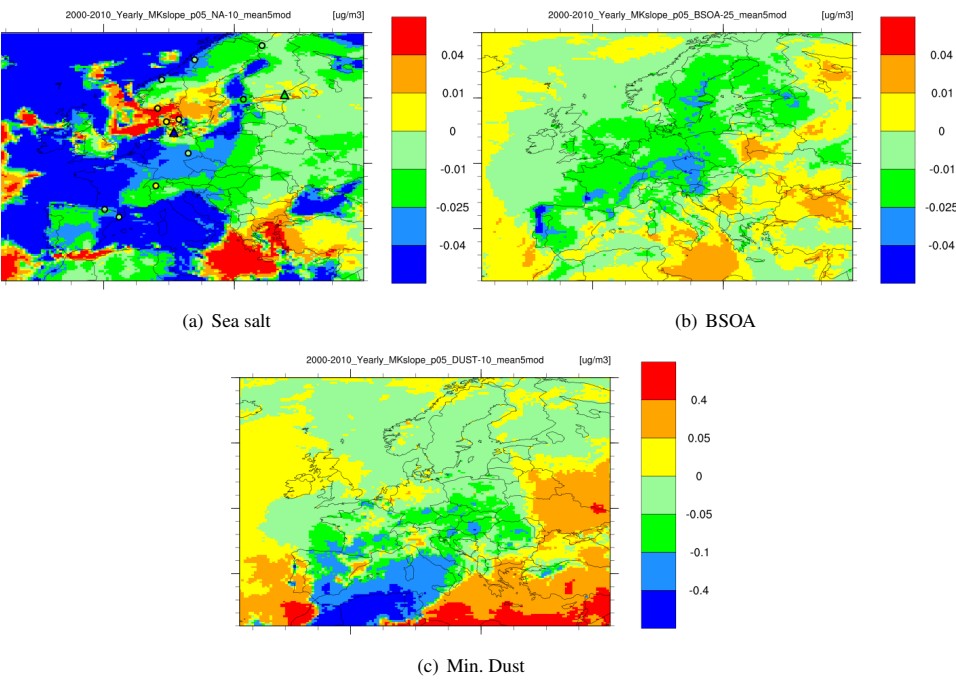

(a) Sea salt

(b) BSOA

(c) Min. Dust

**Figure A10.** Mean Sen's trend slopes for 2000-2010 simulated by the 6-model ensemble for natural aerosols: (a) sea salt ( observed trends also shown), (b) BSOA and (c) mineral dust. The modelled trends are shown as coloured contour map (grey or white means non-significant trends) and the observed trends as coloured triangles (significant) and circles (non-significant).

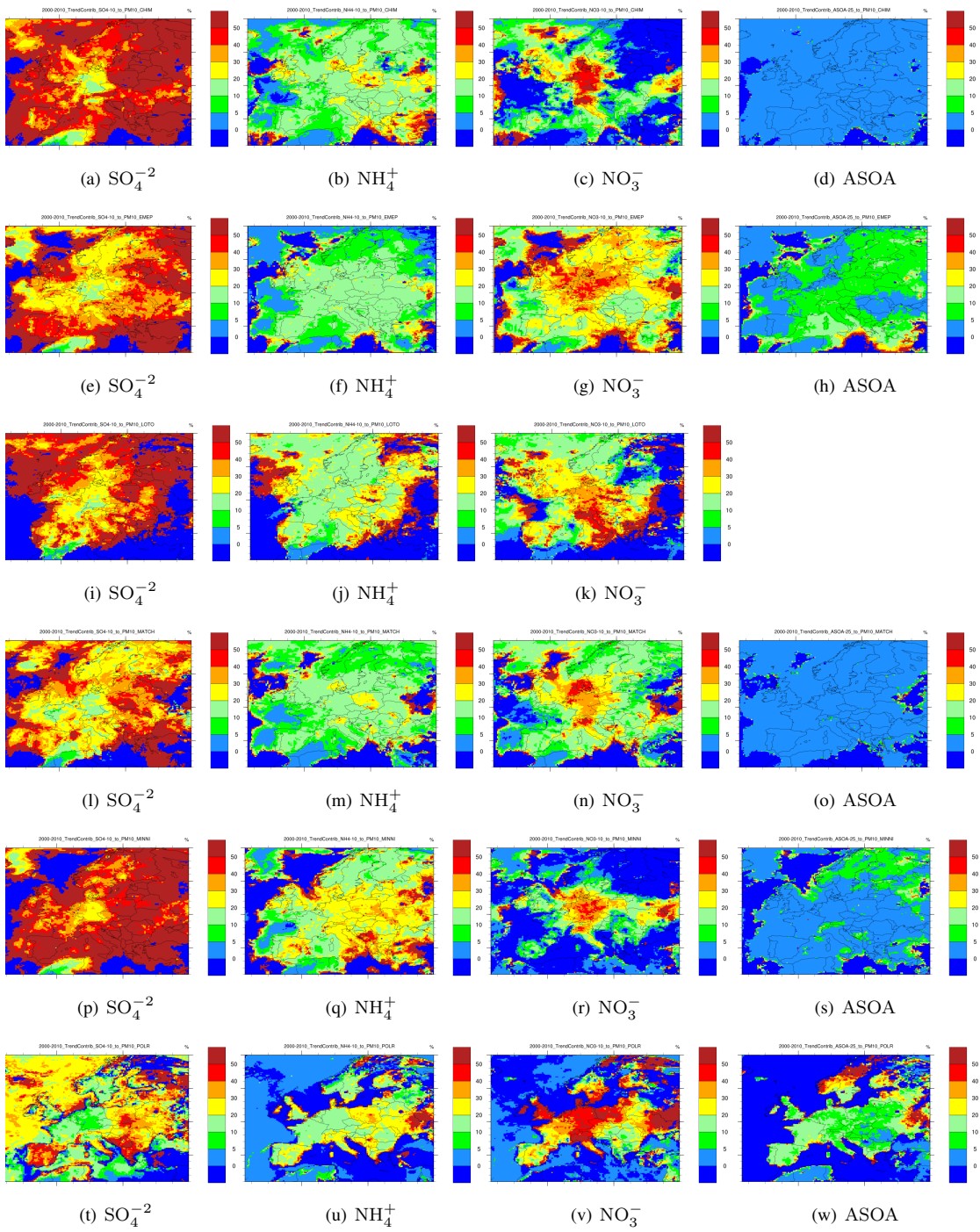

(a) $SO_4^{-2}$     (b) $NH_4^+$     (c) $NO_3^-$     (d) ASOA

(e) $SO_4^{-2}$     (f) $NH_4^+$     (g) $NO_3^-$     (h) ASOA

(i) $SO_4^{-2}$     (j) $NH_4^+$     (k) $NO_3^-$

(l) $SO_4^{-2}$     (m) $NH_4^+$     (n) $NO_3^-$     (o) ASOA

(p) $SO_4^{-2}$     (q) $NH_4^+$     (r) $NO_3^-$     (s) ASOA

(t) $SO_4^{-2}$     (u) $NH_4^+$     (v) $NO_3^-$     (w) ASOA

**Figure A11.** Relative contributions of (from left to right) $SO_4^{-2}$, $NH_4^+$, $NO_3^-$ and ASOA to $PM_{10}$ trends between 2000 and 2010 calculated by (from top to bottom) CHIMERE, EMEP, LOTOS-EUROS, MATCH, MINNI and Polair3D models.

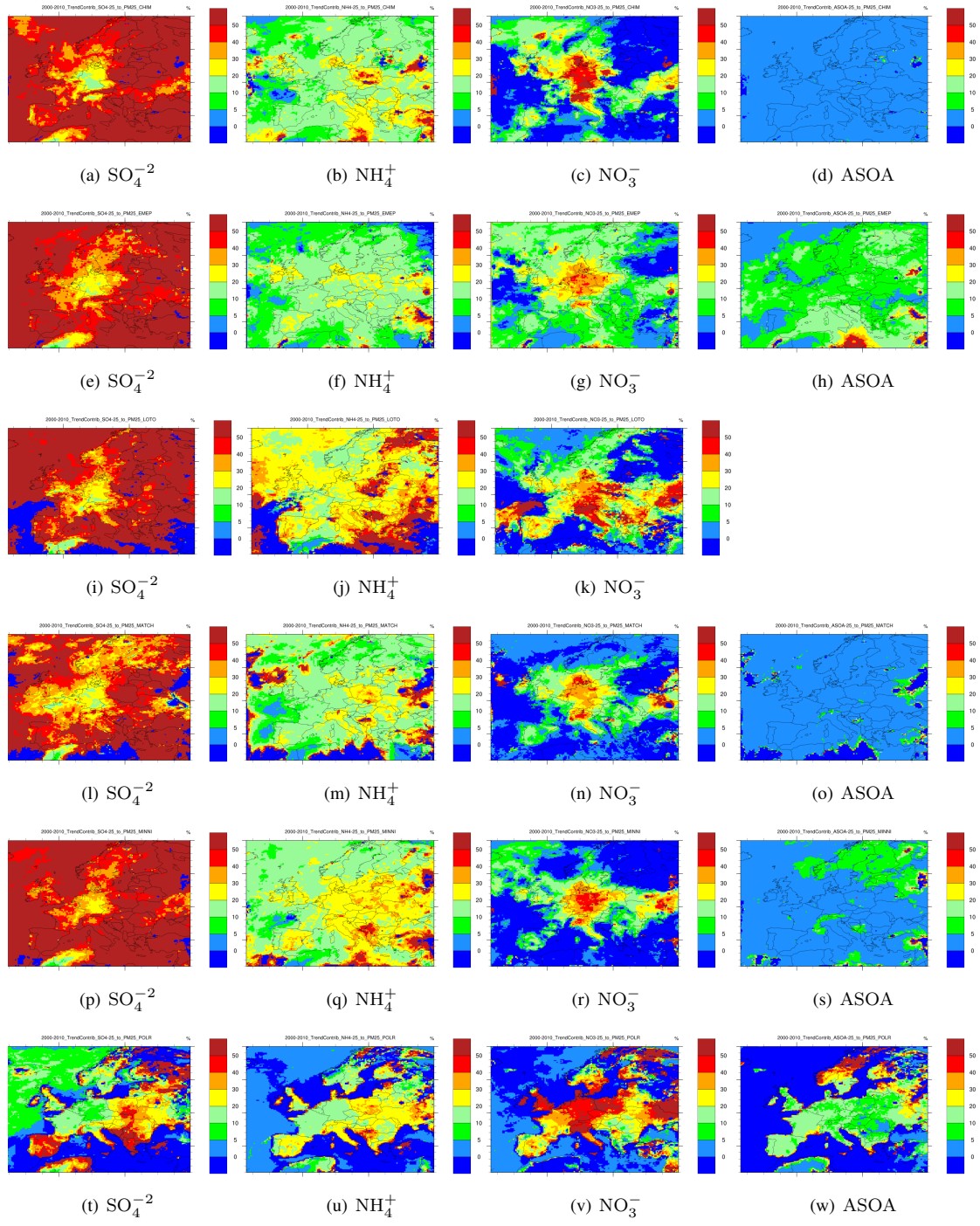

**Figure A12.** Relative contributions of (from left to right) $SO_4^{-2}$, $NH_4^+$, $NO_3^-$ and ASOA to $PM_{2.5}$ trends between 2000 and 2010 calculated by (from top to bottom) CHIMERE, EMEP, LOTOS-EUROS, MATCH, MINNI and Polair3D models.

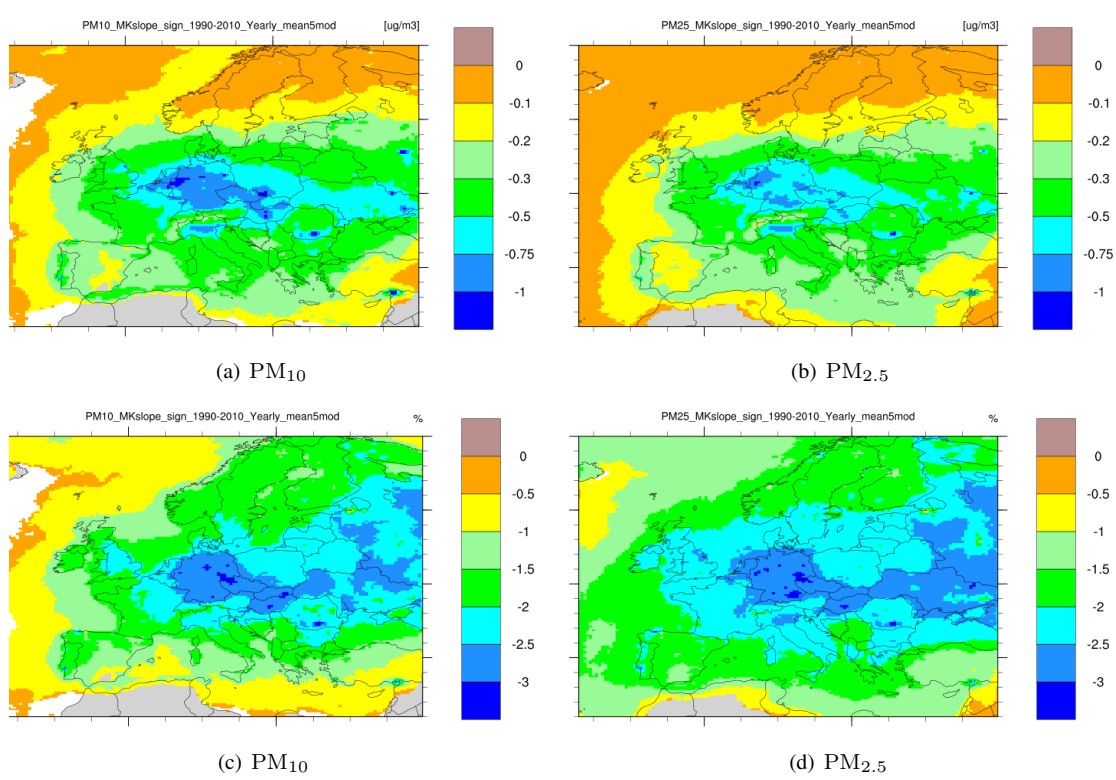

**Figure A13.** Annual mean Sen's slope for trends in the period 1990-2010 as calculated by the 6-model ensemble (left) for $PM_{10}$ and (right) $PM_{2.5}$. Upper panels – absolute ($\mu$g m$^{-3}$ yr$^{-1}$) and lower panels – relative to 1990 (% yr$^{-1}$).

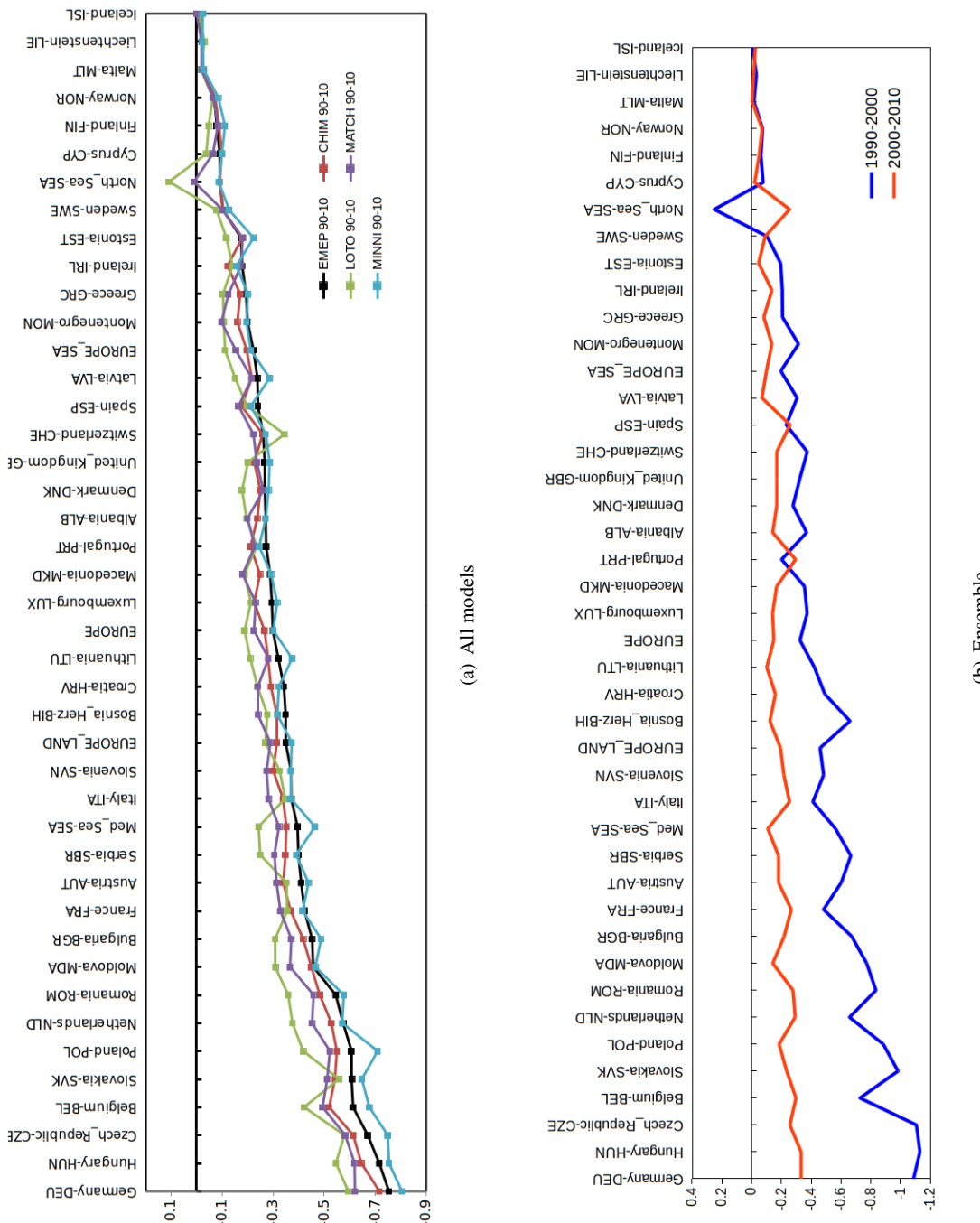

**Figure A14.** Modelled $PM_{10}$ trends calculated for European countries ($\mu$g m$^{-3}$ yr$^{-1}$): a. the individual models for the period 1990-2010, and b. the model ensemble for the periods 1990-2000 and 2000-2010 separately. The countries are ranged according to descending 1990-2010 negative trends from the EMEP model (a).

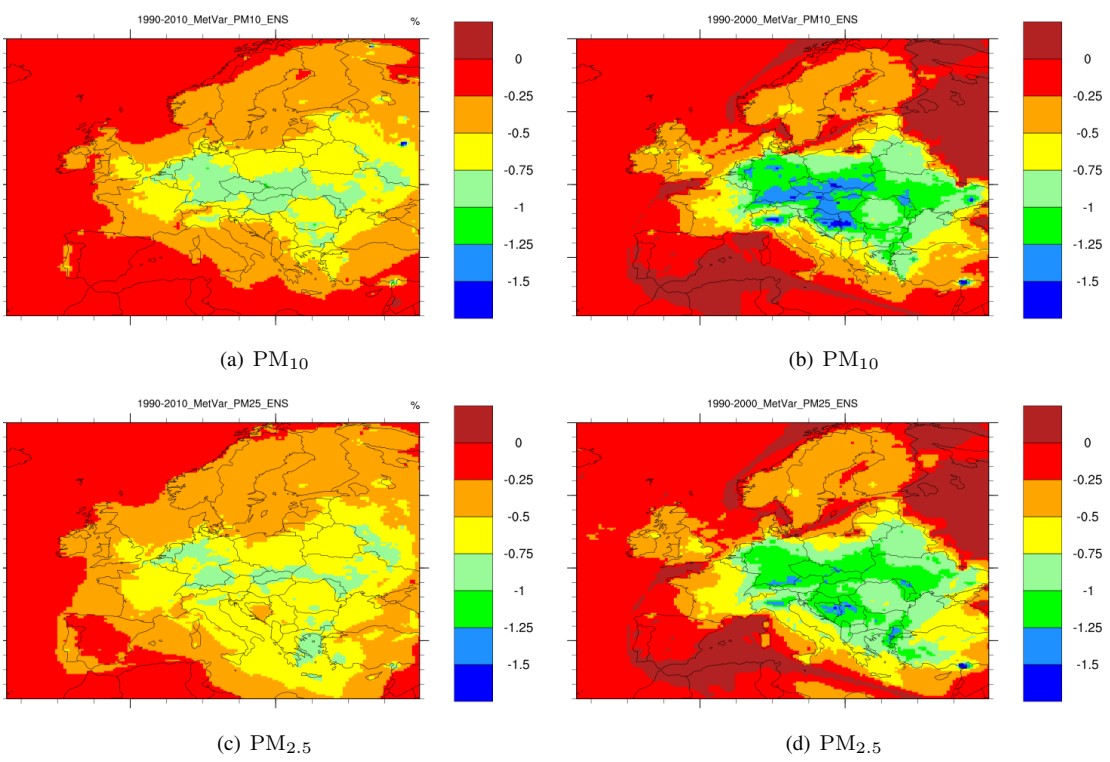

**Figure A15.** The ratio of PM changes due to emission changes to those due to inter-annual meteorological variability for $PM_{10}$ and $PM_{2.5}$ for 1990-2010 (left panels) and 1990-2000 (right panels) periods.

**Figure A16.** (a) Brief description of the Chemistry-Transport Models involved in the Eurodelta-Trends modeling exercise (extended version of Table S1 in Supplementary Material to Colette et al. (2017a).)

| MODEL | CHIMERE | EMEP | LOTOS-EUROS | MATCH | MINNI | POLYPHEMUS |
|---|---|---|---|---|---|---|
| version | Modified Chimere 2013 | rv4.7 | v1.10.005 | VSOA April 2016 | V4.7 | V1.9.1 |
| operator | INERIS | MET Norway | TNO | SMHI | ENEA/Arianet S.r.l. | CEREA |
| Chemistry/Meteorology coupling | offline | offline | offline | offline | offline | Offline |
| Vertical layers | 9 sigma | 20 sigma | 5(4 dynamic layers and a surface layer) | 39 hybrid eta utilizing the meteorological model layers | 16 fixed terrain following layers | 9 Fixed terrain following layers |
| Vertical extent | 500 hPa | 100 hPa | 5000 m | ca 5000 m (4700 – 6000 m) | 10000 m | 12000m |
| Advection scheme | (van Leer, 1984) | (Bott, 1989) | (Walcek, 2000) | Fourth order mass conserved advection scheme based on (Bott, 1989) | Blackman cubic polynomials (Yamartino, 1993) | Third-order Direct Space Time scheme (Spee, 1998) with Koren-Sweby flux limiter function |
| Vertical diffusion | Kz approach following (Troen and Mahrt, 1986) | Kz approach following (O'Brien, 1970) and (Jeričević et al., 2010) | Kz approach Yamartino et al (2004) | Implicit mass conservative Kz approach, see (Robertson et al., 1999) Boundary layer parameterisation as detailed in (Robertson et al., 1999) forms the basis for vertical diffusion and dry deposition | Kz approach following (Lange, 1989) | Kz approach following (Troen and Mahrt, 1986) |
| Depth first layer | 20 m | 90 m | 25 m | ca 60m | 40 m | 40m |
| Surface concentration | First model level | Downscaled to 3m using dry deposition velocity and similarity theory | Downscaled to 3m | Downscaled to 3m | First model level | First model level |

**Figure A16.** (b) Continuation of Main Features of CTMs involved in the EDT modelling.

| | | | | | | |
|---|---|---|---|---|---|---|
| **Biogenic VOC** | MEGAN model v2.1 with high resolution spatial and temporal LAI (Yuan et al., 2011) and recomputed emissions factors based on the land-use (Guenther et al., 2006) | Based upon maps of 115 species from (Koeble and Seufert, 2001), and hourly temperature and light using (Guenther et al., 1993; Guenther et al., 1994). See (Simpson et al., 1995; Simpson et al., 2012) | Based upon maps of 115 species from (Koeble and Seufert, 2001), and hourly temperature and light (Guenther et al 1991, Guenther et al 1993). See (Beltman et al., 2013) | (Simpson et al., 2012), based on hourly temperature and light. | MEGAN v2.04 (Guenther et al., 2006) | MEGAN V2.04 (Guenther et al., 2006) |
| **Forest fires** | None | None | None | None | None | None |
| **Soil-NO** | MEGAN model v2.04 | See in (Simpson et al., 2012) | Not used here | None | MEGAN v2.04 | MEGAN V2.04 |
| **Lightning** | None | Monthly climatological fields, (Köhler et al., 1995) | None | None | None | None |
| **Dust traffic suspension** | None | (Denier van der Gon et al., 2010) | None | Not used here | None | None |
| **Land-use database** | GLOBCOVER (24 classes) | CCE/SEI for Europe, elsewhere GLC2000 | Corine Land Cover 2000 (13 classes) | CCE/SEI for Europe | Corine Land Cover 2006 (22 classes) | Global Land Cover 2000 (24 classes) |
| **Ammonia compensation points** | None | None, but zero NH3 deposition over growing crops | Only for NH3 (for stomatal, external leaf surface and soil (= 0) | None | None | None |
| **Sea salt** | (Monahan, 1986) | (Monahan, 1986) and (Martensson et al., 2003), see (Tsyro et al., 2011) | (Martensson et al., 2003) and (Monahan, 1986), see (Schaap et al., 2009) | Based on parameterization by (Sofiev et al (2011) | (Zhang K.M. et al., 2005) | (Monahan, 1986) |
| **Windblown Dust** | No windblown dust emissions within domain; Boundary conditions from EMEP | Based on Marticorena and Bergametti (1995), Marticorena et al. (1997), Alfaro and Gomes (2001), Gomes et al. (2003), and Zender et al. (2003). Boundary conditions from EMEP | Based on Marticorena and Bergametti (1995), Gomes et al. (2003) and Alfaro et al. (2004). See Schaap et al. (2009) Boundary conditions from EMEP | No windblown dust emissions within domain; Boundary conditions from EMEP | Within domain based on Vautard et al. (2005); Boundary conditions from EMEP | No windblown dust emissions within domain; Boundary conditions from EMEP |

**Figure A16.** (c) Continuation of Main Features of CTMs involved in the EDT modelling.

| | | | | | | |
|---|---|---|---|---|---|---|
| **Dry deposition** | Resistance approach (Emberson et al., 2000a; Emberson et al., 2000b) | Resistance approach for gasses (Venkatram and Pleim, 1999) for aerosols, (Simpson et al., 2012) | Resistance approach, DEPAC3.11 for gasses, (Van Zanten et al., 2010) and (Zhang et al., 2001) for aerosols | Resistance approach depending on aerodynamic resistance, and land use (vegetation). Similar to (Andersson et al., 2007) | Resistance model based on (Wesely, 1989) | Resistance approach for gasses (Zhang et al., 2003) and aerosols (Zhang et al., 2001) |
| **Stomatal resistance** | (Emberson et al., 2000a; Emberson et al., 2000b) | DO3SE-EMEP: (Emberson et al., 2000a; Emberson et al., 2000b), Tuovinen et al., 2004; Simpson et al., 2012) | Emberson et al., (2000a); Emberson et al., (2000b) | Simple, seasonally varying, diurnal variation of surface resistance for gasses with stomatal resistance (similar to Andersson et al., 2007) | Wesely (1989) | Zhang et al. (2003) |
| **Wet deposition gasses** | In-cloud and sub-cloud scavenging coefficients | In-cloud and sub-cloud scavenging coefficients | sub-cloud scavenging coefficient | In-cloud scavenging of some species based on Henry's law constants. Simple in-cloud and sub-cloud scavenging coefficients for other gasses. | In-cloud and sub-cloud scavenging coefficients (EMEP, 2003) | In-cloud (monodispersed raindrops with constant collection efficiency) and bellow cloud (Sportisse and Dubois, 2002) scavenging coefficients |
| **Wet deposition particles** | In-cloud and sub-cloud scavenging | In-cloud and sub-cloud scavenging | sub-cloud scavenging coefficient | In-cloud and sub-cloud scavenging. Similar to (Simpson et al., 2012) | In-cloud and sub-cloud scavenging coefficients | In-cloud (as for gas) and bellow cloud (Slinn, 1983) scavenging coefficients |
| **Gas phase chemistry** | MELCHIOR2 | EmChem09 (Simpson et al., 2012) | TNO-CBM-IV | Based on EMEP (Simpson et al., 2012), with modified isoprene chemistry (Carter, 1996;Langner et al., 1998) | SAPRC99 (Carter, 1996; Carter, 2000) | CB-05 (Yarwood. G. et al., 2005) |
| **Speciation of anthropogenic PM emissions** | No split applied, PPM for all primary anthropogenic PM | EMEP split of $PM_{2.5}$ and coarse PM emissions to EC, POA, rest PPM per country/SNAP sector using fractions provided by IIASA (see Simpson at al. 2012) | No split applied, PPM for all primary anthropogenic PM | Split of $PM_{2.5}$ emissions to EC, POA, rest PPM per country/SNAP sector sing fractions by Kuenen et al. (2014), Andersson et al. (2015). No split of coarse PM emissions | $PM_{2.5}$ are split to EC, POA and rest PPM based on Based on Kuenen et al. (2014) per country and SNAP sectors. No split for coarse PM. | $PM_{2.5}$ are split between EC (~ 20%), POA (~45%) and rest PPM (~35%). No split for coarse PM. Detailed speciation are applied per SNAP sector. Based on Kuenen et al. (2014) |

**Figure A16.** (d) Continuation of Main Features of CTMs involved in the EDT modelling.

| | | | | | | |
|---|---|---|---|---|---|---|
| **Cloud chemistry** | Aqueous $SO_2$ chemistry to form $SO_4^{2-}$ by $H_2O_2$, $O_3$ and $O_2$; pH dependent | Aqueous $SO_2$ chemistry to form $SO_4^{2-}$ by $H_2O_2$, $O_3$ and $O_2$; pH dependent (Simpson et al., 2012). ($N_2O_5$ hydrolysis to form nitrate) | Aqueous $SO_2$ chemistry, particle formation, pH dependent (Banzhaf et al., 2012). | Aqueous $SO_2$ chemistry to form $SO_4^{2-}$ by $H_2O_2$, $O_3$ and $O_2$ | Aqueous $SO_2$ chemistry to form $SO_4^{2-}$ by $H_2O_2$, $O_3$ and $O_2$ (Seinfeld and Pandis, 1998) | Aqueous $SO_2$ chemistry to form $SO_4^{2-}$ by $H_2O_2$, $O_3$ and $O_2$ (Seinfeld and Pandis, 1998) |
| **Ammonium nitrate equilibrium** | ISORROPIA v2.1 (Nenes et al., 1999) | MARS (Binkowski and Shankar, 1995) | ISORROPIA v.2 | RH & T dependent equilibrium constant (Mozurkewich, 1993) | ISORROPIA v1.7 (Nenes et al., 1998) | ISORROPIA v1.7 (Nenes et al., 1999) |
| **Coarse nitrate** | $HNO_3$ condensation on coarse sea salts computed as a function of pH, condensation kinetics and accounting for HCl evaporation. Local equilibrium using ISORROPIA in reverse mode. | Two formation rates of coarse $NO_3$ from $HNO_3$ for relative humidity below/above 90% | Dynamic formation on Na (Wichink Kruit et al., 2012) | Transfer of $HNO_3$(g) to aerosol nitrate using rate from (Strand and Hov, 1994) | None | No heterogeneous nitrate formation |
| **SOA formation** | $H_2O$ (Couvidat et al., 2012) mechanism coupled with the thermodynamic model SOAP (Couvidat and Sartelet, 2015) | VBS-NPAS (Simpson et al., 2012; Bergström et al., 2012) | Not used here | Similar to VBS-NPNA (Bergström et al., 2012) | SORGAM module (Schell et al., 2001) | $H_2O$ (Couvidat et al., 2012) |
| **Aerosol model** | 9 bins (10 nm to 10 µm) | Bulk- approach (fine and coarse modes) | Bulk- approach (2 modes) | Bulk approach | 3 modes as in AERO3 (Binkowski, 1999); | 5 bins (0.01 to 10µm) |
| **Aerosol physics** | Coagulation/condensation/nucleation. Computation of the wet diameter for each bins as a function of humidity (used for coagulation, condensation, deposition) | Not used here | Not used here | Not used here | Coagulation/ condensation/nucleation | Coagulation/ Condensation |

*) EC - elemental carbon, POA - primary organic aerosol, ASOA and BSOA - anthropogenic and biogenic secondary aerosol, TPPM - total primary PM (EC+POA+rest PPM)

*Author contributions.* ACo coordinated the EURODELTA-Trends (EDT) exercise and WA was responsible for the compilation and quality control of the observations. The following modelling teams set-up, pre-processed, ran and post-processed the simulations for each model: FC, BB, MGV and ACo for CHIM; ST and PW for EMEP; AM and MS for LOTO; CA and RB for MATCH; MM, MA, GB, ACa and MD for MINNI; YR and VR for POLR. WA and CK contributed with production of figures. Additional post-processing of model output and uploading to the AeroCom server was done by KC. All of the analyses presented in this paper were carried out by ST, assisted by discussions with WA and HF and with valuable contributions from ACo, CA, AM, GC, YR, BB, MT, MGV, NO, KM, FC, and MTP. Special thanks for language proofreading go to MT.

*Competing interests.* No competing interest are present.

*Acknowledgements.* The Ineris coordination of the EURODELTA-Trends exercise was supported by the French Ministry in charge of Ecology in the context of the Task Force on Measurement and Modelling of the EMEP program of the LRTAP Convention. Meteorological forcing with the WRF model was provided by Robert Vautard and Annemiek Stegehuis from LSCE/IPSL. The CHIMERE simulations where performed using the TGCC supercomputers under GENCI computing allocation. The participation of CIEMAT was financed by the Spanish Ministry of Agriculture and Fishing, Food and Environment.

Computer time for EMEP model runs was supported by the Research Council of Norway through the NOTUR project EMEP (NN2890K) for CPU and the NorStore project European Monitoring and Evaluation Programme (NS9005K) for storage of data.

Funding for the MATCH participation was jointly divided between Nordforsk through the research programme Nordic Welfair (grant no 75007), the Swedish Environmental Protection Agency through the SCAC research programme and the 2017-2018 Belmont Forum and BiodivERsA joint call for research proposals, under the BiodivScen ERA-Net COFUND programme, with the funding organisations AKA (contract no 326328), ANR (ANR-18-EBI4-007), BMBF (KFZ: 01LC1810A), FORMAS (contract no's: 2018-02434, 2018-02436, 2018-02437, 2018-02438) and MICINN (APCIN; PCI2018-093149).

LOTOS-team thanks Erik van Meijgaard of the Royal Netherlands Meteorological Institute (KNMI) for providing the RACMO2 simulations that were used by LOTOS-EUROS.

The computing resources and the related technical support used for MINNI simulations have been provided by CRESCO/ENEAGRID High Performance Computing infrastructure and its staff. The infrastructure is funded by ENEA, the Italian National Agency for New Technologies, Energy and Sustainable Economic Development and by Italian and European research programmes (http://www.cresco.enea. it/english, last access: 21 December 2018). MINNI participation to this project was supported by the "Cooperation Agreement for support to international Conventions, Protocols and related negotiations on air pollution issues", funded by the Italian Ministry for the Environment, Land and Sea.

CIEMAT acknowledges the Ministry for the Ecological Transition and Demographic Challenge (MITERD) for financial support.

Giancarlo Ciarelli was supported by ADEME and the Swiss National Science Foundation (grant no. P2EZP2_175166).

The GAINS emission trends were produced as part of the FP7 European Research Project ECLIPSE (Evaluating the Climate and Air Quality Impacts of Short-Lived Pollutants) grant no. 282688.

**Table A1.** Selected set of EMEP monitoring stations for $PM_{10}$ and $PM_{2.5}$ trend analysis for the period 2000-2010

| Sites Code | Name | latitude | longitude | altitude | $PM_{10}$ measurements Sampler | Frequency | Missing years | $PM_{2.5}$ measurements Sampler | Frequency | Missing years |
|---|---|---|---|---|---|---|---|---|---|---|
| AT0002R | Ilmitz | 47.767 | 16.767 | 117.0m | high vol | daily | | high vol | daily | 2000 |
| AT0005R | Vorhegg | 46.678 | 12.972 | 1020.0m | high vol | daily | 2000 | high vol | daily | 11 |
| CH0002R | Payerne | 46.813 | 6.945 | 489.0m | high vol | daily | | | | |
| CH0003R | Tänikon | 47.480 | 8.905 | 539.0m | high vol | daily | | | | |
| CH0004R | Chaumont | 47.050 | 6.979 | 1137.0m | high vol | daily | | | | |
| CH0005R | Rigi | 47.068 | 8.464 | 1031.0m | high vol | daily | | | | |
| CZ0003R | Kosetice (NOAK) | 49.573 | 15.080 | 535.0m | b-attenuation[1] | hourly | 2000 | | | |
| DE0001R | Westerland | 54.926 | 8.310 | 12.0m | high vol | daily | | high vol | daily | |
| DE0002R | Waldhof | 52.802 | 10.759 | 74.0m | high vol | daily | | high vol | daily | 2000 |
| DE0003R | Schauinsland | 47.915 | 7.909 | 1205.0m | high vol | daily | | | | |
| DE0007R | Neuglobsow | 53.167 | 13.033 | 62.0m | high vol | daily | 11 | | | |
| DE0008R | Schmücke | 50.650 | 10.767 | 937.0m | high vol | daily | | | | |
| DE0009R | Zingst | 54.437 | 12.725 | 1.0m | high vol | daily | | | | |
| DE0044R | Melpitz | 51.530 | 12.934 | 86.0m | high vol | daily | | | | |
| ES0007R | Viznar | 37.233 | -3.533 | 1265.0m | high vol | daily | 2000 | high vol | daily | 2000 |
| ES0008R | Niembro | 43.442 | -4.850 | 134.0m | high vol | daily | 2000 | | | |
| ES0011R | Barcarrota | 38.476 | -6.923 | 393.0m | high vol | daily | 2000 | high vol | daily | 2000 |
| ES0012R | Zarra | 39.086 | -1.102 | 885.0m | high vol | daily | 2000 | high vol | daily | 2000 |
| ES0013R | Penausende | 41.283 | -5.867 | 985.0m | high vol | daily | 2000 | | | |
| ES0014R | Els Torms | 41.400 | 0.717 | 470.0m | high vol | daily | 2000 | high vol | daily | 2000 |
| FI0050R | Hyytiälä | 61.850 | 24.283 | 181.0m | low vol[3] | daily | | low vol[3] | daily | |
| GB0006R | Lough Navar | 54.443 | -7.870 | 126.0m | TEOM FDMS | daily | | | | |
| GB0036R | Harwell | 51.573 | -1.317 | 137.0m | TEOM FDMS | daily | | TEOM FDMS | hourly | |
| GB0043R | Narberth | 51.782 | -4.691 | 160.0m | TEOM FDMS | daily | 2003, 2004 | | | |
| IT0004R | Ispra | 45.800 | 8.633 | 209.0m | low vol | weekly | 11 | low vol | daily | 2000 |
| NO0002R | Birkenes II | 58.389 | 8.252 | 219.0m | low vol | weekly | | low vol | weekly | 2000 |
| SE0012R | Aspvreten | 58.800 | 17.383 | 20.0m | b-attenuation[2] | daily | 2009 | TEOM | hourly | 2010 |

[1] Thermo FH 62 I-R, [2] OPSIS SM200, [3] with 4 stage cascade impactor

**Table A2.** Relative bias (in %) and correlation (R) for modelled $PM_{10}$ with respect to available observations at 26 EDT sites for the years 2000 to 2010

| Year | | CHIM | | EMEP | | LOTO | | MATCH | | MINNI | | POLR[*] | |
|---|---|---|---|---|---|---|---|---|---|---|---|---|---|
| | Nsite | Bias | R | Bias | R | Bias | R | Bias | R | Bias | R | Bias | R |
| 2000 | 15 | -2.0 | 0.64 | 2.1 | 0.58 | -6.3 | 0.47 | -11 | 0.60 | -4.3 | 0.58 | -24 | 0.62 |
| 2001 | 21 | -4.4 | 0.41 | -8.2 | 0.61 | -4.0 | 0.60 | -9.4 | 0.71 | -8.3 | 0.48 | -30 | 0.59 |
| 2002 | 22 | -7.5 | 0.55 | -14 | 0.60 | -13 | 0.46 | -16 | 0.63 | -14 | 0.61 | -35 | 0.60 |
| 2003 | 22 | -8.5 | 0.59 | -13 | 0.63 | -16 | 0.40 | -12 | 0.55 | -12 | 0.63 | -36 | 0.64 |
| 2004 | 22 | -7.3 | 0.50 | -14 | 0.64 | -9.7 | 0.66 | -14 | 0.78 | -8.9 | 0.60 | -33 | 0.64 |
| 2005 | 23 | -7.2 | 0.55 | -11 | 0.63 | -5.3 | 0.51 | -12 | 0.65 | -7.0 | 0.62 | -31 | 0.65 |
| 2006 | 21 | -6.3 | 0.48 | -12 | 0.48 | 4.3 | 0.24 | -7.6 | 0.34 | -10 | 0.52 | -32 | 0.42 |
| 2007 | 21 | -2.6 | 0.39 | -10 | 0.50 | -11 | 0.43 | -9.8 | 0.61 | -6.7 | 0.48 | -28 | 0.50 |
| 2008 | 21 | -3.7 | 0.37 | -10 | 0.49 | -8.23 | 0.44 | -11 | 0.63 | -7.5 | 0.48 | -28 | 0.53 |
| 2009 | 22 | -0.4 | 0.53 | -8.0 | 0.61 | 2.4 | 0.39 | -5.1 | 0.50 | -3.5 | 0.59 | -27 | 0.57 |
| 2010 | 21 | -9.4 | 0.58 | -16 | 0.62 | -15 | 0.23 | -18 | 0.46 | -17 | 0.60 | -35 | 0.53 |

Bias - relative bias expressed in %;

[*] Excluding coarse sea salt

**Table A3.** Relative bias (in %) and correlation (R) for modelled $PM_{2.5}$ with respect to available observations at 13 EDT sites for the years 2000 to 2010

| Year | | CHIM | | EMEP | | LOTO | | MATCH | | MINNI | | POLR | |
|---|---|---|---|---|---|---|---|---|---|---|---|---|---|
| | Nsite | Bias | R | Bias | R | Bias | R | Bias | R | Bias | R | Bias | R |
| 2000 | 5 | -2.4 | 0.71 | -5.3 | 0.63 | -6.4 | 0.69 | -3.1 | 0.64 | 6.1 | 0.71 | -1.7 | 0.69 |
| 2001 | 12 | -18 | 0.59 | -21 | 0.61 | -18 | 0.79 | -18 | 0.70 | -10 | 0.53 | -22 | 0.73 |
| 2002 | 12 | -19 | 0.64 | -25 | 0.68 | -18 | 0.73 | -21 | 0.70 | -12 | 0.60 | -26 | 0.72 |
| 2003 | 12 | -18 | 0.74 | -20 | 0.72 | -25 | 0.70 | -16 | 0.69 | -6.3 | 0.70 | -25 | 0.74 |
| 2004 | 12 | -17 | 0.61 | -23 | 0.62 | -15 | 0.77 | -16 | 0.73 | -6.2 | 0.54 | -23 | 0.72 |
| 2005 | 13 | -21 | 0.68 | -24 | 0.63 | -15 | 0.73 | -17 | 0.69 | -8.7 | 0.59 | -22 | 0.74 |
| 2006 | 11 | -20 | 0.69 | -24 | 0.50 | -12 | 0.65 | -16 | 0.55 | -9.1 | 0.45 | -25 | 0.61 |
| 2007 | 11 | -18 | 0.59 | -25 | 0.52 | -18 | 0.63 | -16 | 0.73 | -8.7 | 0.40 | -22 | 0.69 |
| 2008 | 11 | -11 | 0.60 | -17 | 0.60 | -6.6 | 0.71 | -6.3 | 0.69 | 1.1 | 0.54 | -12 | 0.71 |
| 2009 | 11 | -7.3 | 0.65 | -14 | 0.61 | 0.9 | 0.76 | -3.8 | 0.64 | 6.1 | 0.57 | -12 | 0.67 |
| 2010 | 10 | -17 | 0.62 | -24 | 0.58 | -14 | 0.68 | -14 | 0.66 | -10 | 0.64 | -19 | 0.67 |

Bias - relative bias expressed in %

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
