# Peer review of "Europe in the period of 1990-2010"

_Atmospheric Chemistry and Physics, 2021_

## Author Response (AR1)

Dear Editor(s),

We have uploaded Authors' Comments to referees' comments to our paper acp-2021-970 "Eurodelta multi-model simulated and observed PM trends in Europe in the period of 1990–2010" (Tsyro et al.). In addition, DIFF_PMtrends_acp-2021-970.pdf, showing the changes in the revised manuscript, is also uploaded in order to facilitate tracking our updates and editing.

We would like to notify you that an inconsistency in concentration units was spotted while addressing a comment by Referee #1 (see the comments to Lines 399 – 400), which consequently has been corrected and small changes to the text have been made. The mentioned unit inconsistency did not affect any major conclusions.

Unfortunately, it did not work to check the figures in the manuscript the Coblis – Color Blindness Simulator as it is not designed for pdf-files.

Best regards,

Dr. Svetlana Tsyro
Norwegian Meteorological Institute

**Eurodelta multi-model simulated and observed PM trends in Europe in the period of 1990–2010**

Svetlana Tsyro,Wenche Aas,Augustin Colette,Camilla Andersson,Bertrand Bessagnet,Giancarlo Ciarelli,Florian Couvidat,Kees Cuvelier,Astrid Manders,Kathleen Mar,Mihaela Mircea,Noelia Otero,Maria-Teresa Pay,Valentin Raffort,Yelva Roustan,Mark R. Theobald,Marta G. Vivanco,Hilde Fagerli,Peter Wind,Gino Briganti,Andrea Cappelletti,Massimo D'Isidoro,and Mario Adani

**Author comments to Referee #1**

**Overarching comments:**

The manuscript provides a comprehensive evaluation of modeled PM mass and species trends against observations for the 2000 – 2010 time period, with a more limited analysis of trends over the 1990 – 2010 time period. Evaluating whether models can capture historic trends resulting from changes in emissions is of key importance when using such models to predict future changes. The model simulations and analyses presented in the manuscript are sound and of good quality. The manuscript is well written and structured clearly, the presentation quality of all figures and tables is good, and references to other published studies on similar topics are provided where appropriate.

*Our reply: We are grateful to the referee for valuable questions and comments, as well as suggestions for editing the text. We have tried to address all issues raised, which we hope has helped to improve the quality of our paper. Our answers to the referee's comments can be found below, in Cursive text format.*

My only major comment is that the impacts of not considering forest fire emissions in these simulations should be discussed more prominently in the description of results as well as the abstract and summary. In addition, the list below contains a number of minor comments that the authors may want to consider when revising their manuscript.

*Our reply: We thank the referee for this valuable comment. Forest fires were not included in this work partly because of the uncertainties in emissions and modelling of those, but also because we aimed to look at PM trends due to emission regulation in Europe. As we discuss below, there are no model indications that the pollution from the major forest fires had any strong effect on the annual mean PM at the EDT sites. Therefore we do not think that excluding forest fire emissions from the consideration in EDT analysis had any significant consequences on models vs observations comparison.*
*Though an in-depth analysis of the effects of forest fires on PM trends was not in the scope of this work, it is indeed important to discuss the possible consequences of omitting forest fire emissions in the trend modelling. Since no EDT model ensemble simulations are available in order to address this issue, we made use of trend simulations with the EMEP model which included FINN (Fire INventory from NCAR) emissions. We looked at the effect of primary PM (which were available as individual tracers) from fires on PM10 and PM2.5*

*trends and also at the enhancement in secondary organic aerosol due to exceptionally large 2010 Russian forest fires and other major fires and included a short discussion based on the main findings (see further replies below).*

*EMEP trend simulations show only minor effects from primary PM emissions from forest fire to PM2.5 trends in Russia and Eastern Europe and negligible ones in other parts of Europe (Fig. R1). The EMEP model simulates enhanced concentrations of secondary organic aerosols (SOA) associated with the Russian fires (due to the enhanced amount of pre-existing organics and more efficient VOC oxidation by ozone), but most of the pollution from those fires mostly stayed over European Russia due to particular air circulation in the stagnant high-pressure system. Witte, J. C., et al. "NASA A-Train and Terra observations of the 2010 Russian wildfires." Atmospheric Chemistry and Physics 11.17 (2011) found that elevated levels of aerosols detected by satellites over western Russia corresponded to the time period of peak wildfire activity (22 July–18 August 2010) and a persistent anticyclonic circulation. In this situation air was trapped, allowing smoke pollutants to accumulate as the air mass re-circulated. Also Konovalov et al (2011) wrote that the accumulation of anthropogenic pollution over European Russia was favoured by a stagnant and dry meteorological situation (Konovalov, I. B., et al. "Atmospheric impacts of the 2010 Russian wildfires: integrating modelling and measurements of an extreme air pollution episode in the Moscow region." Atmospheric Chemistry and Physics 11.19 (2011): 10031-10056).*

*A discussion on this is included in Section 7.2 "Uncertainties in emissions" (see PMtrends_diff.pdf). The issue is mentioned in Summary ("Not accounting for forest fires in EDT simulations should also affect the accuracy of simulated PM trends, at least in the regions of large fires, whilst this does not appear to have a major impact on the modelled trends at the EDT sites") and in Abstract: "Further discussions are given with respect to emission uncertainties, implications of not accounting for forest fires…"*

[Figure]

Fig R1. EMEP modelled trends for PM2.5 including forest fires emissions (left) and the difference in PM2.5 trends when PM emissions from forest fires are included and excluded (right) (shown are both significant and insignificant trends)

**Specific comments:**

Line 25: maybe mention that the exclusion of forest fire emissions and (in some models) windblown dust emissions may have contributed to a higher fraction of modeled trends being significant due to the lower amount of interannual variability being present in such model simulations

***Our reply:*** *The referee is right that excluding forest fire emissions in EDT simulations and not accounting for windblown dust by some of the models (CHIM, MATCH and POLR only included dust from Boundary conditions) may have smoothed PM annual variations, which could have resulted in a larger fraction of modelled significant trends in the areas affected by those emissions.*

*Regarding forest fires, most of the EDT trend-sites were not significantly influenced by those emissions (see our comments above) so that model evaluation against observed trends in this respect was not feasible. Effects of mineral dust on PM trends, namely stronger PM modelled trends and larger fraction of significant ones compared to observations, should be seen more obviously for Spanish sites influenced by mineral dust.  However, Figure 6 shows that the model ensemble trends of PM10 and PM2.5 are not significant for all Spanish sites, whereas observations indicate that there were significant trends at 4 out of 6 Spanish sites for PM10 and 3 out of 4 sites for PM2.5. Further, we could expect to see weaker and more of not significant trends simulated by EMEP, LOTO and MINNI, which accounted for both African dust (from boundary conditions) and windblown dust within the simulation domain, still the results are rather mixed (Figure A8). For example, both EMEP and MINNI include windblown dust; while EMEP simulates significant PM10 and PM2.5 trends for the most number of sites compared with the other models, MINNI doesn't recognise any significant trend at any of the Spanish sites.*

Line 61: suggest inserting "over the" before "last years"

***Our reply:*** *The suggestion is followed.*

Lines 80 – 83: as noted below, to more fully accomplish this goal, it would be beneficial to provide more information on the speciation of primary PM2.5 emissions (especially EC and OC) and the representation of secondary organic aerosols in the models, and then reference this information when discussing speciated results in section 5.4

***Our reply:*** *We agree. The details regarding PM2.5 emissions speciation to elemental and organic carbon, as well as SOA representation in the models are included in the new Table A16: Main features of the Chemistry-Transport Models involved in the Eurodelta-Trends modeling exercise.*

Line 96: insert "that" before "participated"

***Our reply:*** *Done*

Line 109: insert "was" before "performed"

***Our reply:*** *Done*

Lines 105 – 116: Please provide information how the ECLIPSE PM2.5 and PM10 emissions were speciated into different compounds. If this was handled differently for each model,

please provide a summary of the approach for each model to better understand the results presented in section 5.4

*Our reply: The models were not required to use the same split of the Eclipse PM2.5 and PM10 emissions. The details regarding PM emissions speciation to elemental and organic carbon is included in the new Table A16: Main features of the Chemistry-Transport Models involved in the Eurodelta-Trends modeling exercise, and are referenced to in the discussion in Section 5.4.*

Lines 113-114: Please provide more information on the temporal resolution the ECLIPSE inventories for the different sectors – are these all annual total emissions, and did each model then apply the same EMEP profiles to perform monthly, weekly, and diurnal allocation? Did the vertical distribution of emissions depend on the meteorology used by each model to account for the effects of meteorology on plume rise, or are these EMEP profiles static? Given the effects of atmospheric stability on near-source PM concentrations, uncertainties in representing plume rise may be an important factor of model error especially during wintertime.

*Our reply: For temporal distribution of ECLIPSE annual emissions, the models applied the same monthly and hourly profiles prepared by TNO (Kuenen et al., 2014); also the same static emission vertical profiles (per sector) provided by INERIS were used. This information is included in section 2.1 (Models, runs setup) and Table A16.*

Lines 126-127: The exclusion of forest fire emissions (and in some models dust emissions) may have important implications for the interpretation of modeled and observed trends, specifically the smaller amount of interannual variability the exclusion of such emissions causes in the models which in turn leads to a tendency for more significance in the trends estimated from the models. The current version of the manuscript does not reference the exclusion of fire emissions in any of the discussions in sections 4 – 7.

*Our reply: We thank the referee for this valuable comment. A discussion regarding possible implications of omitting forest fires on modelled trends has been included in Section 7.2 Uncertainties in emissions: "Finally, as described in ~\ref{sec:models}, pollution from forest fires were not accounted for in EDT simulations mainly because of considerable uncertainties in forest fire emissions and modelling of those, but also because we aimed to look at PM trends due to emission regulation in Europe. An in-depth analysis of the effect of forest fires on PM trends is beyond the scope of the paper, but we have tested whether the discrepancies between the modelled and observed trends, in particular in terms of a relatively larger fraction of significant trends from the model results, could be due to not including forest fire emissions in the EDT simulations. Additional simulations suggest that the effects from even large fires during the studied period (like 2010 Russian forest fires) were mostly negligible outside the regions where wildfires occurred. In fact, the pollution from major forest fires did not seem to have any large impact on simulated annual mean PM at the EDT sites in the 2000-2010 period. Therefore we believe that not accounting for forest fires in EDT analysis did not have any significant consequences for models vs observations comparison." References to this discussion have been added where relevant.*

Section 2.1: It would be useful if this section also included a summary of the aerosol treatment of the different models (e.g. number of modeled species, sectional vs. modal size distribution representation, representation of biogenic and anthropogenic secondary organic aerosols, etc.) to help with the interpretation of results from individual models in subsequent sections.

*Our reply: This definitely useful information has now been included in the new Table A16: Main features of the Chemistry-Transport Models involved in the Eurodelta-Trends modeling exercise.*

Lines 136 – 146: Please clarify if the modeled annual mean values at the observation locations were computed by discarding any time periods for which observations were not available at that site. In other words, if a given site had 80% data completeness in a given year, was the corresponding model mean for that site and year computed over the same 80% of all modeled time periods in that year? Such temporal matching of observations and model values at the underlying temporal resolution and completeness of the observations would be the preferred approach for ensuring consistency between the observed and modelled annual means.

*Our reply: Very relevant question. Annual mean concentrations were calculated individually for observations and model results. We recognise that including only common days would give a more accurate   comparison between the models and observations, but we believe that the uncertainties associated with the adopted approach are in general small enough and that most of the results from the presented trend analysis are valid.*

Lines 175 – 176: If possible, it might be interesting to discuss which meteorological and/or emission features (e.g. precipitation, dust or fire emissions) may have caused the elevated PM levels in 2003, 2006, and 2010

*Our reply: Though an in-depth investigation of the causes of PM inter-annual variations is beyond the scope of the paper, we made a closer look at the timeseries at the individual sites.  It appears that that the enhanced PM in 2003, 2006 and 2010 were mostly due to high annual PM at Central-European sites (primarily in Germany, Austria, Switzerland), rather far away from major forest fires.Those elevated annual PM were due to heat waves in summers 2003 and 2006 (no PM removal by precipitation in dry weather conditions, efficient BSOA formation) and extensive PM episodes in winter 2010. Furthermore, based on the results from independent tests with the EMEP model, we do not believe that PM increases in 2003, 2006 and 2010 were due to forest fires as the trend sites are situated in the locations not (or negligibly) affected by the fire emissions. The year 2010 was already discussed in Section 5.1 (lines 265-273), a short discussion was added concerning 2003 and 2006.*

Line 226: consider changing "does  not probably indicate" to "hardly indicates"

Line 254: change LOTOS to LOTO for consistency

Line 261: change "they are an underestimation" to "they are underestimated"

Lines 283 - 284: remove double parentheses

*Our reply: Thank you for spotting the typos and suggesting better formulations*

Lines 338 – 343: This essentially seems like a repeat of the results in lines 291 – 296. Both paragraphs discuss the results averaged over all sites.

*Our reply: We thank the reviewer for spotting that. We have sorted out the repetitive discussions on the site average ensemble simulated trends and focus analysis in Section 5.2 on the trends from the individual models.*

Line 358: suggest changing "calculated" to "modeled"

Line 397: suggest inserting "between species" after "not the same"

*Our reply: The suggestions above were followed*

Lines 399 – 400: I would suggest converting these results from ugN/m3 and ugS/m3 to ug/m3 for consistency with the analysis of total PM mass

*Our reply: Traditionally, ugN/m3 and ugS/m3 have been for decades used in EMEP for sulphur and nitrogen components in model vs observation comparison, but we agree with the reviewer that it's a good idea to use ug/m3 for all species it this study. We have made the unit conversion and new plots are included.*

*Furthermore, while making the unit conversion, we have spotted that there appeared to be a unit inconsistency in calculations of the contributions of SO4, NO3 and NH4 to PM trends. This has been corrected and the correspondent maps and numbers have been updated in the last manuscript version.*

*Slightly modified text (lines 415-424): "The decrease in $SO_4^{-2}$ concentrations (Figure 10a) played the dominating role over most of the EDT domain, except from parts of Central Europe and Northern Italy. Namely, relatively large contributions of $NO_3^-$ to $PM_{10}$ trends are seen in Germany (and neighbouring parts of France, Czechia and Poland), Denmark, the Netherlands, and in the Po Valley (Figure 10c). The reduction of $NH_4^+$ levels, which includes both ammonium sulphate and ammonium nitrate, appears to be quite an important contributor to the $PM_{10}$ decreasing trends, with the largest effects estimated for Poland, Denmark, and the Po Valley (Figure 10b). The reduction of primary PM emissions was according to the model ensemble simulations the dominating factor for $PM_{10}$ trends in Portugal and southern parts of the Balkan; as well as in many European cities (due to emission reductions from traffic and residential heating) (Figure 10d)."*

Line 453: change "dependency of" to "dependency on"

*Our reply: The suggestion above was followed*

Line 578 and 581: what is the estimated uncertainty of VOC and NH3 emissions?

*Our reply: NMVOC emission estimates in Europe are thought to have an uncertainty of about ±30% due in part to the difficulty in obtaining good emission estimates for some sectors and partly due to the absence of good activity data for some sources ( APE 004Published 15 Oct 2010 Last modified 04 Sep 2015 https://www.eea.europa.eu/data-and-maps/indicators/emissions-of-ozone-precursors-version -2/eea-32-non-methane-volatile-1). NH3 emission estimates in Europe are due largely to the diverse nature of major agricultural sources (https://www.eea.europa.eu/data-and-maps/indicators/eea-32-ammonia-nh3-emissions-1). It is estimated that they are around ±30%. For both NH3 and NMVOC, the trend is likely to be more accurate than the individual absolute annual values. Primary $PM_{2.5}$ and $PM_{10}$ data is said to be of relatively higher uncertainty compared to emission estimates for the secondary PM precursors.*

Lines 588 – 593: The authors may want to explicitly discuss that in the real world, these emissions are dependent on meteorology (higher on colder days) while this effect likely is not represented in the current models.

*Our reply: That's a very relevant point. A brief discussion has been added. It is also worth noting that the models did not account for the dependence of residential heating emissions on the outdoor temperature, i.e. they increase as it gets colder. This may lead to model underestimation of winter pollution episodes, resulting in under-predictions of annual mean PM (as for 2010, see Sec. 5.1).*

Line 589: year is missing for the Simpson et al. reference

*Our reply: Thanks for spotting that - the year is inserted!*

Line 628: what is the authors' interpretation of this finding? Also see my earlier comments on fire and dust emissions.

*Our reply: We have re-written the text on lines 615-628 to provide hopefully a more clear interpretation of the results. Regarding our finding that statement "..large number of sites (and areas) where the models do not estimate significant trends for 2000-2010 period", our explanation is that due to considerable inter-annual variability in PM concentrations and not very strong pM trends, 10-year period is probably not long enough for identification of significant trends with high confidence.*

Line 639: consider changing "downsized" to "partially masked"

*Our reply: The suggestion is accepted.*

Line 665: this section does not include an outlook, consider changing the section title to just "Summary"

***Our reply:*** *Thanks, it's changed.*

**Author comments to Referee #2**

The paper describes the analysis of both particulate matter observations and a multi-model ensemble for two decades of simulation in Europe, showing decreasing trends in particulate matter through both analysis, albeit at different levels of significance for observations versus model ensemble. As such, the work is of interest to the readers of ACP and worth publishing in the journal, subject to the comments and corrections I provide below.

My rating for this paper is minor revisions in that I do not think that the paper requires additional model simulations or detailed analysis at this point in time. However, there are a number of clarifications needed in the paper, as well as some discussion of an important aspect of particle chemistry that is apparently missing from the models used in the ensemble, that will be needed prior to final publication.

I've divided my comments up into "Clarifications Needed" and "Missing Process(?)", then follow with suggested corrections for several spelling and grammar mistakes throughout the paper.

Clarifications Needed (those marked with a * are the ones I feel are the most important).

*Our reply: We would like to thank the referee for valuable (though challenging) questions and relevant comments, as well as his suggestions for editing the text. We have tried to address all issues raised, which we hope has helped to improve the quality of our paper. Our answers to the referee's comments can be found below, in Cursive text format.*

**Background description of the models is missing from the manuscript.**

The authors make use of a 6 member ensemble to analyse the impacts of emissions changes on particulate matter trends in Europe – without providing a summary table in the paper of the main features of the models. A reference to a previous paper is insufficient here – a summary table that allows the reader to see the manner in which each model treats particle formation and deposition is critical, in order to allow the reader to understand the limitations (or lack thereof) of the ensemble. An additional table including references and a paragraph of explanatory text is needed in the final version of the paper, with the table including:

- Gas phase mechanism used in each model (and reference), along with number of species and reactions.
- Particle size distribution representation used in each model (sectional, modal and number of bins or modes, etc.).
- Particle chemical composition used in each model – list of chemical components speciated and/or treated as a lumped species (e.g. "SOA").
- Organic particle formation methodology used in each model. g. Yield approach (reference to yields used), VBS method (reference to specific approach), etc.
- Inorganic particle heterogeneous thermodynamics approach used, along with a reference and the names of the chemical species within the approach used.

- Cloud processing of aerosols: number of aqueous reactions and a reference for same, which particle species names may be formed or removed by cloud processing.
- Particle dry deposition parameterization used and reference for same.
- Wind blown dust algorithm (if the model has one) + reference
- Sea-salt emissions algorithm (if the model has one) + reference
- Mixing state of the aerosols (e.g., homogeneous or heterogeneous mixtures, combinations)

This summary Table is essential for the paper to be publishable – it and the additional text provides a clear description of the models and their potential weak or strong points, and would aid in the subsequent interpretation of model results. It also allows the reader to place the work in the context of other current published material. It can also be referred to at different places in the text to help place some of the findings in the context of the model construction (e.g. discussion of Figure A12, paragraph ending line 445, and a few other places in the text, as I describe below.

*Our reply:* Yes, we agree that it's worthwhile to provide a table summarising the model details directly in this paper. We have included a table (Fig A16) based on the Supplement S1 from Colette et al. (2016), extended with more details regarding aerosol treatment in EDT models, as suggested by the referee.

Additional clarifications needed:

- Line 15: The Abstract mentions 2 and 6 ug m-3 m-3 reduction (I think that should be ug m-3 year-1 ?), but the Abstract not clear if the 2 and 6 are referring to PM10 and PM2.5 respectively, or the range of decreasing trends seen across all models, or the range of decreasing trends across all models for PM10 and PM2.5 respectively. Please clarify.

*Our reply:* Thank you! Indeed ug m-3 m-3 was a typo, which is now corrected to ug m-3 (same in Summary). In that sentence, we refer to the total reduction of both PM10 and PM2.5 across Europe during the studied 11-year period (derived from absolute trend values in Fig. 2a,b). The updated, hopefully more clear sentence, now reads: "The model ensemble simulations indicate overall decreasing trends in PM10 and PM2.5 from 2000 to 2010, with the total reductions of annual mean concentrations by between 2 and 5 (7 for PM10) ug m-3 (or between 10 and 30\%) across most of Europe (by 0.5-2 ug m-3 in Fennoscandia, north-west of Russia and Eastern Europe) during the studied period."

- **\***A general comment: when providing trend levels such as quoting the range from the ensemble, please provide the mean and standard deviation of the ensemble as well (e.g. "between 2 and 6 ug m-3 year-1; mean+/-standard deviation of 4.51 +/- 0.53"). Similarly, for the observations, provide the mean and standard deviation. I'm wondering to what extent the ensemble-estimated variability and observation variability overlap each other: are the differences between model and observed trends significant? The authors have dealt with significance of model and observed trends... but I don't think the extent to which the model trends versus observed

trends are significantly different from each other. Similarly, the figures with time series to show trends (Figure 4, Figure 8) should show the 95% confidence level; (z* sigma)/sqrt(N), where z* is 1.960 for a 95% confidence level) calculated for each year as a shaded region around the lines: do the model and observed trends lie within each other's shaded regions – i.e. are the modelled and observed trends significantly different from each other, given the variability of both the ensemble and the observations?

*Our reply: Thank you for this comment. For model simulated trend maps, Fig 3 shows Coefficient of Variability (COV) as a measure of the inter-model variability in PM trend slopes. COV is discussed in section 5.1 and mentioned in Abstract: "There is a reasonable agreement in PM trends estimated by the individual models, with the inter-model variability below 30-40\% over most of Europe, increasing to 50-60\% in northern and eastern parts of EDT domain. STD was already given for annual trends at the trend-sites in Table 1. As recommended by the referee, we have also included STDs for modelled and observed seasonal trends and through the text where relevant (but not in the Abstract to avoid it getting oversized). Finally, 95% confidence intervals have been added in Fig. 4 and 8, showing PM annual and seasonal timeseries.*

- Line 25: "significant modelled trends" – authors need to define what is meant by "significant" here (e.g at 95% confidence level?).

*Our reply: The explanation is added (at 95% confidence level). The choice of 95% confidence level as a criterion for trend significance is described in Section 2.3.*

- Line 65, "2008-2014": please include a sentence describing the possible reasons for the spread in results noted by previous authors (if they were provided by those authors). Does the current paper address those possible causes of spread in the results (e.g. I think that the attempt to harmonize emissions, and especially to use a common grid resolution, help reduce between-model variability)?

*Our reply: The authors did not focus on explaining the differences, but those model studies were not as harmonised as EDT model simulations. The models mostly used the same emissions, but otherwise had different setups, including different meteorology, grid resolution, boundary conditions for regional models etc., which is certainly an important source of the spread in their results. A comment on this is added in Introduction.*

- Line 125 – 127: the different sources of wind-blown dust information are potentially a large source of variability between models. The authors should estimate the relative magnitude of the annual wind blow dust emissions if possible, and mention the values here, with reference to the new Table requested above. Similarly – can the magnitude of the forest fire emissions differences be quantified here? Ditto for the volcanic SO2 (e.g. compare the mass emitted by volcanoes versus European sources? What I'm hoping for here is some quantitative statement of the potential relative impact of the differences in emissions between the models on the predictions.

***Our reply:*** *The referee raises a very valid question. Indeed, forest fire emissions and SOx from volcanoes were not included in the EDT study. The main reason was that the research focus was to investigate whether the models could reproduce the trends caused by anthropogenic emission changes and changes in meteorology, and at the same time to see if the simulation results reproduce observed PM trends. A comprehensive analysis of the effects of the mentioned natural sources are beyond the scope of this study, but it is absolutely relevant to make comments on this issue in the paper. The referee correctly points to the uncertainties in windblown dust emissions as a serious source of variability between EDT model results. Not only the models used different schemes, some of them (CHIM, MATCH and POLR) did not include windblown dust emissions from within the domain, but only dust from boundary conditions. Unfortunately the available output from the models would not allow estimating annual windblown dust emissions, but it's possible to look at the variability in modelled PM trends, especially in the areas influenced by mineral dust, like in the Mediterranean. Namely, we would expect stronger PM trends and a larger fraction of significant trends from CHIM, MATCH and POLR with respect to those from EMEP, LOTOS, MINNI and from observations, but this does not appear in the results (see e.g. Spanish sites).*

*Based on the data reported to EMEP, total SOx emissions from Italian volcanoes (Etna, Stromboli and Vulcano) were about 4000 Ktonnes in 2000 and 656 Ktonnes in 2010 (compared to around 18473 and 11623 Ktonnes of total anthropogenic SOx emissions used in EDT simulations). For example, EMEP source-receptor simulations indicated a rather limited contribution to PM2.5 in European countries from volcanic emissions (including Eyjafjallajökull eruption) in 2010 (EMEP Status Report 1/2012,* [https://emep.int/publ/common_publications.html#2012](https://emep.int/publ/common_publications.html#2012)). *Regarding forest fires, our additional investigation during the review process has shown that those emissions did not have any considerable effect on PM concentrations on an annual basis, and on modelled PM trends at the EDT sites. Thus, the main findings and conclusions from EDT analysis are considered valid. As an example, the figures below present* EMEP modelled trends for PM2.5 including forest fires emissions (left) and the difference in PM2.5 trends when PM emissions from forest fires are included and excluded (right) (shown are both significant and insignificant trends). The maps show rather tiny differences due to PM emissions from e.g. Portuguese, south-east European and Russian fires on PM2.5 trends. A brief discussion regarding forest fire and volcano emissions have been included in Section 7.2 *Uncertainties in emissions.* More comments regarding forest fires and mineral dust are given in the answers below.

[Figure]

- Line 134: need to mention the sampling interval (hourly, daily?), and frequency (every hour, every day, one-day-in-three, etc., for the PM2.5 and PM10 observations here.

Maybe mention the time span of observations (most seem to be daily averages?) with reference to table A1?

*Our reply: We added "At most of the sites, 24-hourly samples were taken on a daily basis (See Table A2)." Note that a new Table A16 summarizes model details.*

- Line 141: what is meant by "rather many"? Better to state the number of sites out of the total.

*Our reply: That's a good point. The years with missing data are now included in Table A2, and the updated sentences in Section 2.2 is: "Among those 'trend-sites', PM10 observations are available for all eleven years of the 2000-2010 period at 16 sites, and at 4 sites for PM2.5 (Table A2). For most of the sites with incomplete data series, 2000 is a gap-year, as PM monitoring was not started before in 2001 at those sites. The other gap years are: 2009 for the Czech CZ0003R site, 2003 and 2004 for the British GB0043R, and 2009 for PM10 and 2010 for PM2.5 for the Swedish SE0002R (for detailed info see Table A2)."*

- Line 146: Why were there gap years? An explanation is needed. Also, on the time series Figures (4, 8), include a number above each year with the number of stations out of the total (e.g. "5/15") which are being gap-filled in that year, and an addition to the captions explaining those numbers. One of the questions I have is whether some of the differences between model and observations in the trends from one year to the next might be influenced by the gap-filling in the observations. Including those numbers would allow the reader to see the potential influence of gap-filling on the observed trends as a function of time. Please also state the reasons for the measurement gaps in the text, if known. This also applies to the text describing Figures 4 and 8: mention the years that have gaps.

*Our reply: The reason for gap years is either PM was not measured in that year, or the criterion of 75% for data coverage was not satisfied. For most of the sites, 2000 is a gap-year as they had not started PM monitoring before 2001. As the issue with gap filling of the years with missing observations in annual/seasonal timeseries appears controversial, we have decided to drop this fix. The main consequence of dropping gap-filling was an absence of all Spanish sites for 2000 in Figures 4 and 8, which led to lower PM10 and PM2.5 for 2000, with a larger decrease seen in observations. As the gap-filled data set was never used in the calculations of trend slopes and statistics, all quantitative results remain unchanged.*

*The information about the years, for which observations are missing, is now included in Table A2, and we have added an explanatory text in Section 2.2:*
*"The reason for gap years is either PM was not measured in that year, or the criterion of 75% for data coverage was not satisfied. For most of the sites, 2000 is a gap-year as they had not started PM monitoring before 2001. The other gap years are: 2009 at the Czech CZ0003R site, 2003 and 2004 at the British GB0043R, and 2009 for PM10 and 2010 for PM2.5 at the Swedish SE0002R."*
*And also in Section 5.1: "Note that in the year of 2000 is a gap-year at 7 out of 26 sites for PM10 and at 8 out of 13 sites for PM2.5, as described in Section \ref{sec:observations}. In*

*particular, none of Spanish sites are included for 2000, bringing in some inconsistency in site averaged PM10 and PM2.5 annual mean series."*

- Line 158-159: Suggest mentioning here that the extent to which the data series duration, natural variability and weak trends might have affected the authors' analysis is discussed later in the paper. The intent of line 160 (closing sentence) is a bit unclear: maybe "taking into consideration (averaging)" should just be "averaging"?

***Our reply:*** *We followed both suggestions, and the modified text is: "It showed that the chance that the MK method detects the long-term trend decreased for shorter data-series, large natural variability and relatively weak trends. The extent to which these factors could have affected the results of our trend analysis is discussed in section 7.3. Furthermore, the aforementioned document also demonstrates that averaging significant trends only would overestimate mean absolute trends, therefore both significant and insignificant trends have been included when calculating site-average PM trends.*

- Line 177: is this R or R2? Please specify.

***Our reply:*** *R is given. We have specifies that in the text and in Tables A3 and A4*

- Line 217: the model results suggest significant trends in some locations while the observations do not, and vice versa. At this point, the reader is wondering why that might be. Either include a few lines of explanation in this paragraph, or a bridging sentence to the later discussion on causes for differences in significance between model and observed trends.

***Our reply:*** *We do understand what the referee means, still we think it makes sense to keep the first paragraph in the section 5.1, explaining what is shown in Figure 2, as it is, while the discussion on the model vs observation differences comes just two paragraphs later, on the same page.*

- Line 235-236, "large uncertainties in modelling of the coarse fraction of PM": please include some text describing how the different models differ in how this is done (with reference to the Table mentioned above).

***Our reply:*** *We have added a short text with a reference to the new Table A1 with model details: "This reflects larger uncertainties in modelling of the coarse fraction of PM, which is mostly due to natural origin, i.e. sea salt and windblown dust. As shown in Table A1, the models used different parameterisations for those aerosols, also some of them did not include online simulations of windblown dust, but only mineral dust from boundary conditions". More discussion is given in section 5.4 dedicated to PM individual components*

- Line 249: What are the contributing factors to the model differences? Discuss here or add a sentence mentioning where it is discussed later in the paper.

***Our reply:*** *We have mentioned several differences in model formulations (with reference to the new table with model details) and run setup contributing to the differences in model results and referred to Section 5.4 regarding individual PM components. The following text is added: "As most of the input and setup for the model runs were harmonised (Section 2.1), the differences we see here are due to differences in model configurations and process descriptions (see Table A1), leading to different responses of the models to the changes in emissions and inter-annual meteorological variability. Differences in the formulations of secondary aerosol formations (inorganic and organic) can be pointed at as a very important reason for discrepancies in PM modelled trends. Differences in aerosol removal, in particular wet scavenging efficiency, also play a certain role (besides LOTO and MATCH were driven with different meteorology). Further note that the models have a different thickness of the lowest layer which affects the concentrations, removal and transport distances of primary PM and their gaseous precursors."*

- Line 276: Might also be worth noting that the inter-annual variability introduced by forest fires can be a large addition to the net variability. 2010 was also a year in which very large fires occurred in Russia during the summer (late July to mid-August). The extent to which the models have accurately captured these fires may determine the extent to which they simulate PM2.5, PM10 correctly in the trends, especially for eastern Europe. Makar et al, Atm Env. 115, 499-526, 2015, figures 11 and 14. I'm also wondering if the high annual mean PM concentrations in 2003, 2006, 2010 (line 329) might also correspond to high forest fire years.

***Our reply:*** *A good point again! As described in Section 2.1, emissions from forest fires were not accounted for in the model simulations. A discussion on possible effects of this omission on modelled trends is now included in Section 7.2. Based on the results from independent tests with the EMEP model, we do not believe that PM increases in 2003, 2006 and 2010 were due to forest fires as the trend sites are situated in the locations not (or negligibly) affected by the fire emissions. A close look at the timeseries at the individual sites show that the enhanced PM in 2003, 2006 and 2010 were mostly due to high annual PM at Central-European sites (primarily in Germany, Austria, Switzerland), rather far away from major forest fires.Those elevated annual PM were due to heat waves in summers 2003 and 2006 (no PM removal by precipitation in dry weather conditions, efficient BSOA formation) and extensive PM episodes in winter 2010.*

- Line 295: standard deviations of trends across models and obs should be added here; see earlier comment.

***Our reply:*** *Thank you. Added.*

- Line 298 and Figure 5: the station locations are not obvious from the names – can the authors provide an additional panel to this figure showing the station locations as an inset map?

***Our reply:*** *Done!*

- Also Figure 5, Figure 6 and later on line 319: were all of the observed trends significant? The text elsewhere implies this is not the case, but the Figures only show model trend significance levels with two colours. If some of the observed trends were not significant (as seems to be implied in the text), please show this using a similar two-tone red pair of bars for the observations, in addition to the two tone blue bars for the model.

*Our reply: The sites with significant observed trends were highlighted by stars. We have followed the recommendation and in the updated Figures 5 and 6, the sites with non-significant trends are shown as striped bars for both observations and model ensemble.*

- Paragraphs between lines 301 – 306, 307-313, 319-325, 331-335 state the result, but not the possible reasons for it. Why are there differences in variability? Why might the trends be more/less significant at different sites? How might the differences in the models result in the different trends (e.g. with respect to the Table requested above)? The authors have some discussion later in the paper – maybe a sentence mentioning this discussion "The possible causes for these differences are discussed in section …" should be included here.

*Our reply: It is indeed correct that we discuss model vs observations and the inter-model differences in Section 7 (Discussion), and a discussion regarding significant/insignificant trends is given on lines 320-330. Following the referee's suggestion we have added mentioning possible reasons for the results presented in Section 5.2 and included references for further discussions given in Section 7 (see PMtrends_diff.pdf)*

*Also here, we have spotted some erroneous numbers (remaining from the earlier version) which are now corrected: on Line 319 " All in all, the observations show significant PM 10 trends at 14 (corrected to 11) out of 26 sites and…." and on Line 328 "..the German sites..significant observed trends are found for only 3 (corrected to 1) out of 7 sites for PM 10 and for 1 (corrected to none) of 2 sites for PM 2.5".*

- Line 502: "rather moderate" – can this be made more quantitative?

*Our reply: Of course! We have added: "the mean STD between the models is 0.054 ug m-3 yr-1, varying between 0.005 and 0.104 ug m-3 yr-1 for different countries"*

- Lines 505-509: Is there a potential explanation? E.g. significant emissions reductions in the first of the two decades?

*Our reply: Indeed, the emission reductions of PM and their gaseous precursors (except ammonia) were greater in the 1990s compared to those during the 2000s. The explanation is added.*

- Line 535: what are the reasons why Spain might have a higher variability than elsewhere? Local emission sources with high variability?

*Our reply: Good question from the referee.We looked at the annual series at Spanish sites from the individual models and added the following: "Only the EMEP model (and MATCH for PM2.5) simulated significant PM trends for most of Spain, whereas PM trends from the other models were found to be insignificant due to smaller PM decreases from 2000 to 2010 or/and larger inter-annual variability (as in the results from LOTO and MATCH, using different meteorology)."*

- Line 543: Suggest  "spatially" should be "spatially and temporally".

*Our reply: Added as suggested.*

- Lines 624-628: I found these 3 sentences a bit hard to follow.  Clarify?  A lack of significance may be due to the low magnitude of the trends (requiring a larger sample size to identify the trends as significant relative to noise) and/or high magnitude of the variability (e.g. larger standard deviation).  The authors identify the latter as the main reason for the non-significant PM trends (I think), though its really the relative magnitude of variability to trend that matters…

*Our reply: We think the referee and the authors actually mean the same thing, namely that "..that the weaker the trend is relative to the inter-annual meteorological variability, the longer the time series that is needed in order to identify a significant trend" (lines 616-617) . We have re-written the first two paragraphs in Section 7.3 (Effect of inter-annual variability), trying to make it more clear and easier for the reader to understand our message.*

- Line 680: again, better to include ensemble mean and standard deviation rather than just the maximum and minimum of the range.

*Our reply:*  As we replied above, here we describe the maps (spatial distribution) of total reductions of *PM10 and PM2.5 during the studied 11-year period, simulated with the model ensemble, saying what the reductions were in different parts of Europe: "*That would mean that the annual mean PM concentrations decreased by between 2 and 5 (7 for PM10) ug m-3 across most of Europe (by 0.5-2 ug m-3 in Fennoscandia, north-west of Russia and Eastern Europe) during the 2000-2010 period.

**Missing Process (?)

- There are a number of places in the text which imply that the models in the ensemble might not include the reactions of inorganic heterogeneous chemistry associated with base cations (Ca(2+), Mg(2+), Na(+), K(+)). These are sometimes a significant component of mineral dust and sea-salt, and can have a significant impact on particle chemistry, particularly via a competition between the fine mode and coarse mode for nitrate.  The text between lines 395 and 412, and again lines 554-558, mentions secondary inorganic aerosol (SIA) only in the context of the SO4(2-), NH4(+), NO3(-) system – which is incomplete.  This is why I want the model speciation included in the Table requested earlier:  its not clear from the text whether this speciation is included in the ensemble of models – and its absence could potentially have a big

impact on model results. Conversely if some models in the ensemble do include base cation chemistry, then this might also help explain some of the inter-model variability.

*Our reply:* We thank the referee for pointing out this. Whether the models account or not for base cations is definitely very relevant information. This has been added in Table Fig. A16 and pointed out in Section 5.4. We have added in Abstract: "In particular, our results are rather inconclusive regarding the implications of not accounting for forest fires, and also for windblown dust (by some of EDT models) and for nitrate formation on base cations, for PM trend analysis for the period 2000-2010".

- The issue with base cations is that they are "stronger" cations than ammonium, and hence may perturb the balance of nitrate between the fine and coarse modes of the particle distribution. The authors mention NH3 + HNO3 <-> NH4NO3, NH4+,NO3-: to this equilibrium, the base cations add reactions such as CaCO3 + 2 HNO3 <-> Ca(NO3)2 + CO2 + H2O, and NaCl + HNO3 <-> NaNO3 + HCl, with these base cation equilibria being strongly biased towards the right and formation of base cation nitrates. What this can mean (and has been observed in observational studies (see Anlauf et al, Atm Env., 40, 2662-2675, 2006 for a sea-salt example, and Makar et al, JGR-Atm, **https://doi.org/10.1029/98JD00978**, 1998 for a calcium nitrate example), is that the nitrate can off-gas as HNO3 from the fine mode ammonium nitrate particles to go to the coarse mode as base cation nitrates. Models such as CMAQ and GEOS-Chem capture this process through the use of inorganic heterogeneous chemistry solvers that include base cation equilibria, such as Athanasios Nene's ISORROPIA2.
- The absence of this process in some or all of the authors' ensemble of models may help account for some of the differences between model and observed trends. For example, on page 13, line 410-412, the authors mention that "The models appear to overestimate the observed negative trends for NO3- and also for NH4+, though to a smaller degree": this is what I would expect if the models in the ensemble have not included coarse mode base cation chemistry: the base cations are largely coming from sources that have a natural component (wind blown soil dust, sea salt) and hence are not affected by emissions controls on particulate matter. Particle nitrate in the sulphate, ammonium, nitrate –only system will decrease rapidly if both ammonium and NOx are decreasing – however, if base cations are present, they will slow down the nitrate decrease by providing an additional sink other than ammonium – with the result that the ammonium may be in excess and remain as ammonia gas. That is, the decrease in fine mode nitrate may not be as strong if base cations are present.

*Our reply:* As detailed in Table A1, the models adopt different thermodynamic aerosol models, and none of them included base cations in their aerosol/gas partitioning schemes for fine aerosols (actually among the thermodynamic models used in the EDT models, only ISORROPIA allows including base cations). Four out of six models simulated coarse NO3 (CHIM and LOTO include NO3 formation on Na, and EMEP and MATCH used reaction rates for HNO3 -> coarse NO3 irrespective of base cation availability). The differences in both formulations of fine and coarse NO3 formation contribute to the differences in model results,

increasing (or maybe decreasing in case of compensating effects) the inter-model variability. An in-depth analysis of the trends in the individual aerosol species was not in the scope of this manuscript (more details regarding SIA trends can be found in Ciarelli et al. 2019 and Theobald et al., 2019), still this interplay between anthropogenic aerosol/gas pollution and natural particles is indeed important to be mentioned. In order to address the referee's request we have taken a look at NO3 trends from the individual EDT models (Figure R2 below).

As CHIM and LOTO include coarse NO3 formation on sea salt particles, we could expect to see smaller NO3NH4 trends from these two models in coastal regions. The formation rate of coarse NO3 in EMEP and MATCH was proportional to HNO3, thus should follow the trends in NOx emissions. Regarding models vs. observations discrepancies, we should see the largest effects of models not accounting for base cations in coastal areas influenced by sea salt and southern parts of Europe/Mediterranean influenced by African dust, and other places affected by windblown dust generated in some arid areas and arable lands within European domain. However, we could not see any consistent differences in NO3 (and NH4, not shown here) relative trends between the models with/without coarse NO3 (see Figure R2). Though NO3 relative trends from CHIM and LOTO are indeed weaker than those from EMEP and MATCH, the trends from MINNI and POLR (not accounting for coarse NO3) are rather different. Neither the results of comparison of NO3 and NH4 trends from the individual models with observations appear conclusive, i.e. for sites experiencing sea salt influence, though NO3 trends from CHIM and LOTO are closer to observations at Dutch and Norwegian sites, EMEP and MATCH seem to do better for Swedish sites. At the Italian Montelibretti (which could experience both sea salt and dust influence), EMEP, LOTO and POLR (not accounting for any base cations) appear to agree better with observations. Here again. Note rather sparse NO3 data available for NO3 trends analysis (no Spanish data and practically no data for Mediterranean region, where mineral dust plays an important role). Furthermore, it can be noticed in Figure R2 that the trend maps for the individual models accounting for coarse NO3 are quite similar for NO3 in PM2.5 (NO3-25) and (NO3 in PM10) NO3-10. This indicates that the inter-model differences in NO3 trends are most likely associated with ammonium nitrate. Finally, we think it would be interesting to perform a dedicated study on the effect of base cations from natural emissions on PM changes due to anthropogenic emission control. We agree with this reasoning, but unfortunately it is not feasible to verify this due to the lack of long-term observations of fine nitrate.

- Line 434-435 "Thus, as the formation of ammonium sulphate…" while the ammonium was becoming more available, it won't necessarily result in ammonium nitrate formation, if the base cations are in excess to remove the available nitric acid. The models are showing what would happen in a base-cation-less world, I suspect.

*Our reply:* We agree with this referee's comments. The following sentence has been added: "However it should be kept in mind that in the regions influenced by mineral dust and/or sea salt, some of nitric acid would be consumed in the formation of NO3 associated with base cations (as discussed above, this is not fully accounted for in the EDT models), so that less NH4NO3 would be formed compared to what the EDT models simulate".

- The authors should discuss the base cation issue, in the context of the model speciation Table requested above, and as an addition to the SIA analysis (lines 395 – 412). The nitrate chemistry of the models may not be simulating these effects (sounds like it, from lines mentioned above and 456-458) – so this should be acknowledged as a source of uncertainty in the analysis and the results. Conversely, if some of the models in the ensemble do include base cation chemistry – does this explain some of the differences between those models and others in the ensemble?

*Our reply:* Yes, it's indeed important to discuss this issue. The following test has been included:"It should be noted that none of the models accounts base cations (i.e. $Ca(2+)$, $Mg(2+)$, $Na(+)$, $K(+)$) in gas-aerosol partitioning of $HNO3$ (see Table "Model description"). Those base cations are significant components of sea salt and mineral dust. They participate in aerosol chemistry and facilitate the formation of coarse $NO3(-)$, consuming $HNO3$ and thus making less of it available for $NH4NO3$ formation. As the emissions of sea salt and mineral dust strongly depend on meteorology (especially on surface wind speed), $NO3(-)$ formed on the base cations (and consequently total $NO3(-)$) is subject to inter-annual variability, that could weaken $NO3(-)$ trends and lead to a larger fraction of insignificant trends. Thus, not including base cations in aerosol chemistry could be one reason for models' overestimating of the observed $NO3(-)$ trends (see also discussion in Section "Discussion"}. Among the EDT models, MINNI and POLR did not included coarse $NO3(-)$, CHIM and LOTO included $NO3(-)$ formation on sea salt \chem{Na^{+}}, while EMEP and MATCH used constant reaction rates for coarse $NO3(-)$ formation from $HNO3$, irrespective of base cation availability (Table ~\ref{apfig:models}). However, we could not see any consistent differences in the relative trends of $NO3(-)$ and $NH4(+)$ between the models with and without coarse $NO3(-)$ (not shown here), neither the comparison of $NO3(-)$ trends from the individual models with observations at the rather limited number of sites gave conclusive results.

- Comment on paragraph ending line 427: similar effects have been observed in North America, I think: as the sulphate decreases, the available ammonia in the fine mode is more likely to allow $HNO3$ to enter the fine mode, as long as base cations are not present as an alternative sink for $HNO3$, and/or there's sufficient $HNO3$ to replace both the anions in the coarse mode and charge balance the excess ammonium in the fine mode. The extent to which ammonia is increasing or decreasing may also play a role. The authors should have a look at the analysis by Robert Vet et al (Atm. Env., 93, 3-100, 2014), particularly Section 4, Figures 4.9 and 4.10 and related text, and the analyses carried out for N in Europe in that paper.

*Our reply:* Again, we agree (see our answer above) Also we have mentioned this source of uncertainties in EDT simulations in Summary: "Among possible reasons for deviations between the modelled and observed PM trends are emission uncertainties, impacts of inter-annual variability in meteorological conditions (on pollutant transport and removal, secondary aerosol formation, natural PM emissions etc.), model uncertainties associated with aerosol formation and removal processes, i.e. SOA formation, cloud pH dependency of $SO4$ formation, **heterogeneous chemistry (including gas/aerosol partitioning of anthropogenic precursors and aerosol formation on base cations of natural origin)**, $SO2$ and $NH3$ co-deposition etc..

- Lines 452-453: "only BSOA have some dependency on anthropogenic emissions" – not true: wind-blown dust and sea-salt can be a significant sink for nitric acid resulting from anthropogenic NOx emissions.

*Our reply: This is an interesting issue of the interplay between anthropogenic and natural sources and sinks, complicating aerosol trends due man-made emission reduction, which certainly should be mentioned in the manuscript. Namely, that the changes in NO3 formed from anthropogenic NOx are also dependent on natural sea salt and mineral dust, whereas biogenic SOA formation is affected by the presence of anthropogenic organic aerosols. This is mentioned now in Section 5.4 and also in Summary and Abstract.*
*Still, we think that the sentence about BVOC was not wrong, as it referred to natural aerosol formation. We have made this more clear: "Among natural aerosols, only formation of BSOA has some dependency on anthropogenic emissions…."*

- Line 640: note that inorganic heterogeneous chemistry is also highly dependent on meteorological conditions, particularly the temperature, with particle nitrate formation equilibria being biased towards particulate phase by a factor of 1E6 for a 25C drop in temperature.

*Our reply: Thank you for pointing this out. We have included in Section 7.3 (Inter-annual variability) the following sentence: Most of aerosol processes (some emissions, gaseous and especially heterogeneous chemistry, transport and removal) depend on the meteorological conditions.*

Spelling/Grammar corrections:

- Line 6: "do" and "how" should have the first letter capitalized.
- Line 60: "published last years" should be "published recently".
- Line 63: "analysis" should be "analyses"
- Line 76: "prior 2000" should be "prior to 2000"
- Line 80: "obtained under the controlled setup" is a bit unclear in the context of the rest of the sentence, maybe "under this controlled setup"?
- Line 118: "shipping Russia" should be "shipping for Russia"
- Line 141: "4 site" should be "4 sites"
- Line 283 to 284: double brackets unnecessary.
- Line 349: shouldn't "insignificant" be "significant" here? Looks like it, from the context.
- Line 415: "individual aerosols" have been used by the authors where I think they mean "individual chemical components of the aerosols"

*Our reply: all suggestions above have been attended to*

- Line 464: explain where the 3.26 x sea salt Na formula originated?

*Our reply: the explanation was added: "assuming 30.7 \% sodium content in sea salt aerosols, same as in sea water"*

- Line 534: "Besides, the emissions of primary PM2.5 went up in the same period." would be better as "The emissions of primary PM2.5 in Poland increased during the same period."
- Line 624: "by far and large below" is unclear – maybe "considerably below" or "significantly below"?
- Line 639: Maybe "significance" or "apparent significance" of emission reduction might be better than "effect of emission reduction" here?

*Our reply: all suggestions above have been attended to*

- Tables A2, A3: not clear why the number of significant figures in the bias changes, e.g. -2 versus -16 would be better as -2.0 versus -16, or use scientific notation.

*Our reply: Done according to the referee's suggestion.*

**To discussion regarding NO3 and base cations:**